# Offline Reinforcement Learning with Differentiable Function Approximation is Provably Efficient

**Ming Yin**[*] , **Mengdi Wang**[†] , **Yu-Xiang Wang**[*]
[*]Department of Computer Science
 University of California, Santa Barbara
{ming_yin,yuxiangw}@cs.ucsb.edu

[†]Department of Electrical and Computer Engineering
Princeton University
mengdiw@princeton.edu

## Abstract

*Offline reinforcement learning*, which aims at optimizing sequential decision-making strategies with historical data, has been extensively applied in real-life applications. *State-Of-The-Art* algorithms usually leverage powerful function approximators (*e.g.* neural networks) to alleviate the sample complexity hurdle for better empirical performances. Despite the successes, a more systematic understanding of the statistical complexity for function approximation remains lacking. Towards bridging the gap, we take a step by considering offline reinforcement learning with *differentiable function class approximation* (DFA). This function class naturally incorporates a wide range of models with nonlinear/nonconvex structures. We show offline RL with differentiable function approximation is provably efficient by analyzing the *pessimistic fitted Q-learning* (PFQL) algorithm, and our results provide the theoretical basis for understanding a variety of practical heuristics that rely on Fitted Q-Iteration style design. In addition, we further improve our guarantee with a tighter instance-dependent characterization. We hope our work could draw interest in studying reinforcement learning with differentiable function approximation beyond the scope of current research.

## 1 Introduction

*Offline reinforcement learning* (Lange et al., 2012; Levine et al., 2020) refers to the paradigm of learning a policy in the sequential decision making problems, where only the logged data are available and were collected from an unknown environment (*Markov Decision Process* / MDP). Inspired by the success of scalable supervised learning methods, modern reinforcement learning algorithms (*e.g.* Silver et al. (2017)) incorporate high-capacity function approximators to acquire generalization across large state-action spaces and have achieved excellent performances along a wide range of domains. For instance, there are a huge body of deep RL-based algorithms that tackle challenging problems such as the game of Go and chess (Silver et al., 2017; Schrittwieser et al., 2020), Robotics (Gu et al., 2017; Levine et al., 2018), energy control (Degrave et al., 2022) and Biology (Mahmud et al., 2018; Popova et al., 2018). Nevertheless, practitioners also noticed that algorithms with general function approximators can be quite data inefficient, especially for deep neural networks where the models may require million of steps for tuning the large number of parameters they contain.[1]

On the other hand, statistical analysis has been actively conducted to understand the sample/statistical efficiency for reinforcement learning with function approximation, and fruitful results have been achieved under the respective model representations (Munos, 2003; Chen and Jiang, 2019; Yang and Wang, 2019; Du et al., 2019; Sun et al., 2019; Modi et al., 2020; Jin et al., 2020b; Ayoub et al., 2020; Zanette et al., 2020; Jin et al., 2021a; Du et al., 2021; Jin et al., 2021b; Zhou et al., 2021a; Xie et al., 2021a; Min et al., 2021; Nguyen-Tang et al., 2022; Li et al., 2021; Zanette et al., 2021; Yin et al., 2022; Uehara et al., 2022; Cai et al., 2022). However, most works consider *linear* model approximators (*e.g.* linear (mixture) MDPs) or its variants. While the explicit linear

---

[1]Check Arulkumaran et al. (2017) and the references therein for an overview.

structures make the analysis trackable (linear problems are easier to analyze), they are unable to reveal the sample/statistical complexity behaviors of practical algorithms that apply powerful function approximations (which might have complex structures).

In addition, there is an excellent line of works tackling provably efficient offline RL with general function approximation (*e.g.* (Chen and Jiang, 2019; Xie et al., 2021a; Zhan et al., 2022)). Due to the generic function approximation class considered, those complexity bounds are usually expressed in the standard worst-case fashion $O(V_{\max}^2 \sqrt{\frac{1}{n}})$ which lack the characterizations of individual instance behaviors. However, as mentioned in Zanette and Brunskill (2019), practical reinforcement learning algorithms often perform far better than what these problem-independent bounds would suggest.

These observations motivate us to consider function approximation schemes that can help address the existing limitations. In particular, in this work we consider offline reinforcement learning with *differentiable function class* approximations. Its definition is given in below.

**Definition 1.1** (Parametric Differentiable Function Class)**.** *Let $\mathcal{S}, \mathcal{A}$ be arbitrary state, action spaces and a feature map $\phi(\cdot, \cdot) : \mathcal{S} \times \mathcal{A} \to \Psi \subset \mathbb{R}^m$. The parameter space $\Theta \in \mathbb{R}^d$. Both $\Theta$ and $\Psi$ are compact spaces. Then the parametric function class (for a model $f : \mathbb{R}^d \times \mathbb{R}^m \to \mathbb{R}$) is defined as*

$$\mathcal{F} := \{f(\theta, \phi(\cdot, \cdot)) : \mathcal{S} \times \mathcal{A} \to \mathbb{R}, \theta \in \Theta\}$$

*that satisfies differentiability/smoothness condition: 1. for any $\phi \in \mathbb{R}^m$, $f(\theta, \phi)$ is third-time differentiable with respect to $\theta$; 2. $f, \partial_\theta f, \partial_{\theta, \theta}^2 f, \partial_{\theta, \theta, \theta}^3 f$ are jointly continuous for $(\theta, \phi)$.*

**Remark 1.2.** *Differentiable Function Class was recently proposed for studying Off-Policy Evaluation (OPE) Problem (Zhang et al., 2022a) and we adopt it here for the policy learning task. Note by the compactness of $\Theta$, $\Psi$ and continuity, there exists constants $C_\Theta, B_\mathcal{F}, \kappa_1, \kappa_2, \kappa_3 > 0$ that bounds: $\|\theta\|_2 \le C_\Theta, |f(\theta, \phi(s, a))| \le B_\mathcal{F}, \|\nabla_\theta f(\theta, \phi(s, a))\|_2 \le \kappa_1, \|\nabla_{\theta\theta}^2 f(\theta, \phi(s, a))\|_2 \le \kappa_2, \|\nabla_{\theta\theta\theta}^3 f(\theta, \phi(s, a))\|_2 \le \kappa_3$ for all $\theta \in \Theta, \ s, a \in \mathcal{S} \times \mathcal{A}$.*[2]

**Why consider differentiable function class (Definition 1.1)?** There are two main reasons why differentiable function class is worth studying for reinforcement learning.

- Due to the limitation of statistical tools, existing analysis in reinforcement learning usually favor basic settings such as *tabular MDPs* (where the state space and action space are finite (Azar et al., 2013; 2017; Sidford et al., 2018; Jin et al., 2018; Cui and Yang, 2020; Agarwal et al., 2020; Yin et al., 2021a;b; Li et al., 2020; Ren et al., 2021; Xie et al., 2021b; Li et al., 2022; Zhang et al., 2022b; Qiao et al., 2022; Cui and Du, 2022)) or linear MDPs (Yang and Wang, 2020; Jin et al., 2020b; Wang et al., 2020; Jin et al., 2021b; Ding et al., 2021; Wang et al., 2021a; Min et al., 2021) / linear Mixture MDPs (Modi et al., 2020; Cai et al., 2020; Zhang et al., 2021a; Zhou et al., 2021b;a) (where the transition dynamic admits linear structures) so that well-established techniques (*e.g.* from linear regression) can be applied. In addition, subsequent extensions are often based on linear models (*e.g.* Linear Bellman Complete models (Zanette et al., 2020) and Eluder dimension (Russo and Van Roy, 2013; Jin et al., 2021a)). Differentiable function class strictly generalizes over the previous popular choices, *i.e.* by choosing $f(\theta, \phi) = \langle \theta, \phi \rangle$ or specifying $\phi$ to be one-hot representations, and is far more expressive as it encompasses nonlinear approximators.

- Practically speaking, the flexibility of selecting model $f$ provides the possibility for handling a variety of tasks. For instance, when $f$ is specified to be neural networks, $\theta$ corresponds to the weights of each network layers and $\phi(\cdot, \cdot)$ corresponds to the state-action representations (which is induced by the network architecture). When facing with easier tasks, we can deploy simpler model $f$ such as polynomials. Yet, our statistical guarantee is not affected by the specific choices as we can plug the model $f$ into Theorem 3.2 to obtain the respective bounds (we do not need separate analysis for different tasks).

## 1.1 RELATED WORKS

**Reinforcement learning with function approximation.** RL with function approximation has a long history that can date back to Bradtke and Barto (1996); Tsitsiklis and Van Roy (1996). Later,

---

[2]Here $\|\nabla_{\theta\theta\theta}^3 f(\theta, \phi(s, a))\|_2$ is defined as the 2-norm for 3-$d$ tensor and in the finite horizon setting we simply instantiate $\mathcal{B}_\mathcal{F} = H$.

| Algorithm | Assumption | Suboptimality Gap $v^\star - v^{\widehat{\pi}}$ |
|---|---|---|
| VFQL, Theorem 3.1 | Concentrability 2.2 | $\sqrt{C_{\text{eff}}}H \cdot \sqrt{\frac{H^2 d + \lambda C_\Theta^2}{K}} + \sqrt[4]{\frac{H^3 d \epsilon_{\mathcal{F}}}{K}} + \sqrt{C_{\text{eff}} H^3 \epsilon_{\mathcal{F}}} + H \epsilon_{\mathcal{F}}$ |
| PFQL, Theorem 3.2 | Uniform Coverage 2.3 | $\sum_{h=1}^{H} 16dH \cdot \mathbb{E}_{\pi^\star} \left[ \sqrt{\nabla_\theta^\top f(\theta_h^\star, \phi(s_h, a_h)) \Sigma_h^{\star-1} \nabla_\theta f(\theta_h^\star, \phi(s_h, a_h))} \right]$ |
| VAFQL, Theorem 4.1 | Uniform Coverage 2.3 | $16d \cdot \sum_{h=1}^{H} \mathbb{E}_{\pi^\star} \left[ \sqrt{\nabla_\theta^\top f(\theta_h^\star, \phi(s_h, a_h)) \Lambda_h^{\star-1} \nabla_\theta f(\theta_h^\star, \phi(s_h, a_h))} \right]$ |

Table 1: Suboptimality gaps for different algorithms with differentiable function class 1.1. Here we omit the higher order term for clear comparison. With Concentrability, we can only achieve the worst case bound that does not explicit depend on the function model $f$. With the stronger uniform coverage 2.3, better instance-dependent characterizations become available. Here $C_{\text{eff}}$ is in 2.2, $\Sigma^\star$ in 3.2, $\Lambda^\star$ in 4.1 and $\epsilon_{\mathcal{F}}$ in 2.1.

it draws significant interest for the finite sample analysis (Jin et al., 2020b; Yang and Wang, 2019). Since then, people put tremendous efforts towards generalizing over linear function approximations and examples include Linear Bellman complete models (Zanette et al., 2020), Eluder dimension (Russo and Van Roy, 2013; Jin et al., 2021a), linear deterministic $Q^\star$ (Wen and Van Roy, 2013) or Bilinear class (Du et al., 2021). While those extensions are valuable, the structure conditions assumed usually make the classes hard to track beyond the linear case. For example, the practical instances of Eluder Dimension are based on the linear-in-feature (or its transformation) representations (Section 4.1 of Wen and Van Roy (2013)). As a comparison, differentiable function class contains a range of functions that are widely used in practical algorithms (Riedmiller, 2005).

**Offline RL with general function approximation (GFA).** Another interesting thread of work considered offline RL with general function approximation (Ernst et al., 2005; Chen and Jiang, 2019; Liu et al., 2020; Xie et al., 2021a) which only imposes realizability and completeness/concentrability assumptions. The major benefit is that the function hypothesis can be arbitrary with no structural assumptions and it has been shown that offline RL with GFA is provably efficient. However, the generic form of functions in GFA makes it hard to go beyond worst-case analysis and obtain fine-grained instance-dependent learning bounds similar to those under linear cases. On the contrary, our results with DFA can be more problem adaptive by leveraging gradients and higher order information. In addition to the above, there are more connected works. Zhang et al. (2022a) first considers the differentiable function approximation (DFA) for the off-policy evaluation (OPE) task and builds the asymptotic theory, Fan et al. (2020) analyzes the *deep Q-learning* with the specific ReLu activations, and Kallus and Uehara (2020) considers semi-parametric / nonparametric methods for offline RL (as opposed to our parametric DFA in 1.1). These are nice complementary studies to our work.

**Our contribution.** We provide the first Instance-dependent offline learning bound under non-linear function approximation. Informally, we show that (up to a lower order term) the natural complexity measure is proportional to $\sum_{h=1}^{H} \mathbb{E}_{\pi^\star, h} \left[ \sqrt{g_\theta(s, a)^\top \Sigma_h^{-1} g_\theta(s, a)} \right]$ where $g_\theta(s, a) := \nabla f(\theta, \phi(s, a))$ is the gradient *w.r.t.* the parameter $\theta^\star$ at feature $\phi$ and $\Sigma_h = \sum_i g(s_{i,h}, a_{i,h}) g(s_{i,h}, a_{i,h})^\top$ is the Fisher information matrix of the observed data at $\widehat{\theta}$. This is achieved by analyzing the *pessimistic fitted Q-learning* (PFQL) algorithm (Theorem 3.2). In addition, we further analyze its variance-reweighting variant, which recovers the variance-dependent structure and can yield faster sample convergence rate. Last but not least, existing offline RL studies with tabular models, linear models and GLM models can be directly indicated by the appropriate choice of our model $\mathcal{F}$.

## 2 PRELIMINARIES

**Episodic Markov decision process.** Let $M = (\mathcal{S}, \mathcal{A}, P, r, H, d_1)$ to denote a finite-horizon *Markov Decision Process* (MDP), where $\mathcal{S}$ is the arbitrary state space and $\mathcal{A}$ is the arbitrary action space which can be infinite or continuous. The transition kernel $P_h : \mathcal{S} \times \mathcal{A} \mapsto \Delta^\mathcal{S}$ ($\Delta^\mathcal{S}$ represents a distribution over states) maps each state action$(s_h, a_h)$ to a probability distribution $P_h(\cdot | s_h, a_h)$ and $P_h$ can be different for different $h$ (time-inhomogeneous). $H$ is the planning horizon and $d_1$ is the initial state distribution. Besides, $r : \mathcal{S} \times \mathcal{A} \mapsto \mathbb{R}$ is the mean reward function satisfying $0 \le r \le 1$. A policy $\pi = (\pi_1, \ldots, \pi_H)$ assigns each state $s_h \in \mathcal{S}$ a probability distribution over actions by

mapping $s_h \mapsto \pi_h(\cdot|s_h) \,\forall h \in [H]$ and induces a random trajectory $s_1, a_1, r_1, \ldots, s_H, a_H, r_H, s_{H+1}$ with $s_1 \sim d_1, a_h \sim \pi(\cdot|s_h), s_{h+1} \sim P_h(\cdot|s_h, a_h), \forall h \in [H]$.

Given a policy $\pi$, the $V$-value functions and state-action value function (Q-functions) $Q_h^\pi(\cdot, \cdot) \in \mathbb{R}^{S \times A}$ are defined as: $V_h^\pi(s) = \mathbb{E}_\pi[\sum_{t=h}^H r_t|s_h = s]$, $Q_h^\pi(s, a) = \mathbb{E}_\pi[\sum_{t=h}^H r_t|s_h, a_h = s, a]$, $\forall s, a, h \in \mathcal{S}, \mathcal{A}, [H]$. The Bellman (optimality) equations follow $\forall h \in [H], s, a \in \mathcal{S} \times \mathcal{A}: Q_h^\pi(s, a) = r_h(s, a) + \int_{\mathcal{S}} V_{h+1}^\pi(\cdot) dP_h(\cdot|s, a)$, $V_h^\pi(s) = \mathbb{E}_{a \sim \pi_h(s)}[Q_h^\pi(s, a)], Q_h^\star(s, a) = r_h(s, a) + \int_{\mathcal{S}} V_{h+1}^\star(\cdot) dP_h(\cdot|s, a)$, $V_h^\star(s) = \max_a Q_h^\star(s, a)$. We define Bellman operator $\mathcal{P}_h$ for any function $V \in \mathbb{R}^{\mathcal{S}}$ as $\mathcal{P}_h(V) = r_h + \int_{\mathcal{S}} V dP_h$, then $\mathcal{P}_h(V_{h+1}^\pi) = Q_h^\pi$ and $\mathcal{P}_h(V_{h+1}^\star) = Q_h^\star$. The performance measure is $v^\pi := \mathbb{E}_{d_1}[V_1^\pi] = \mathbb{E}_{\pi, d_1}\left[\sum_{t=1}^H r_t\right]$. Lastly, the induced state-action marginal occupancy measure for any $h \in [H]$ is defined to be: for any $E \subseteq \mathcal{S} \times \mathcal{A}$, $d_h^\pi(E) := \mathbb{E}[(s_h, a_h) \in E|s_1 \sim d_1, a_i \sim \pi(\cdot|s_i), s_i \sim P_{i-1}(\cdot|s_{i-1}, a_{i-1}), 1 \leq i \leq h]$ and $\mathbb{E}_{\pi, h}[f(s, a)] := \int_{\mathcal{S} \times \mathcal{A}} f(s, a) d_h^\pi(s, a) ds da$.

**Offline Reinforcement Learning.** The goal of Offline RL is to learn the policy $\pi^\star := \arg\max_\pi v^\pi$ using only the historical data $\mathcal{D} = \{(s_h^\tau, a_h^\tau, r_h^\tau, s_{h+1}^\tau)\}_{\tau \in [K]}^{h \in [H]}$. The data generating behavior policy is denoted as $\mu$. In the offline regime, we have neither the knowledge about $\mu$ nor the access to further exploration for a different policy. The agent is asked to find a policy $\widehat{\pi}$ such that $v^\star - v^{\widehat{\pi}} \leq \epsilon$ for the given batch data $\mathcal{D}$ and a specified accuracy $\epsilon > 0$.

## 2.1 ASSUMPTIONS

Function approximation in offline RL requires sufficient expressiveness of $\mathcal{F}$. In fact, even under the *realizability* and *concentrability* conditions, sample efficient offline RL might not be achievable (Foster et al., 2021). Therefore, under the differentiable function setting (Definition 1.1), we make the following assumptions.

**Assumption 2.1** (Realizability+Bellman Completeness). *The differentiable function class $\mathcal{F}$ in Definition 1.1 satisfies:*

- *Realizability: for optimal $Q_h^\star$, there exists $\theta_h^\star \in \Theta$ such that $Q_h^\star(\cdot, \cdot) = f(\theta_h^\star, \phi(\cdot)) \,\forall h$;*

- *Bellman Completeness: Let $\mathcal{G} := \{V(\cdot) \in \mathbb{R}^{\mathcal{S}} : such that \|V\|_\infty \leq H\}$. Then in this case $\sup_{V \in \mathcal{G}} \inf_{f \in \mathcal{F}} \|f - \mathcal{P}_h(V)\|_\infty \leq \epsilon_{\mathcal{F}}$ for some $\epsilon_{\mathcal{F}} \geq 0$.*

*Realizability* and *Bellman Completeness* are widely adopted in the offline RL analysis with general function approximations (Chen and Jiang, 2019; Xie et al., 2021a) and Assumption 2.1 states its differentiable function approximation version. There are other forms of completeness, *e.g.* optimistic closure defined in Wang et al. (2021b).

**Data coverage assumption.** Furthermore, in the offline regime, it is known that function approximation cannot be sample efficient for learning a $\epsilon$-optimal policy without data-coverage assumptions when $\epsilon$ is small (*i.e.* high accuracy) (Wang et al., 2021a). In particular, we consider two types of coverage assumptions and provide guarantees for them separately.

**Assumption 2.2** (Concentrability Coverage). *For any fixed policy $\pi$, define the marginal state-action occupancy ratio as $d_h^\pi(s, a)/d_h^\mu(s, a) \,\forall s, a$. Then the concentrability coefficient is defined as $C_{eff} := \sup_\pi \sup_{h \in [H]} \|d_h^\pi/d_h^\mu\|_{2, d_h^\mu}^2$, where $\|g(\cdot, \cdot)\|_{2, d^\mu} := \sqrt{\mathbb{E}_{d^\mu}[g(\cdot, \cdot)^2]}$ and $C_{eff} < \infty$.*

This is the standard coverage assumption that has has been widely assumed in (Ernst et al., 2005; Szepesvári and Munos, 2005; Chen and Jiang, 2019; Xie and Jiang, 2020a), and 2.2 is fully characterized by the MDPs. In addition, we can make an alternative assumption 2.3 that depends on both the MDPs and the function approximation class $\mathcal{F}$.[3] It assumes a curvature condition for $\mathcal{F}$.

**Assumption 2.3** (Uniform Coverage). *We have $\forall h \in [H]$, there exists $\kappa > 0$,*

- $\mathbb{E}_{\mu, h}\left[(f(\theta_1, \phi(\cdot, \cdot)) - f(\theta_2, \phi(\cdot, \cdot)))^2\right] \geq \kappa \|\theta_1 - \theta_2\|_2^2, \,\, \forall \theta_1, \theta_2 \in \Theta; (\star)$

- $\mathbb{E}_{\mu, h}\left[\nabla f(\theta, \phi(s, a)) \cdot \nabla f(\theta, \phi(s, a))^\top\right] \succ \kappa I, \forall \theta \in \Theta. (\star\star)$

---

[3]Generally speaking, 2.2 and 2.3 are not directly comparable. However, for the specific function class $f = \langle \theta, \phi \rangle$ with $\phi = \mathbf{1}(s, a)$ and tabular MDPs, it is easy to check 2.3 is strong than 2.2.

In the linear function approximation regime, Assumption 2.3 reduces to 2.4 since ($\star$) and ($\star\star$) are identical assumptions. Concretely, if $f(\theta, \phi) = \langle\theta, \phi\rangle$, then ($\star$) $\mathbb{E}_{\mu,h}[(f(\theta_1, \phi(\cdot, \cdot)) - f(\theta_2, \phi(\cdot, \cdot)))^2] = (\theta_1 - \theta_2)^\top \mathbb{E}_{\mu,h}[\phi(\cdot, \cdot)\phi(\cdot, \cdot)^\top](\theta_1 - \theta_2) \geq \kappa\|\theta_1 - \theta_2\|_2^2 \,\forall\theta_1, \theta_2 \in \Theta \Leftrightarrow 2.4 \Leftrightarrow (\star\star)\mathbb{E}_{\mu,h}\left[\nabla f(\theta, \phi(s,a)) \cdot \nabla f(\theta, \phi(s,a))^\top\right] \succ \kappa I$. Therefore, 2.3 can be considered as a natural extension of 2.4 for differentiable class. We do point that 2.3 can be violated for function class $\mathcal{F}$ that is "not identifiable" by the data distribution $\mu$ (*i.e.*, there exists $f(\theta_1), f(\theta_2) \in \mathcal{F}, \theta_1 \neq \theta_2$ s.t. $\mathbb{E}_{\mu,h}[(f(\theta_1, \phi(\cdot, \cdot)) - f(\theta_2, \phi(\cdot, \cdot)))^2] = 0$). Nevertheless, there are representative non-linear differentiable classes (*e.g.* generalized linear model (GLM)) satisfying 2.3.

**Example 2.4** (Linear function coverage assumption (Wang et al., 2021a; Min et al., 2021; Yin et al., 2022; Xiong et al., 2022)). $\Sigma_h^{\text{feature}} := \mathbb{E}_{\mu,h}\left[\phi(s,a)\phi(s,a)^\top\right] \succ \kappa I \,\forall h \in [H]$ *with some* $\kappa > 0$.

**Example 2.5** (offline generalized linear model (Li et al., 2017; Wang et al., 2021b)). *For a known feature map* $\phi : \mathcal{S} \times \mathcal{A} \to \mathcal{B}_d$ *and link function* $f : [-1, 1] \mapsto [-1, 1]$ *the class of GLM is* $\mathcal{F}_{\text{GLM}} := \{(s,a) \mapsto f(\langle\phi(s,a), \theta\rangle) : \theta \in \Theta\}$ *satisfying* $\mathbb{E}_{\mu,h}\left[\phi(s,a)\phi(s,a)^\top\right] \succ \kappa I$. *Furthermore,* $f(\cdot)$ *is either monotonically increasing or decreasing and* $0 < \kappa \leq |f'(z)| \leq K < \infty, |f''(z)| \leq M < \infty$ *for all* $|z| \leq 1$ *and some* $\kappa, K, M$. *Then* $\mathcal{F}_{\text{GLM}}$ *satisfies* 2.3, *see Appendix B*.

## 3 DIFFERENTIABLE FUNCTION APPROXIMATION IS PROVABLY EFFICIENT

In this section, we present our solution for offline reinforcement learning with differentiable function approximation. As a warm-up, we first analyze the *vanilla fitted Q-learning* (VFQL, Algorithm 2), which only requires the concentrability Assumption 2.2. The algorithm is presented in Appendix I.

**Theorem 3.1.** *Choose* $0 < \lambda \leq 1/2C_\Theta^2$ *in Algorithm 2. Suppose Assumption 2.1,2.2. Then if* $K \geq \max\left\{512\frac{\kappa_1^4}{\kappa^2}\left(\log(\frac{2Hd}{\delta}) + d\log(1 + \frac{4\kappa_1^3\kappa_2 C_\Theta K^3}{\lambda^2})\right), \frac{4\lambda}{\kappa}\right\}$, *with probability* $1 - \delta$, *the output* $\widehat{\pi}$ *of VFQL guarantees:* $v^\star - v^{\widehat{\pi}} \leq \sqrt{C_{\text{eff}}}H \cdot \widetilde{O}\left(\sqrt{\frac{H^2d + \lambda C_\Theta^2}{K}} + \sqrt[4]{\frac{H^3 d\epsilon_\mathcal{F}}{K}}\right) + O(\sqrt{C_{\text{eff}}H^3\epsilon_\mathcal{F}} + H\epsilon_\mathcal{F})$

If the model capacity is insufficient, 3.1 will induce extra error due to the large $\epsilon_\mathcal{F}$. If $\epsilon_\mathcal{F} \to 0$, the parametric rate $\frac{1}{\sqrt{K}}$ can be recovered and similar results are derived with general function approximation (GFA) (Chen and Jiang, 2019). However, using concentrability coefficient conceals the problem-dependent structure and omits the specific information of differentiable functions in the complexity measure. Owing to this, we switch to the stronger "uniform" coverage 2.3 and analyze the *pessimistic fitted Q-learning* (PFQL, Algorithm 1).

**Motivation of PFQL.** The PFQL algorithm mingles the two celebrated algorithmic choices: Fitted Q-Iteration (FQI) and Pessimism. However, before going into the technical details, we provide some interesting insights that motivate our analysis.

First of all, the square error loss used in FQI (Gordon, 1999; Ernst et al., 2005) naturally couples with the differentiable function class as the resulting optimization objective is more computationally tractable (since *stochastic gradient descent* (SGD) can be readily applied) comparing to other information-theoretical algorithms derived with general function approximation (*e.g.* the *maxmin* objective in Xie et al. (2021a), eqn (3.2)).[4] In particular, FQI with differentiable function approximation resembles the theoretical prototype of neural FQI algorithm (Riedmiller, 2005) and DQN algorithm (Mnih et al., 2015; Fan et al., 2020) when instantiating the model $f$ to be deep neural networks. Furthermore, plenty of practical algorithms leverage fitted-Q subroutines for updating the *critic* step (*e.g.* (Schulman et al., 2017; Haarnoja et al., 2018)) with different differentiable function choices.

In addition, we also incorporate pessimism for the design. Indeed, one of the fundamental challenges in offline RL comes from the *distributional shift*. When such a mismatch occurs, the estimated/optimized Q-function (using batch data $\mathcal{D}$) may witness severe overestimation error due to the extrapolation of model $f$ (Levine et al., 2020). Pessimism is the scheme to mitigate the error / overestimation bias via penalizing the Q-functions at state-action locations that have high uncertainties (as opposed to the *optimism* used in the online case), and has been widely adopted (*e.g.* (Buckman et al., 2020; Kidambi et al., 2020; Jin et al., 2021b)).

---

[4] We mention Xie et al. (2021a) has a nice practical version PSPI, but the convergence is slower (the rate $O(n^{-\frac{1}{3}})$).

**Algorithm 1 description.** Inside the backward iteration of PFQL, Fitted Q-update is performed to optimize the parameter (Line 4). $\widehat{\theta}_h$ is the root of the first-order stationarity equation $\sum_{k=1}^{K} \left( f(\theta, \phi_{h,k}) - r_{h,k} - \widehat{V}_{h+1}(s_{h+1}^k) \right) \cdot \nabla_\theta^\top f(\theta, \phi_{h,k}) + \lambda\theta = 0$ and $\Sigma_h$ is the Gram matrix with respect to $\nabla_\theta f|_{\theta=\widehat{\theta}_h}$. Note for any $s, a \in \mathcal{S} \times \mathcal{A}$, $m(s,a) := (\nabla_\theta f(\widehat{\theta}_h, \phi(s,a))^\top \Sigma_h^{-1} \nabla_\theta f(\widehat{\theta}_h, \phi(s,a)))^{-1}$ measures the effective sample size that explored $s, a$ along the gradient direction $\nabla_\theta f|_{\theta=\widehat{\theta}_h}$, and $\beta/\sqrt{m(s,a)}$ is the estimated uncertainty at $(s,a)$. However, the quantity $m(s,a)$ depends on $\widehat{\theta}_h$, and $\widehat{\theta}_h$ needs to be close to the true $\theta_h^\star$ (i.e. $\widehat{Q}_h \approx f(\widehat{\theta}_h, \phi)$ needs to be close to $Q_h^\star$) for the uncertainty estimation $\Gamma_h$ to be valid, since putting a random $\theta$ into $m(s,a)$ can cause an arbitrary $\Gamma_h$ that is useless (or might even deteriorate the algorithm). Such an "implicit" constraint over $\widehat{\theta}_h$ imposes the extra difficulty for the theoretical analysis due to that general differentiable functions encode nonlinear structures. Besides, the choice of $\beta$ is set to be $\widetilde{O}(dH)$ in Theorem 3.2 and the extra term $\widetilde{O}(\frac{1}{K})$ in $\Gamma_h$ is for theoretical reason only.

---

**Algorithm 1** Pessimistic Fitted Q-Learning (PFQL)

1: **Input:** Offline Dataset $\mathcal{D} = \left\{ \left( s_h^k, a_h^k, r_h^k, s_{h+1}^k \right) \right\}_{k,h=1}^{K,H}$. Require $\beta$. Denote $\phi_{h,k} := \phi(s_h^k, a_h^k)$.
2: **Initialization:** Set $\widehat{V}_{H+1}(\cdot) \leftarrow 0$ and $\lambda > 0$.
3: **for** $h = H, H-1, \ldots, 1$ **do**
4:     Set $\widehat{\theta}_h \leftarrow \arg\min_{\theta \in \Theta} \left\{ \sum_{k=1}^{K} \left[ f\left(\theta, \phi_{h,k}\right) - r_{h,k} - \widehat{V}_{h+1}(s_{h+1}^k) \right]^2 + \lambda \cdot \|\theta\|_2^2 \right\}$
5:     Set $\Sigma_h \leftarrow \sum_{k=1}^{K} \nabla_\theta f(\widehat{\theta}_h, \phi_{h,k}) \nabla_\theta^\top f(\widehat{\theta}_h, \phi_{h,k}) + \lambda I_d$.
6:     Set $\Gamma_h(\cdot, \cdot) \leftarrow \beta \sqrt{\nabla_\theta f(\widehat{\theta}_h, \phi(\cdot, \cdot))^\top \Sigma_h^{-1} \nabla_\theta f(\widehat{\theta}_h, \phi(\cdot, \cdot))} \left( +\widetilde{O}(\frac{1}{K}) \right)$
7:     Set $\bar{Q}_h(\cdot, \cdot) \leftarrow f(\widehat{\theta}_h, \phi(\cdot, \cdot)) - \Gamma_h(\cdot, \cdot)$
8:     Set $\widehat{Q}_h(\cdot, \cdot) \leftarrow \min \left\{ \bar{Q}_h(\cdot, \cdot), H - h + 1 \right\}^+$
9:     Set $\widehat{\pi}_h(\cdot \mid \cdot) \leftarrow \arg\max_{\pi_h} \left\langle \widehat{Q}_h(\cdot, \cdot), \pi_h(\cdot \mid \cdot) \right\rangle_{\mathcal{A}}$, $\widehat{V}_h(\cdot) \leftarrow \max_{\pi_h} \left\langle \widehat{Q}_h(\cdot, \cdot), \pi_h(\cdot \mid \cdot) \right\rangle_{\mathcal{A}}$
10: **end for**
11: **Output:** $\{\widehat{\pi}_h\}_{h=1}^H$.

---

**Model-Based vs. Model-Free.** PFQL can be viewed as the strict generalization over the previous value iteration based algorithms, *e.g.* PEVI algorithm (Jin et al. (2021b), linear MDPs) and the VPVI algorithm (Yin and Wang (2021), tabular MDPs). On one hand, *approximate value iteration* (AVI) algorithms (Munos, 2005) are usually model-based algorithms (for instance the tabular algorithm VPVI uses empirical model $\widehat{P}$ for planning). On the other hand, FQI has the form of batch Q-learning update (*i.e.* Q-learning is a special case with batch size equals to one), therefore is more of model-free flavor. Since FQI is a concrete instantiation of the abstract AVI procedure (Munos, 2007), PFQL draws a unified view of model-based and model-free learning.

Now we are ready to state our main result for PFQL and the full proof can be found in Appendix D,E,F.

**Theorem 3.2.** *Let $\beta = 8dH\iota$ and choose $0 < \lambda \leq 1/2C_\Theta^2$ in Algorithm 1. Suppose Assumption 2.1,2.3 with $\epsilon_\mathcal{F} = 0$.[5] Then if $K \geq \max \left\{ 512\frac{\kappa_1^4}{\kappa^2} \left( \log(\frac{2Hd}{\delta}) + d\log(1 + \frac{4\kappa_1^3\kappa_2 C_\Theta K^3}{\lambda^2}) \right), \frac{4\lambda}{\kappa} \right\}$, with probability $1 - \delta$, for all policy $\pi$ simultaneously, the output of PFQL guarantees*

$$v^\pi - v^{\widehat{\pi}} \leq \sum_{h=1}^{H} 8dH \cdot \mathbb{E}_\pi \left[ \sqrt{\nabla_\theta^\top f(\widehat{\theta}_h, \phi(s_h, a_h)) \Sigma_h^{-1} \nabla_\theta f(\widehat{\theta}_h, \phi(s_h, a_h))} \right] \cdot \iota + \widetilde{O}(\frac{C_{\text{hot}}}{K}),$$

*where $\iota$ is a Polylog term and the expectation of $\pi$ is taken over $s_h, a_h$. In particular, if further $K \geq \max\{\widetilde{O}(\frac{(\kappa_1^2+\lambda)^2\kappa_2^2\kappa_1^4 H^4 d^2}{\kappa^6}), \frac{128\kappa_1^4\log(2d/\delta)}{\kappa^2}\}$ we have*

$$0 \leq v^{\pi^\star} - v^{\widehat{\pi}} \leq \sum_{h=1}^{H} 16dH \cdot \mathbb{E}_{\pi^\star} \left[ \sqrt{\nabla_\theta^\top f(\theta_h^\star, \phi(s_h, a_h)) \Sigma_h^{\star-1} \nabla_\theta f(\theta_h^\star, \phi(s_h, a_h))} \right] \cdot \iota + \widetilde{O}(\frac{C_{\text{hot}}'}{K}).$$

---

[5]Here we assume model capacity is sufficient to make the presentation concise. If $\epsilon_\mathcal{F} > 0$, the complexity bound will include the term $\epsilon_\mathcal{F}$. We include more discussion in Appendix H.

*Here $\Sigma_h^\star = \sum_{k=1}^K \nabla_\theta f(\theta_h^\star, \phi(s_h^k, a_h^k)) \nabla_\theta^\top f(\theta_h^\star, \phi(s_h^k, a_h^k)) + \lambda I_d$ and the definition of higher order parameter $C_{\text{hot}}$, $C'_{\text{hot}}$ can be found in List A.*

**Corollary 3.3** (Offline Generalized Linear Models (GLM)). *Consider the GLM function class defined in 2.5. Suppose $\beta, \lambda, K$ are defined the same as Theorem 3.2. $\epsilon_{\mathcal{F}} = 0$. Then with probability $1 - \delta$, for all policy $\pi$ simultaneously, PFQL guarantees*

$$v^\pi - v^{\widehat{\pi}} \leq \sum_{h=1}^H 8dH \cdot \mathbb{E}_\pi \left[ \sqrt{f'(\langle \widehat{\theta}_h, \phi(s_h, a_h) \rangle)^2 \cdot \phi^\top(s_h, a_h) \Sigma_h^{-1} \phi(s_h, a_h)} \right] \cdot \iota + \widetilde{O}(\frac{C_{\text{hot}}}{K}).$$

**PFQL is provably efficient.** Theorem 3.2 verifies PFQL is statistically efficient. In particular, by Lemma L.5 we have $\|\nabla_\theta f(\theta_h^\star, \phi)\|_{\Sigma_h^{-1}} \lesssim \frac{2\kappa_1}{\sqrt{\kappa K}}$, resulting the main term to be bounded by $\frac{32dH^2\kappa_1}{\sqrt{\kappa K}}$ that recovers the standard statistical learning convergence rate $\frac{1}{\sqrt{K}}$.

**Comparing to Jin et al. (2021b).** Theorem 3.2 strictly subsumes the linear MDP learning bound in Jin et al. (2021b). In fact, in the linear case $\nabla_\theta f(\theta, \phi) = \nabla_\theta \langle \theta, \phi \rangle = \phi$ and 3.2 reduces to $O(dH \sum_{h=1}^H \mathbb{E}_{\pi^\star}[\sqrt{\phi(s_h, a_h)^\top (\Sigma_h^{\text{linear}})^{-1} \phi(s_h, a_h)}])$.

**Instance-dependent learning.** Previous studies for offline RL with general function approximation (GFA) (Chen and Jiang, 2019; Xie and Jiang, 2020b) are more of worst-case flavors as they usually rely on the *concentrability* coefficient $C$. The resulting learning bounds are expressed in the form[6] $O(V_{\max}^2 \sqrt{\frac{C}{n}})$ that is unable to depict the behavior of individual instances. In contrast, the guarantee with differentiable function approximation is more adaptive due to the instance-dependent structure $\sum_{h=1}^H \mathbb{E}_{\pi^\star} \left[ \sqrt{\nabla_\theta^\top f(\theta_h^\star, \phi) \Sigma_h^{\star-1} \nabla_\theta f(\theta_h^\star, \phi)} \right]$. This Fisher-Information style quantity characterizes the learning hardness of separate problems explicitly as for different MDP instances $M_1, M_2$, the coupled $\theta_{h,M_1}^\star, \theta_{h,M_2}^\star$ will generate different performances via the measure $\sum_{h=1}^H \mathbb{E}_{\pi^\star} \left[ \sqrt{\nabla_\theta^\top f(\theta_{h,M_i}^\star, \phi) \Sigma_h^{\star-1} \nabla_\theta f(\theta_{h,M_i}^\star, \phi)} \right]$ $(i = 1, 2)$. Standard worst-case bounds (*e.g.* from GFA approximation) cannot explicitly differentiate between problem instances.

**Feature representation vs. Parameters.** One interesting observation from Theorem 3.2 is that the learning complexity does not depend on the feature representation dimension $m$ but only on parameter dimension $d$ as long as function class $\mathcal{F}$ satisfies differentiability definition 1.1 (not even in the higher order term). This seems to suggest, when changing the model $f$ with more complex representations, the learning hardness will not grow as long as the number of parameters need to be learned does not increase. Note in the linear MDP analysis this phenomenon is not captured since the two dimensions are coupled ($d = m$). Therefore, this heuristic might help people rethink about what is the more essential element (feature representation vs. parameter space) in the *representation learning RL* regime (*e.g.* low rank MDPs (Uehara et al., 2022)). We leave the concrete understanding the connection between features and parameters as the future work.

**Technical challenges with differentiable function approximation (DFA).** Informally, one key step for the analysis is to bound $|f(\widehat{\theta}_h, \phi) - f(\theta_h^\star, \phi)|$. This can be estimated by the first order approximation $\nabla f(\widehat{\theta}_h, \phi)^\top \cdot (\widehat{\theta}_h - \theta_h^\star)$. However, different from the least-square value iteration (LSVI) objective (Jin et al., 2020b; 2021b), the *fitted Q-update* (Line 4, Algorithm 1) no longer admits a closed-form solution for $\widehat{\theta}_h$. Instead, we can only leverage $\widehat{\theta}_h$ is a stationary point of $Z_h(\theta) := \sum_{k=1}^K \left[ f(\theta, \phi_{h,k}) - r_{h,k} - \widehat{V}_{h+1}(s_{h+1}^k) \right] \nabla f(\theta, \phi_{h,k}) + \lambda \cdot \theta$ (since $Z_h(\widehat{\theta}_h) = 0$). To measure the difference $\widehat{\theta}_h - \theta_h^\star$, for any $\theta \in \Theta$, we do the *Vector Taylor expansion* $Z_h(\theta) - Z_h(\widehat{\theta}_h) = \Sigma_h^s(\theta - \widehat{\theta}_h) + R_K(\theta)$ (where $R_K(\theta)$ is the higher-order residuals) at the point $\widehat{\theta}_h$ with

$$
\begin{aligned}
\Sigma_h^s &:= \left. \frac{\partial}{\partial \theta} Z_h(\theta) \right|_{\theta = \widehat{\theta}_h} = \frac{\partial}{\partial \theta} \left( \sum_{k=1}^K \left[ f(\theta, \phi_{h,k}) - r_{h,k} - \widehat{V}_{h+1}(s_{h+1}^k) \right] \nabla f(\theta, \phi_{h,k}) + \lambda \cdot \theta \right)_{\theta = \widehat{\theta}_h} \\
&= \underbrace{\sum_{k=1}^K \left( f(\widehat{\theta}_h, \phi_{h,k}) - r_{h,k} - \widehat{V}_{h+1}(s_{h+1}^k) \right) \cdot \nabla_{\theta\theta}^2 f(\widehat{\theta}_h, \phi_{h,k})}_{:=\Delta_{\Sigma_h^s}} + \underbrace{\sum_{k=1}^K \nabla_\theta f(\widehat{\theta}_h, \phi_{h,k}) \nabla_\theta^\top f(\widehat{\theta}_h, \phi_{h,k}) + \lambda I_d}_{:=\Sigma_h} .
\end{aligned}
\tag{1}
$$

---

[6] Here $n$ is the number of samples used in the infinite horizon discounted setting and is similar to $K$ in the episodic setting.

The perturbation term $\Delta_{\Sigma_h^s}$ encodes one key challenge for solving $\widehat{\theta}_h - \theta_h^\star$ since it breaks the positive definiteness of $\Sigma_h^s$, and, as a result, we cannot invert the $\Sigma_h^s$ in the Taylor expansion of $Z_h$. This is due to DFA (Definition 1.1) is a rich class that incorporates *nonlinear* curvatures. In the linear function approximation regime, this hurdle will not show up since $\nabla_{\theta\theta}^2 f \equiv 0$ and $\Delta_{\Sigma_h^s}$ is always invertible as long as $\lambda > 0$. Moreover, for the *off-policy evaluation* (OPE) task, one can overcome this issue by expanding over the population counterpart of $Z_h$ at underlying true parameter of the given behavior target policy (Zhang et al., 2022a).[7] However, for the policy learning task, we cannot use either population quantity or the true parameter $\theta_h^\star$ since we need a computable/data-based pessimism $\Gamma_h$ to make the algorithm practical.

## 3.1 SKETCH OF THE PFQL ANALYSIS

Due to the space constraint, here we only overview the key components of the analysis. To begin with, by following the result of general MDP in Jin et al. (2021b), the suboptimality gap can be bounded by (Appendix D) $\sum_{h=1}^H 2\mathbb{E}_\pi\left[\Gamma_h(s_h, a_h)\right]$ if $|(\mathcal{P}_h \widehat{V}_{h+1} - f(\widehat{\theta}_h, \phi))(s, a)| \le \Gamma_h(s, a)$. To deal with $\mathcal{P}_h \widehat{V}_{h+1}$, by Assumption 2.1 we can leverage the *parameter Bellman operator* $\mathbb{T}$ (Definition D.1) so that $\mathcal{P}_h \widehat{V}_{h+1} = f(\theta_{\mathbb{T}\widehat{V}_{h+1}}, \phi)$. Next, we apply the second-order approximation to obtain

$$\mathcal{P}_h\widehat{V}_{h+1} - f(\widehat{\theta}_h, \phi) \approx \nabla f(\widehat{\theta}_h, \phi)^\top (\theta_{\mathbb{T}\widehat{V}_{h+1}} - \widehat{\theta}_h) + \frac{1}{2}(\theta_{\mathbb{T}\widehat{V}_{h+1}} - \widehat{\theta}_h)^\top \nabla_{\theta\theta}^2 f(\widehat{\theta}_h, \phi)(\theta_{\mathbb{T}\widehat{V}_{h+1}} - \widehat{\theta}_h).$$

Later, we use (1) to represent $Z_h(\theta_{\mathbb{T}\widehat{V}_{h+1}}) - Z_h(\widehat{\theta}_h) = \Sigma_h^s(\theta_{\mathbb{T}\widehat{V}_{h+1}} - \widehat{\theta}_h) + R_K(\theta_{\mathbb{T}\widehat{V}_{h+1}}) = \Sigma_h(\theta_{\mathbb{T}\widehat{V}_{h+1}} - \widehat{\theta}_h) + \widetilde{R}_K(\theta_{\mathbb{T}\widehat{V}_{h+1}})$ by denoting $\widetilde{R}_K(\theta_{\mathbb{T}\widehat{V}_{h+1}}) = \Delta_{\Sigma_h^s}(\widehat{\theta}_h - \theta_{\mathbb{T}\widehat{V}_{h+1}}) + R_K(\theta_{\mathbb{T}\widehat{V}_{h+1}})$. Now that $\Sigma_h^{-1}$ is invertible thus provides the estimation (note $Z_h(\widehat{\theta}_h) = 0$)

$$\theta_{\mathbb{T}\widehat{V}_{h+1}} - \widehat{\theta}_h = \Sigma_h^{-1} \cdot Z_h(\theta_{\mathbb{T}\widehat{V}_{h+1}}) - \Sigma_h^{-1}\widetilde{R}_K(\theta_{\mathbb{T}\widehat{V}_{h+1}}).$$

However, to handle the higher order terms, we need the explicit finite sample bound for $\|\theta_{\mathbb{T}\widehat{V}_{h+1}} - \widehat{\theta}_h\|_2$ (or $\|\theta_h^\star - \widehat{\theta}_h\|_2$). In the OPE literature, Zhang et al. (2022a) uses *asymptotic theory* (Prohorovs Theorem) to show the existence of $B(\delta)$ such that $\|\widehat{\theta}_h - \theta_h^\star\| \le \frac{B(\delta)}{\sqrt{K}}$. However, this is insufficient for *finite sample/non-asymptotic* guarantees since the abstraction of $B(\delta)$ might prevent the result from being sample efficient. For example, if $B(\delta)$ has the form $e^H \log(\frac{1}{\delta})$, then $\frac{e^H \log(\frac{1}{\delta})}{\sqrt{K}}$ is an inefficient bound since $K$ needs to be $e^H/\epsilon^2$ large to guarantee $\epsilon$ accuracy.

To address this, we use a novel reduction to *general function approximation* (GFA) learning proposed in Chen and Jiang (2019). We first bound the loss objective $\mathbb{E}_\mu[\ell_h(\widehat{\theta}_h)] - \mathbb{E}_\mu[\ell_h(\theta_{\mathbb{T}\widehat{V}_{h+1}})]$ via a "orthogonal" decomposition and by solving a quadratic equation. The resulting bound can be directly used to further bound $\|\theta_{\mathbb{T}\widehat{V}_{h+1}} - \widehat{\theta}_h\|_2$ for obtaining efficient guarantee $\widetilde{O}(\frac{dH}{\sqrt{\kappa K}})$. During the course, the covering technique is applied to extend the finite function hypothesis to all the differentiable functions in 1.1. See Appendix G and Appendix D,E,F for the complete proofs.

## 4 IMPROVED LEARNING VIA VARIANCE AWARENESS

In addition to knowing the provable efficiency for differentiable function approximation (DFA), it is of great interest to understand what is the statistical limit with DFA, or equivalently, what is the "optimal" sample/statistical complexity can be achieved in DFA (measured by minimaxity criteria)? Towards this goal, we further incorporate *variance awareness* to improve our learning guarantee. Variance awareness is first designed for linear Mixture MDPs (Talebi and Maillard, 2018; Zhou et al., 2021a) to achieve the near-minimax sample complexity and it uses estimated conditional variances $\text{Var}_{P(\cdot|s,a)}(V_{h+1}^\star)$ to reweight each training sample in the LSVI objective.[8] Later, such a technique is leveraged by Min et al. (2021); Yin et al. (2022) to obtained the instance-dependent results. Intuitively, conditional variances $\sigma^2(s, a) := \text{Var}_{P(\cdot|s,a)}(V_{h+1}^\star)$ serves as the uncertainty measure of the sample $(s, a, r, s')$ that comes from the distribution $P(\cdot|s, a)$. If $\sigma^2(s, a)$ is large, then the distribution $P(\cdot|s, a)$ has high variance and we should put less weights in a single sample $(s, a, r, s')$

---

[7]*i.e.* expanding over $Z_h^p(\theta) := \mathbb{E}_{s,a,s'}[(f(\theta, \phi(s, a)) - r - V_{h+1}^\pi(s'))\nabla f(\theta, \phi(s, a))]$, and the corresponding $\Delta_{\Sigma_h^s}$ in $\frac{\partial}{\partial\theta}Z_h(\theta)|_{\theta=\theta_h^\pi}$ is zero by Bellman equation.

[8]We mention Zhang et al. (2021b) uses variance-aware confidence sets in a slightly different way.

rather than weighting all the samples equally. In the differentiable function approximation regime, the update is modified to

$$\widehat{\theta}_h \leftarrow \arg\min_{\theta \in \Theta} \left\{ \sum_{k=1}^{K} \left[ f(\theta, \phi_{h,k}) - r_{h,k} - \widehat{V}_{h+1}(s_{h+1}^k) \right]^2 / \sigma_h^2(s_h^k, a_h^k) + \lambda \cdot \|\theta\|_2^2 \right\}$$

with $\sigma_h^2(\cdot, \cdot)$ estimated by the offline data. Notably, empirical algorithms have also shown uncertainty reweighting can improve the performances for both online RL (Mai et al., 2022) and offline RL (Wu et al., 2021). These motivates our *variance-aware fitted Q-learning* (VAFQL) algorithm 3.

**Theorem 4.1.** *Suppose Assumption 2.1,2.3 with $\epsilon_{\mathcal{F}} = 0$. Let $\beta = 8d\iota$ and choose $0 < \lambda \leq 1/2C_\Theta^2$ in Algorithm 3. Then if $K \geq K_0$ and $\sqrt{d} \geq \widetilde{O}(\zeta)$, with probability $1 - \delta$, for all policy $\pi$ simultaneously, the output of VAFQL guarantees*

$$v^\pi - v^{\widehat{\pi}} \leq \sum_{h=1}^{H} 8d \cdot \mathbb{E}_\pi \left[ \sqrt{\nabla_\theta^\top f(\widehat{\theta}_h, \phi(s_h, a_h)) \Lambda_h^{-1} \nabla_\theta f(\widehat{\theta}_h, \phi(s_h, a_h))} \right] \cdot \iota + \widetilde{O}\left(\frac{\bar{C}_{\text{hot}}}{K}\right),$$

*where $\iota$ is a Polylog term and the expectation of $\pi$ is taken over $s_h, a_h$. In particular, we have $0 \leq v^{\pi^\star} - v^{\widehat{\pi}} \leq 16d \cdot \sum_{h=1}^{H} \mathbb{E}_{\pi^\star} \left[ \sqrt{\nabla_\theta^\top f(\theta_h^\star, \phi(s_h, a_h)) \Lambda_h^{\star-1} \nabla_\theta f(\theta_h^\star, \phi(s_h, a_h))} \right] \cdot \iota + \widetilde{O}\left(\frac{\bar{C}'_{\text{hot}}}{K}\right)$. Here $\Lambda_h^\star = \sum_{k=1}^{K} \nabla_\theta f(\theta_h^\star, \phi_{h,k}) \nabla_\theta^\top f(\theta_h^\star, \phi_{h,k}) / \sigma_h^\star(s_h^k, a_h^k)^2 + \lambda I_d$ and the $\sigma_h^\star(\cdot, \cdot)^2 := \max\{1, \text{Var}_{P_h} V_{h+1}^\star(\cdot, \cdot)\}$. The definition of $K_0, \bar{C}_{\text{hot}}, \bar{C}'_{\text{hot}}, \zeta$ can be found in List A.*

In particular, to bound the error for $\boldsymbol{u}_h, \boldsymbol{v}_h$ and $\widehat{\sigma}_h^2$, we need to define an operator $\mathbb{J}$ that is similar to the *parameter Bellman operator* D.1. The Full proof of Theorem 4.1 can be found in Appendix J. Comparing to Theorem 3.2, VAFQL enjoys a net improvement of the horizon dependence since $\text{Var}_P(V_h^\star) \leq H^2$. Moreover, VAFQL provides better instance-dependent characterizations as the main term is fully depicted by the system quantities except the feature dimension $d$. For instance, when the system is fully deterministic (transition $P_h$'s are deterministic), $\sigma_h^\star \approx \text{Var}_{P_h} V_{h+1}^\star(\cdot, \cdot) \equiv 0$ (if ignore the truncation) and $\Lambda^{\star-1} \to 0$. This yields a faster convergence with rate $O(\frac{1}{K})$. Lastly, when reduced to linear MDPs, 4.1 recovers the results of Yin et al. (2022) except an extra $\sqrt{d}$.

**On the statistical limits.** To complement the study, we incorporate a minimax lower bound via a reduction to Zanette et al. (2021). The following theorem reveals we cannot improve over Theorem 4.1 by more than a factor of $\sqrt{d}$ in the most general cases. The full discussion is in K.

**Theorem 4.2** (Minimax lower bound). *Specifying the model to have linear representation $f = \langle \theta, \phi \rangle$. There exist a pair of universal constants $c, c' > 0$ such that given dimension $d$, horizon $H$ and sample size $K > c'd^3$, one can always find a family of MDP instances such that for any algorithm $\widehat{\pi}$ (where $\Lambda_h^{\star,p} = \mathbb{E} \left[ \sum_{k=1}^{K} \frac{\nabla_\theta f(\theta_h^\star, \phi(s_h^k, a_h^k)) \cdot \nabla_\theta f(\theta_h^\star, \phi(s_h^k, a_h^k))^\top}{\text{Var}_h(V_{h+1}^\star)(s_h^k, a_h^k)} \right]$)*

$$\inf_{\widehat{\pi}} \sup_{M \in \mathcal{M}} \mathbb{E}_M \left[ v^\star - v^{\widehat{\pi}} \right] \geq c \sqrt{d} \cdot \sum_{h=1}^{H} \mathbb{E}_{\pi^\star} \left[ \sqrt{\nabla_\theta^\top f(\theta_h^\star, \phi(\cdot, \cdot))(\Lambda_h^{\star,p})^{-1} \nabla_\theta f(\theta_h^\star, \phi(\cdot, \cdot))} \right]. \quad (2)$$

## 5 CONCLUSION, LIMITATION AND FUTURE DIRECTIONS

In this work, we study offline RL with differentiable function approximation and show the sample efficient learning. We further improve the horizon dependence via a variance aware variant. However, the dependence of the parameter space still scales with $d$ (whereas for the linear case it is $\sqrt{d}$), and this is due to applying covering argument for the rich class of differentiable functions. For large deep models, the parameter dimension is huge, therefore it would be interesting to know if certain algorithms can further improve the parameter dependence, or whether this $d$ is essential.

Also, how to relax uniform coverage 2.3 is unknown under the current analysis. In addition, understanding the connections between the differentiable function approximation and overparameterized neural networks approximation Nguyen-Tang and Arora (2023); Xu and Liang (2022) is important. We leave these open problems as future work. Lastly, the differentiable function approximation setting provides a general framework that is not confined to offline RL. Understanding the sample complexity behaviors of online reinforcement learning (Jin et al., 2020b; Wang et al., 2021b), reward-free learning (Jin et al., 2020a; Wang et al., 2020) and representation learning (Uehara et al., 2022) might provide new and unified views over the existing studies.

ACKNOWLEDGMENTS

The authors would like to thank anonymous reviewers for their helpful suggestions. Ming Yin would like to thank Chi Jin for the helpful suggestions regarding the assumption for differentiable function class and Andrea Zanette, Xuezhou Zhang for the friendly discussion. Mengdi Wang gratefully acknowledges funding from Office of Naval Research (ONR) N00014-21-1-2288, Air Force Office of Scientific Research (AFOSR) FA9550-19-1-0203, and NSF 19-589, CMMI-1653435. Ming Yin and Yu-Xiang Wang are gratefully supported by National Science Foundation (NSF) Awards #2007117 and #2003257.

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

# Appendix

## A NOTATION LIST

| | |
|---|---|
| $\Sigma_h^p(\theta)$ | $\mathbb{E}_{\mu,h}\left[\nabla f(\theta, \phi(s,a)) \cdot \nabla f(\theta, \phi(s,a))^\top\right]$ |
| $\kappa$ | $\min_{h,\theta} \lambda_{\min}(\Sigma_h^p(\theta))$ |
| $\sigma_V^2(s,a)$ | $\max\{1, \mathrm{Var}_{P_h}(V)(s,a)\}$ for any $V$ |
| $\delta$ | Failure probability |
| $K_0$ | $\max\left\{512\frac{\kappa_1^4}{\kappa^2}\left(\log(\frac{2Hd}{\delta}) + d\log(1 + \frac{4\kappa_1^3\kappa_2 C_\Theta K^3}{\lambda^2})\right), \frac{4\lambda}{\kappa}\right\}$ |
| $\zeta$ | $2\max_{s'\sim P(\cdot|s,a), h\in[H]} \frac{(\mathcal{P}_h V_{h+1}^\star)(s,a) - r - V_{h+1}^\star(s')}{\sigma_h^\star(s,a)}$ |
| $C_{\mathrm{hot}} = \bar{C}_{\mathrm{hot}}$ | $\frac{\kappa_1 H}{\sqrt{\kappa}} + \frac{\kappa_1^2 H^3 d^2}{\kappa} + \sqrt{\frac{d^3 H^4 \kappa_2^2 \kappa_1^2}{\kappa^3}} + \kappa_2 \max(\frac{\kappa_1}{\kappa}, \frac{1}{\sqrt{\kappa}})d^2 H^3 + \frac{d^2 H^4 \kappa_3 + \lambda\kappa_1 C_\Theta}{\kappa} + \frac{H^3 \kappa_2 d^2}{\kappa}$ |
| $C_{\mathrm{hot}}' = \bar{C}_{\mathrm{hot}}'$ | $C_{\mathrm{hot}} + \frac{\kappa_1\kappa_2 H^4 d^2}{\kappa^{3/2}}$ |

## B FURTHER ILLUSTRATION THAT GENERALIZED LINEAR MODEL EXAMPLE SATISFIES 2.3

Recall the definition in 2.5, then:

For $(\star\star)$,

$$\mathbb{E}_{\mu,h}\left[\nabla f(\theta, \phi(s,a)) \cdot \nabla f(\theta, \phi(s,a))^\top\right] = \mathbb{E}_{\mu,h}\left[f'(\langle\theta, \phi(s,a)\rangle)^2 \phi(\cdot,\cdot) \cdot \phi(\cdot,\cdot)^\top\right]$$
$$\succ \kappa^2 \mathbb{E}_{\mu,h}\left[\phi(\cdot,\cdot) \cdot \phi(\cdot,\cdot)^\top\right] \succ \kappa^3 I, \quad \forall \theta \in \Theta$$

For $(\star)$, by Taylor's Theorem,

$$\mathbb{E}_{\mu,h}\left[(f(\theta_1, \phi(\cdot,\cdot)) - f(\theta_2, \phi(\cdot,\cdot)))^2\right] = \mathbb{E}_{\mu,h}[f'(\theta_{s,a}, \phi(\cdot,\cdot))^2 (\theta_1 - \theta_2)^\top \phi(\cdot,\cdot)\phi(\cdot,\cdot)^\top (\theta_1 - \theta_2)]$$
$$\geq \kappa^2 \mathbb{E}_{\mu,h}[(\theta_1 - \theta_2)^\top \phi(\cdot,\cdot)\phi(\cdot,\cdot)^\top (\theta_1 - \theta_2)] = \kappa^2 (\theta_1 - \theta_2)^\top \mathbb{E}_{\mu,h}[\phi(\cdot,\cdot)\phi(\cdot,\cdot)^\top](\theta_1 - \theta_2) \geq \kappa^3 \|\theta_1 - \theta_2\|_2^2$$

and choose $\kappa^3$ as $\kappa$ in 2.3.

## C ON THE COMPUTATIONAL COMPLEXITY

For storage of Pessimistic Fitted Q-learning, at each time step $h \in [H]$ in Algorithm 1, we need to store $\widehat{\theta}_h, \Sigma_h$ and $\nabla f(\widehat{\theta}_h, \phi_{h,k})$. Therefore, the total space complexity is $O(dH + d^2 H + dKH)$. For computation, assuming $\widehat{\theta}_h$ is solved via SGD and let $M$ denote the number of gradient steps, then the complexity is dominated by computing $\widehat{\theta}_h, \Sigma_h$ and $\Sigma_h^{-1}$, which results in $O(MH + KdH + d^3 H)$ complexity (where $H$ comes from $h = H, \ldots, 1$).

The space complexity and computational complexity for VAFQL has the same order as PFQL except that the constant factors are larger.

## D SOME BASIC CONSTRUCTIONS

First of all, Recall in the first-order condition, we have

$$\nabla_\theta \left\{\sum_{k=1}^K \left[f(\theta, \phi_{h,k}) - r_{h,k} - \widehat{V}_{h+1}(s_{h+1}^k)\right]^2 + \lambda \cdot \|\theta\|_2^2\right\}\bigg|_{\theta=\widehat{\theta}_h} = 0, \quad \forall h \in [H].$$

Therefore, if we define the quantity $Z_h(\cdot, \cdot) \in \mathbb{R}^d$ as

$$Z_h(\theta|V) = \sum_{k=1}^{K} \left[ f(\theta, \phi_{h,k}) - r_{h,k} - V(s_{h+1}^k) \right] \nabla f(\theta, \phi_{h,k}) + \lambda \cdot \theta, \quad \forall \theta \in \Theta, \|V\|_2 \le H,$$

then we have (recall $\widehat{\theta}_h \in \text{Int}(\Theta)$)

$$Z_h(\widehat{\theta}_h | \widehat{V}_{h+1}) = 0.$$

In addition, according to Bellman completeness Assumption 2.1, for any bounded $V(\cdot) \in \mathbb{R}^{\mathcal{S}}$ with $\|V\|_\infty \le H$, $\inf_{f \in \mathcal{F}} \|f - \mathcal{P}_h(V)\|_\infty \le \epsilon_{\mathcal{F}}$, $\forall h$ (recall $\mathcal{P}_h(V) = r_h + \int_{\mathcal{S}} V dP_h$). Therefore, we can define the *parameter Bellman operator* $\mathbb{T}$ as follows.

**Definition D.1** (parameter Bellman operator)**.** *By the Bellman completeness Assumption 2.1, for any* $\|V\|_\infty \le H$, *we can define the* parameter Bellman operator $\mathbb{T} : V \to \theta_{\mathbb{T}V} \in \Theta$ *such that*

$$\theta_{\mathbb{T}V} = \arg\min_{\theta \in \Theta} \|f(\theta, \phi) - \mathcal{P}_h(V)\|_\infty$$

*Denote* $\delta_V := f(\theta_{\mathbb{T}V}, \phi) - \mathcal{P}_h(V)$, *then we have* $\|f(\theta_{\mathbb{T}V}, \phi) - \mathcal{P}_h(V)\|_\infty = \|\delta_V\|_\infty \le \epsilon_{\mathcal{F}}$. *In particular, by realizability Assumption 2.1 it holds* $\theta_{\mathbb{T}V_{h+1}^\star} = \theta_h^\star$ *and this is due to* $f(\theta_{\mathbb{T}V_{h+1}^\star}, \phi) = \mathcal{P}_h(V_{h+1}^\star) = V_h^\star = f(\theta_h^\star, \phi)$.[9]

### D.1 SUBOPTIMALITY DECOMPOSITION

Denote $\iota_h(s, a) := \mathcal{P}_h \widehat{V}_{h+1}(s, a) - \widehat{Q}_h(s, a)$, by Jin et al. (2021b) we have the following decomposition.

**Lemma D.2** (Lemma 3.1 of Jin et al. (2021b))**.** *Let* $\widehat{\pi} = \{\widehat{\pi}_h\}_{h=1}^H$ *a policy and* $\widehat{Q}_h$ *be any estimates with* $\widehat{V}_h = \langle \widehat{Q}_h(s, \cdot), \widehat{\pi}_h(\cdot \mid s) \rangle_{\mathcal{A}}$. *Then for any policy* $\pi$, *we have*

$$v^\pi - v^{\widehat{\pi}} = -\sum_{h=1}^{H} E_{\widehat{\pi}}[\iota_h(s_h, a_h)] + \sum_{h=1}^{H} E_\pi[\iota_h(s_h, a_h)] + \sum_{h=1}^{H} E_\pi[\langle \widehat{Q}_h(s_h, \cdot), \pi_h(\cdot \mid s_h) - \widehat{\pi}_h(\cdot \mid s_h) \rangle_{\mathcal{A}}].$$

*In particular, if we choose* $\widehat{\pi}_h(\cdot|s) := \arg\max_\pi \langle \widehat{Q}_h(s, \cdot), \pi(\cdot \mid s) \rangle_{\mathcal{A}}$, *then*

$$v^\pi - v^{\widehat{\pi}} = -\sum_{h=1}^{H} E_{\widehat{\pi}}[\iota_h(s_h, a_h)] + \sum_{h=1}^{H} E_\pi[\iota_h(s_h, a_h)].$$

**Lemma D.3.** *Let* $\widehat{\mathcal{P}}_h$ *be the general estimated Bellman operator. Suppose with probability* $1 - \delta$, *it holds for all* $h, s, a \in [H] \times \mathcal{S} \times A$ *that* $|(\mathcal{P}_h \widehat{V}_{h+1} - \widehat{\mathcal{P}}_h \widehat{V}_{h+1})(s, a)| \le \Gamma_h(s, a)$, *then it implies* $\forall s, a, h \in \mathcal{S} \times \mathcal{A} \times [H]$, $0 \le \zeta_h(s, a) \le 2\Gamma_h(s, a)$. *Furthermore, it holds for any policy* $\pi$ *simultaneously, with probability* $1 - \delta$,

$$V_1^\pi(s) - V_1^{\widehat{\pi}}(s) \le \sum_{h=1}^{H} 2 \cdot \mathbb{E}_\pi \left[ \Gamma_h(s_h, a_h) \mid s_1 = s \right].$$

*Proof of Lemma D.3.* This is a generic result that holds true for the general MDPs and was first raised by Theorem 4.2 of Jin et al. (2021b). Later, it is summarized in Lemma C.1 of Yin et al. (2022). $\square$

With Lemma D.3, we need to bound the term $|\mathcal{P}_h \widehat{V}_{h+1}(s, a) - \widehat{\mathcal{P}}_h \widehat{V}_{h+1}(s, a)|$.

---

[9]Here without loss of generality we assume $Q_h^\star$ can be uniquely identified, *i.e.* there is a unique $\theta^\star$ such that $f(\theta_h^\star, \phi) = Q_h^\star$.

# E   ANALYZING $|\mathcal{P}_h\widehat{V}_{h+1}(s,a) - \widehat{\mathcal{P}}_h\widehat{V}_{h+1}(s,a)|$ FOR PFQL.

Throughout this section, we suppose $\epsilon_{\mathcal{F}} = 0$, *i.e.* $f(\theta_{\mathbb{T}V}, \phi) = \mathcal{P}_h(V)$. According to the regression oracle (Line 4 of Algorithm 1), the estimated Bellman operator $\widehat{\mathcal{P}}_h$ maps $\widehat{V}_{h+1}$ to $\widehat{\theta}_h$, *i.e.* $\widehat{\mathcal{P}}_h\widehat{V}_{h+1} = f(\widehat{\theta}_h, \phi)$. Therefore (recall Definition D.1)

$$\mathcal{P}_h\widehat{V}_{h+1}(s,a) - \widehat{\mathcal{P}}_h\widehat{V}_{h+1}(s,a) = \mathcal{P}_h\widehat{V}_{h+1}(s,a) - f(\widehat{\theta}_h, \phi(s,a))$$
$$= f(\theta_{\mathbb{T}\widehat{V}_{h+1}}, \phi(s,a)) - f(\widehat{\theta}_h, \phi(s,a)) \tag{3}$$
$$= \nabla f(\widehat{\theta}_h, \phi(s,a))\left(\theta_{\mathbb{T}\widehat{V}_{h+1}} - \widehat{\theta}_h\right) + \mathrm{Hot}_{h,1},$$

where we apply the first-order Taylor expansion for the differentiable function $f$ at point $\widehat{\theta}_h$ and $\mathrm{Hot}_{h,1}$ is a higher-order term. Indeed, the following Lemma E.1 bounds the $\mathrm{Hot}_{h,1}$ term with $\widetilde{O}(\frac{1}{K})$.

**Lemma E.1.** *Recall the definition (from the above decomposition)* $\mathrm{Hot}_{h,1} := f(\theta_{\mathbb{T}\widehat{V}_{h+1}}, \phi(s,a)) - f(\widehat{\theta}_h, \phi(s,a)) - \nabla f(\widehat{\theta}_h, \phi(s,a))\left(\theta_{\mathbb{T}\widehat{V}_{h+1}} - \widehat{\theta}_h\right)$, *then with probability* $1 - \delta$,

$$|\mathrm{Hot}_{h,1}| \le \frac{18H^2\kappa_2(\log(H/\delta) + C_{d,\log K}) + \kappa_2\lambda C_\Theta^2}{\kappa K}, \quad \forall h \in [H].$$

*Proof of Lemma E.1.* By second-order Taylor's Theorem, there exists a point $\xi$ (lies in the line segment of $\widehat{\theta}_h$ and $\theta_{\mathbb{T}\widehat{V}_{h+1}}$) such that

$$f(\theta_{\mathbb{T}\widehat{V}_{h+1}}, \phi(s,a)) - f(\widehat{\theta}_h, \phi(s,a)) = \nabla f(\widehat{\theta}_h, \phi(s,a))^\top\left(\theta_{\mathbb{T}\widehat{V}_{h+1}} - \widehat{\theta}_h\right) + \frac{1}{2}\left(\theta_{\mathbb{T}\widehat{V}_{h+1}} - \widehat{\theta}_h\right)^\top \nabla_{\theta\theta}^2 f(\xi, \phi(s,a))\left(\theta_{\mathbb{T}\widehat{V}_{h+1}} - \widehat{\theta}_h\right)$$

Therefore, by directly applying Theorem G.2, with probability $1 - \delta$, for all $h \in [H]$,

$$|\mathrm{Hot}_{h,1}| = \frac{1}{2}\left|\left(\theta_{\mathbb{T}\widehat{V}_{h+1}} - \widehat{\theta}_h\right)^\top \nabla_{\theta\theta}^2 f(\xi, \phi(s,a))\left(\theta_{\mathbb{T}\widehat{V}_{h+1}} - \widehat{\theta}_h\right)\right|$$
$$\le \frac{1}{2}\kappa_2 \cdot \left\|\theta_{\mathbb{T}\widehat{V}_{h+1}} - \widehat{\theta}_h\right\|_2^2 \le \frac{18H^2\kappa_2(\log(H/\delta) + C_{d,\log K}) + \kappa_2\lambda C_\Theta^2}{\kappa K}$$

$\square$

## E.1   ANALYZING $\nabla f(\widehat{\theta}_h, \phi(s,a))\left(\theta_{\mathbb{T}\widehat{V}_{h+1}} - \widehat{\theta}_h\right)$ VIA $Z_h$.

From (3) and Lemma E.1, the problem further reduces to bounding $\nabla f(\widehat{\theta}_h, \phi(s,a))\left(\theta_{\mathbb{T}\widehat{V}_{h+1}} - \widehat{\theta}_h\right)$. To begin with, we first provide a characterization of $\theta_{\mathbb{T}\widehat{V}_{h+1}} - \widehat{\theta}_h$. Indeed, by first-order Vector Taylor expansion (Lemma L.1), we have (note $Z_h(\widehat{\theta}_h|\widehat{V}_{h+1}) = 0$) for any $\theta \in \Theta$,

$$Z_h(\theta|\widehat{V}_{h+1}) - Z_h(\widehat{\theta}_h|\widehat{V}_{h+1}) = \Sigma_h^s(\theta - \widehat{\theta}_h) + R_K(\theta), \tag{4}$$

where $R_K(\theta)$ is the higher-order residuals and $\Sigma_h^s := \frac{\partial}{\partial\theta}Z_h(\theta|\widehat{\theta}_{h+1})\big|_{\theta=\widehat{\theta}_h}$ with

$$\Sigma_h^s := \frac{\partial}{\partial\theta}Z_h(\theta|\widehat{V}_{h+1})\bigg|_{\theta=\widehat{\theta}_h} = \frac{\partial}{\partial\theta}\left(\sum_{k=1}^K\left[f(\theta, \phi_{h,k}) - r_{h,k} - \widehat{V}_{h+1}(s_{h+1}^k)\right]\nabla f(\theta, \phi_{h,k}) + \lambda \cdot \theta\right)_{\theta=\widehat{\theta}_h}$$

$$= \underbrace{\sum_{k=1}^K\left\{\left(f(\widehat{\theta}_h, \phi_{h,k}) - r_{h,k} - \widehat{V}_{h+1}(s_{h+1}^k)\right) \cdot \nabla_{\theta\theta}^2 f(\widehat{\theta}_h, \phi_{h,k})\right\}}_{:=\Delta_{\Sigma_h^s}}$$

$$+ \underbrace{\sum_{k=1}^K \nabla_\theta f(\widehat{\theta}_h, \phi_{h,k})\nabla_\theta^\top f(\widehat{\theta}_{h,k}, \phi_{h,k}) + \lambda I_d}_{:=\Sigma_h}, \tag{5}$$

here $\nabla^2 = \nabla \bigotimes \nabla$ denotes outer product of gradients.

Note $\Delta_{\Sigma_h^s}$ is not desirable since it could prevent $\Sigma_h^s$ from being positive-definite (and it could cause $\Sigma_h^s$ to be singular). Therefore, we first deal with $\Delta_{\Sigma_h^s}$ in below.

**Lemma E.2.** *With probability $1 - \delta$, for all $h \in [H]$,*

$$\frac{1}{K} \left\| \Delta_{\Sigma_h^s} \right\|_2 = \left\| \frac{1}{K} \sum_{k=1}^K \left( f(\widehat{\theta}_h, \phi_{h,k}) - r_{h,k} - \widehat{V}_{h+1}(s_{h+1}^k) \right) \cdot \nabla_{\theta\theta}^2 f(\widehat{\theta}_h, \phi_h) \right\|_2$$

$$\leq 9\kappa_2 \max(\frac{\kappa_1}{\sqrt{\kappa}}, 1) \sqrt{\frac{dH^2(\log(2H/\delta) + d\log(1 + 2C_\Theta H \kappa_3 K) + C_{d,\log K})}{K}} + \frac{1}{K}.$$

*Proof of Lemma E.2.* **Step1:** We prove for fixed $\bar{\theta} \in \Theta$, with probability $1 - \delta$, for all $h \in [H]$,

$$\left\| \frac{1}{K} \sum_{k=1}^K \left( f(\widehat{\theta}_h, \phi_{h,k}) - r_{h,k} - \widehat{V}_{h+1}(s_{h+1}^k) \right) \cdot \nabla_{\theta\theta}^2 f(\bar{\theta}, \phi_h) \right\|_2 \leq 9\kappa_2 \max(\frac{\kappa_1}{\sqrt{\kappa}}, 1) \sqrt{\frac{H^2(\log(2H/\delta) + C_{d,\log K})}{K}}.$$

Indeed, we have

$$\left\| \frac{1}{K} \sum_{k=1}^K \left( f(\widehat{\theta}_h, \phi_{h,k}) - r_{h,k} - \widehat{V}_{h+1}(s_{h+1}^k) \right) \cdot \nabla_{\theta\theta}^2 f(\bar{\theta}, \phi_h) \right\|_2$$

$$\leq \left\| \frac{1}{K} \sum_{k=1}^K \left( f(\widehat{\theta}_h, \phi_{h,k}) - f(\theta_{\mathbb{T}\widehat{V}_{h+1}}, \phi_{h,k}) \right) \cdot \nabla_{\theta\theta}^2 f(\bar{\theta}, \phi_h) \right\|_2 \qquad (6)$$

$$+ \left\| \frac{1}{K} \sum_{k=1}^K \left( f(\theta_{\mathbb{T}\widehat{V}_{h+1}}, \phi_{h,k}) - r_{h,k} - \widehat{V}_{h+1}(s_{h+1}^k) \right) \cdot \nabla_{\theta\theta}^2 f(\bar{\theta}, \phi_h) \right\|_2.$$

On one hand, by Theorem G.2 with probability $1 - \delta/2$ for all $h \in [H]$

$$\left\| \frac{1}{K} \sum_{k=1}^K \left( f(\widehat{\theta}_h, \phi_{h,k}) - f(\theta_{\mathbb{T}\widehat{V}_{h+1}}, \phi_{h,k}) \right) \cdot \nabla_{\theta\theta}^2 f(\bar{\theta}, \phi_h) \right\|_2 \leq \kappa_2 \cdot \max_{\theta,s,a} \|\nabla f(\theta, \phi(s,a))\|_2 \left\| \widehat{\theta}_h - \theta_{\mathbb{T}\widehat{V}_{h+1}} \right\|_2$$

$$\leq \kappa_2 \kappa_1 \left\| \widehat{\theta}_h - \theta_{\mathbb{T}\widehat{V}_{h+1}} \right\|_2 \leq \kappa_2 \kappa_1 \left( \sqrt{\frac{36H^2(\log(H/\delta) + C_{d,\log K}) + 2\lambda C_\Theta^2}{\kappa K}} + \sqrt{\frac{b_{d,K,\epsilon_\mathcal{F}}}{\kappa}} + \sqrt{\frac{2H\epsilon_\mathcal{F}}{\kappa}} \right).$$

$$(7)$$

On other hand, recall the definition of $\mathbb{T}$, we have

$$\mathbb{E}\left[ (f(\theta_{\mathbb{T}\widehat{V}_{h+1}}, \phi_{h,k}) - r_{h,k} - \widehat{V}_{h+1}(s_{h+1}^k)) \cdot \nabla_{\theta\theta}^2 f(\bar{\theta}, \phi_{h,k}) \Big| s_h^k, a_h^k \right]$$

$$= \mathbb{E}\left[ f(\theta_{\mathbb{T}\widehat{V}_{h+1}}, \phi_{h,k}) - r_{h,k} - \widehat{V}_{h+1}(s_{h+1}^k) \Big| s_h^k, a_h^k \right] \cdot \nabla_{\theta\theta}^2 f(\bar{\theta}, \phi_{h,k})$$

$$= \left( (\mathcal{P}_h \widehat{V}_{h+1})(s_h^k, a_h^k) - \mathbb{E}\left[ r_{h,k} + \widehat{V}_{h+1}(s_{h+1}^k) \Big| s_h^k, a_h^k \right] \right) \cdot \nabla_{\theta\theta}^2 f(\bar{\theta}, \phi_{h,k})$$

$$= \left( (\mathcal{P}_h \widehat{V}_{h+1})(s_h^k, a_h^k) - (\mathcal{P}_h \widehat{V}_{h+1}(s_{h+1}^k))(s_h^k, a_h^k) \right) \cdot \nabla_{\theta\theta}^2 f(\bar{\theta}, \phi_{h,k}) = 0.$$

Also, since $\left\| \left( f(\theta_{\mathbb{T}\widehat{V}_{h+1}}, \phi_{h,k}) - r_{h,k} - \widehat{V}_{h+1}(s_{h+1}^k) \right) \cdot \nabla_{\theta\theta}^2 f(\bar{\theta}, \phi_h) \right\|_2 \leq H\kappa_2$, denote $\sigma^2 := K \cdot H^2 \kappa_2^2$, then by Vector Hoeffding's inequality (Lemma L.2),

$$\mathbb{P}\left( \left\| \frac{1}{K} \sum_{k=1}^K \left( f(\theta_{\mathbb{T}\widehat{V}_{h+1}}, \phi_{h,k}) - r_{h,k} - \widehat{V}_{h+1}(s_{h+1}^k) \right) \cdot \nabla_{\theta\theta}^2 f(\bar{\theta}, \phi_h) \right\|_2 \geq t/K \Big| \{s_h^k, a_h^k\}_{k=1}^K \right) \leq d \cdot e^{-t^2/8dKH^2\kappa_2^2} := \delta$$

which is equivalent to

$$\mathbb{P}\left( \left\| \frac{1}{K} \sum_{k=1}^K \left( f(\theta_{\mathbb{T}\widehat{V}_{h+1}}, \phi_{h,k}) - r_{h,k} - \widehat{V}_{h+1}(s_{h+1}^k) \right) \cdot \nabla_{\theta\theta}^2 f(\bar{\theta}, \phi_h) \right\|_2 \leq \sqrt{\frac{8dH^2\kappa_2^2 \log(d/\delta)}{K}} \Big| \{s_h^k, a_h^k\}_{k=1}^K \right) \geq 1 - \delta$$

Define $A = \{\left\|\frac{1}{K}\sum_{k=1}^{K}\left(f(\theta_{\mathbb{T}\widehat{V}_{h+1}},\phi_{h,k}) - r_{h,k} - \widehat{V}_{h+1}(s_{h+1}^k)\right)\cdot\nabla_{\theta\theta}^2 f(\bar{\theta},\phi_h)\right\|_2 \leq \sqrt{\frac{8dH^2\kappa_2^2\log(d/\delta)}{K}}\}$, then by law of total expectation $\mathbb{P}(A) = \mathbb{E}[\mathbf{1}_A] = \mathbb{E}[\mathbb{E}[\mathbf{1}_A|\{s_h^k,a_h^k\}_{k=1}^K]] = \mathbb{E}[\mathbb{P}[A|\{s_h^k,a_h^k\}_{k=1}^K]] \geq \mathbb{E}[1-\delta] = 1-\delta$, i.e. with probability at least $1-\delta/2$ (and a union bound),

$$\left\|\frac{1}{K}\sum_{k=1}^{K}\left(f(\theta_{\mathbb{T}\widehat{V}_{h+1}},\phi_{h,k}) - r_{h,k} - \widehat{V}_{h+1}(s_{h+1}^k)\right)\cdot\nabla_{\theta\theta}^2 f(\bar{\theta},\phi_h)\right\|_2 \leq \sqrt{\frac{8dH^2\kappa_2^2\log(2Hd/\delta)}{K}}, \; \forall h \in [H].$$

Using above and equation 6, equation 7 and a union bound, w.p. $1-\delta$, for all $h \in [H]$,

$$\left\|\frac{1}{K}\sum_{k=1}^{K}\left(f(\widehat{\theta}_h,\phi_{h,k}) - r_{h,k} - \widehat{V}_{h+1}(s_{h+1}^k)\right)\cdot\nabla_{\theta\theta}^2 f(\bar{\theta},\phi_h)\right\|_2 \leq 6\kappa_2\kappa_1\sqrt{\frac{H^2(\log(2H/\delta) + C_{d,\log K})}{\kappa K}} + \sqrt{\frac{8dH^2\kappa_2^2\log(2Hd/\delta)}{K}}$$

$$\leq 9\kappa_2 \max(\frac{\kappa_1}{\sqrt{\kappa}},1)\sqrt{\frac{dH^2(\log(2H/\delta) + C_{d,\log K})}{K}}$$

**Step2:** we finish the proof of the lemma.

Consider the function class $\left\{f(\bar{\theta}) := \left\|\frac{1}{K}\sum_{k=1}^{K}\left(f(\widehat{\theta}_h,\phi_{h,k}) - r_{h,k} - \widehat{V}_{h+1}(s_{h+1}^k)\right)\cdot\nabla_{\theta\theta}^2 f(\bar{\theta},\phi_h)\right\|_2 \Big| \bar{\theta} \in \Theta\right\}$, then by triangular inequality

$$|f(\bar{\theta}_1) - f(\bar{\theta}_2)| \leq \left\|\frac{1}{K}\sum_{k=1}^{K}\left(f(\widehat{\theta}_h,\phi_{h,k}) - r_{h,k} - \widehat{V}_{h+1}(s_{h+1}^k)\right)\cdot\left[\nabla_{\theta\theta}^2 f(\bar{\theta}_1,\phi_h) - \nabla_{\theta\theta}^2 f(\bar{\theta}_2,\phi_h)\right]\right\|_2$$

$$\leq H\cdot\sup_{s,a}\left\|\nabla_{\theta\theta}^2 f(\bar{\theta}_1,\phi_h) - \nabla_{\theta\theta}^2 f(\bar{\theta}_2,\phi_h)\right\|_2 \leq H\kappa_3\left\|\bar{\theta}_1 - \bar{\theta}_2\right\|_2.$$

By Lemma L.8, the covering number $\mathcal{C}$ of the $\epsilon$-net of the above function class satisfies $\log\mathcal{C} \leq d\log(1 + \frac{2C_\Theta H\kappa_3}{\epsilon})$. By choosing $\epsilon = 1/K$, by a union bound over $\mathcal{C}$ cases we obtain for all $h \in [H]$

$$\left\|\frac{1}{K}\sum_{k=1}^{K}\left(f(\widehat{\theta}_h,\phi_{h,k}) - r_{h,k} - \widehat{V}_{h+1}(s_{h+1}^k)\right)\cdot\nabla_{\theta\theta}^2 f(\widehat{\theta}_h,\phi_h)\right\|_2$$

$$\leq 9\kappa_2 \max(\frac{\kappa_1}{\sqrt{\kappa}},1)\sqrt{\frac{dH^2(\log(2H/\delta) + d\log(1 + 2C_\Theta H\kappa_3 K) + C_{d,\log K})}{K}} + \frac{1}{K}.$$

$\square$

Combing Lemma E.2 and Theorem G.2 (and a union bound), we directly have

**Corollary E.3.** *With probability* $1-\delta$,

$$\left\|\frac{1}{K}\Delta_{\Sigma_h^s}(\widehat{\theta}_h - \theta_{\mathbb{T}\widehat{V}_{h+1}})\right\|_2 \leq \left\|\frac{1}{K}\Delta_{\Sigma_h^s}\right\|_2\left\|\widehat{\theta}_h - \theta_{\mathbb{T}\widehat{V}_{h+1}}\right\|_2 \leq \widetilde{O}(\frac{\kappa_2\max(\frac{\kappa_1}{\kappa},\frac{1}{\sqrt{\kappa}})d^2H^2}{K})$$

*Here $\widetilde{O}$ absorbs all the constants and Polylog terms.*

Now we select $\theta = \theta_{\mathbb{T}\widehat{V}_{h+1}}$ in equation 4, and denote $\widetilde{R}_K(\theta_{\mathbb{T}\widehat{V}_{h+1}}) = \Delta_{\Sigma_h^s}(\widehat{\theta}_h - \theta_{\mathbb{T}\widehat{V}_{h+1}}) + R_K(\theta_{\mathbb{T}\widehat{V}_{h+1}})$, then equation 4 is equivalent to

$$Z_h(\theta_{\mathbb{T}\widehat{V}_{h+1}}|\widehat{V}_{h+1}) - Z_h(\widehat{\theta}_h|\widehat{V}_{h+1}) = \Sigma_h^s(\theta_{\mathbb{T}\widehat{V}_{h+1}} - \widehat{\theta}_h) + R_K(\theta_{\mathbb{T}\widehat{V}_{h+1}}) = \Sigma_h(\theta_{\mathbb{T}\widehat{V}_{h+1}} - \widehat{\theta}_h) + \widetilde{R}_K(\theta_{\mathbb{T}\widehat{V}_{h+1}})$$

Note $\lambda > 0$ implies $\Sigma_h$ is invertible, then we have (recall $Z_h(\widehat{\theta}_h|\widehat{\theta}_{h+1}) = 0$)

$$\theta_{\mathbb{T}\widehat{V}_{h+1}} - \widehat{\theta}_h = \Sigma_h^{-1}[Z_h(\theta_{\mathbb{T}\widehat{V}_{h+1}}|\widehat{V}_{h+1}) - Z_h(\widehat{\theta}_h|\widehat{V}_{h+1})] - \Sigma_h^{-1}\widetilde{R}_K(\theta_{\mathbb{T}\widehat{V}_{h+1}})$$

$$= \Sigma_h^{-1}[Z_h(\theta_{\mathbb{T}\widehat{V}_{h+1}}|\widehat{V}_{h+1})] - \Sigma_h^{-1}\widetilde{R}_K(\theta_{\mathbb{T}\widehat{V}_{h+1}})$$

Plug it back to equation 3 to get

$$
\nabla f(\widehat{\theta}_h, \phi(s,a)) \left( \theta_{\mathbb{T}\widehat{V}_{h+1}} - \widehat{\theta}_h \right)
$$
$$
= \nabla f(\widehat{\theta}_h, \phi(s,a)) \Sigma_h^{-1} [Z_h(\theta_{\mathbb{T}\widehat{V}_{h+1}} | \widehat{V}_{h+1})] - \nabla f(\widehat{\theta}_h, \phi(s,a)) \Sigma_h^{-1} \widetilde{R}_K(\theta_{\mathbb{T}\widehat{V}_{h+1}})
$$
$$
= \nabla f(\widehat{\theta}_h, \phi(s,a)) \Sigma_h^{-1} [\sum_{k=1}^{K} \left( f(\theta_{\mathbb{T}\widehat{V}_{h+1}}, \phi_{h,k}) - r_{h,k} - \widehat{V}_{h+1}(s_{h+1}^k) \right) \cdot \nabla_\theta^\top f(\theta_{\mathbb{T}\widehat{V}_{h+1}}, \phi_{h,k}) + \lambda \theta_{\mathbb{T}\widehat{V}_{h+1}}]
$$
$$
- \nabla f(\widehat{\theta}_h, \phi(s,a)) \Sigma_h^{-1} \widetilde{R}_K(\theta_{\mathbb{T}\widehat{V}_{h+1}})
$$
$$
= \underbrace{\nabla f(\widehat{\theta}_h, \phi(s,a)) \Sigma_h^{-1} [\sum_{k=1}^{K} \left( f(\theta_{\mathbb{T}\widehat{V}_{h+1}}, \phi_{h,k}) - r_{h,k} - \widehat{V}_{h+1}(s_{h+1}^k) \right) \cdot \nabla_\theta^\top f(\theta_{\mathbb{T}\widehat{V}_{h+1}}, \phi_{h,k})]}_{:=I}
$$
$$
- \underbrace{\nabla f(\widehat{\theta}_h, \phi(s,a)) \Sigma_h^{-1} \left[ \widetilde{R}_K(\theta_{\mathbb{T}\widehat{V}_{h+1}}) + \lambda \theta_{\mathbb{T}\widehat{V}_{h+1}} \right]}_{:=\mathrm{Hot}_2}
$$

(8)

We will bound second term $\mathrm{Hot}_2$ to have higher order $O(\frac{1}{K})$ in Section E.5 and focus on the first term. By direct decomposition,

$$
I := \nabla f(\widehat{\theta}_h, \phi(s,a)) \Sigma_h^{-1} [\sum_{k=1}^{K} \left( f(\theta_{\mathbb{T}\widehat{V}_{h+1}}, \phi_{h,k}) - r_{h,k} - \widehat{V}_{h+1}(s_{h+1}^k) \right) \cdot \nabla_\theta^\top f(\theta_{\mathbb{T}\widehat{V}_{h+1}}, \phi_{h,k})]
$$
$$
= \underbrace{\nabla f(\widehat{\theta}_h, \phi(s,a)) \Sigma_h^{-1} [\sum_{k=1}^{K} \left( f(\theta_{\mathbb{T}V_{h+1}^\star}, \phi_{h,k}) - r_{h,k} - V_{h+1}^\star(s_{h+1}^k) \right) \cdot \nabla_\theta^\top f(\widehat{\theta}_h, \phi_{h,k})]}_{:=I_1}
$$
$$
+ \underbrace{\nabla f(\widehat{\theta}_h, \phi(s,a)) \Sigma_h^{-1} [\sum_{k=1}^{K} \left( f(\theta_{\mathbb{T}\widehat{V}_{h+1}}, \phi_{h,k}) - f(\theta_{\mathbb{T}V_{h+1}^\star}, \phi_{h,k}) - \widehat{V}_{h+1}(s_{h+1}^k) + V_{h+1}^\star(s_{h+1}^k) \right) \cdot \nabla_\theta^\top f(\widehat{\theta}_h, \phi_{h,k})]}_{:=I_2}
$$
$$
+ \underbrace{\nabla f(\widehat{\theta}_h, \phi(s,a)) \Sigma_h^{-1} \left[ \sum_{k=1}^{K} \left( f(\theta_{\mathbb{T}\widehat{V}_{h+1}}, \phi_{h,k}) - r_{h,k} - \widehat{V}_{h+1}(s_{h+1}^k) \right) \cdot \left( \nabla_\theta^\top f(\theta_{\mathbb{T}\widehat{V}_{h+1}}, \phi_{h,k}) - \nabla_\theta^\top f(\widehat{\theta}_h, \phi_{h,k}) \right) \right]}_{:=I_3}
$$

### E.2 BOUNDING THE TERM $I_3$

We first bound the term $I_3$. We have the following Lemma.

**Lemma E.4.** *For any fixed $V(\cdot) \in \mathbb{R}^{\mathcal{S}}$ with $\|V\|_\infty \leq H$ and any fixed $\theta$ such that $\|\theta_{\mathbb{T}V} - \theta\|_2 \leq \sqrt{\frac{36H^2(\log(H/\delta) + C_{d,\log K}) + 2\lambda C_\Theta^2}{\kappa K}}$. Let*

$$
\widetilde{I}_3 := \nabla f(\widehat{\theta}_h, \phi(s,a))^\top \Sigma_h^{-1} \left[ \sum_{k=1}^{K} \left( f(\theta_{\mathbb{T}V}, \phi_{h,k}) - r_{h,k} - V(s_{h+1}^k) \right) \cdot \left( \nabla_\theta f(\theta_{\mathbb{T}V}, \phi_{h,k}) - \nabla_\theta f(\theta, \phi_{h,k}) \right) \right],
$$

*and if $K \geq \max \left\{ 512 \frac{\kappa_1^4}{\kappa^2} \left( \log(\frac{2d}{\delta}) + d \log(1 + \frac{4\kappa_1 D^2 \kappa_2 C_\Theta K^3}{\lambda^2}) \right), \frac{4\lambda}{\kappa} \right\}$, then with probability $1 - \delta$, (where $D = \max\{\kappa_1, \sqrt{\frac{(144dH^2\kappa_2^2(H^2 \log(H/\delta) + C_{d,\log K}) + 8dH^2\kappa_2^2\lambda C_\Theta^2)\log(d/\delta)}{\kappa}}\}$)*

$$
|\widetilde{I}_3| \leq 4\kappa_1 \sqrt{\frac{(144dH^2\kappa_2^2 \left( H^2 \log(H/\delta) + C_{d,\log K} \right) + 8dH^2\kappa_2^2\lambda C_\Theta^2) \log(d/\delta)}{\kappa^3}} \frac{1}{K} + O(\frac{1}{K^{3/2}}).
$$

*Proof of Lemma E.4.* Indeed, with probability $1 - \delta/2$,

$$|\widetilde{I}_3| = \left\| \nabla f(\widehat{\theta}_h, \phi(s,a))^\top \Sigma_h^{-1} \left[ \sum_{k=1}^K \left( f(\theta_{\mathbb{T}V}, \phi_{h,k}) - r_{h,k} - V(s_{h+1}^k) \right) \cdot (\nabla_\theta f(\theta_{\mathbb{T}V}, \phi_{h,k}) - \nabla_\theta f(\theta, \phi_{h,k})) \right] \right\|$$

$$\leq \left\| \nabla f(\widehat{\theta}_h, \phi(s,a)) \right\|_{\Sigma_h^{-1}} \left\| \sum_{k=1}^K \left( f(\theta_{\mathbb{T}V}, \phi_{h,k}) - r_{h,k} - V(s_{h+1}^k) \right) \cdot (\nabla_\theta f(\theta_{\mathbb{T}V}, \phi_{h,k}) - \nabla_\theta f(\theta, \phi_{h,k})) \right\|_{\Sigma_h^{-1}}$$

$$\leq \left( \frac{2\kappa_1}{\sqrt{\kappa K}} + O(\frac{1}{K}) \right) \left\| \sum_{k=1}^K \left( f(\theta_{\mathbb{T}V}, \phi_{h,k}) - r_{h,k} - V(s_{h+1}^k) \right) \cdot (\nabla_\theta f(\theta_{\mathbb{T}V}, \phi_{h,k}) - \nabla_\theta f(\theta, \phi_{h,k})) \right\|_{\Sigma_h^{-1}}$$

where, under the condition $K \geq \max \left\{ 512 \frac{\kappa_1^4}{\kappa^2} \left( \log(\frac{2d}{\delta}) + d \log(1 + \frac{4\kappa_1^3 \kappa_2 C_\Theta K^3}{\lambda^2}) \right), \frac{4\lambda}{\kappa} \right\}$, we applied Lemma L.5 .

Next, on one hand, $\|\nabla_\theta f(\theta_{\mathbb{T}V}, \phi_{h,k}) - \nabla_\theta f(\theta, \phi_{h,k})\|_2 \leq \kappa_2 \cdot \|\theta_{\mathbb{T}V} - \theta\|_2 \leq \kappa_2 \sqrt{\frac{36H^2(\log(H/\delta) + C_{d,\log K}) + 2\lambda C_\Theta^2}{\kappa K}}$. On the other hand,

$$\mathbb{E}\left[ \left( f(\theta_{\mathbb{T}V}, \phi_{h,k}) - r_{h,k} - V(s_{h+1}^k) \right) \cdot \left( \nabla_\theta^\top f(\theta_{\mathbb{T}V}, \phi_{h,k}) - \nabla_\theta^\top f(\theta, \phi_{h,k}) \right) \big| s_h^k, a_h^k \right]$$

$$= \mathbb{E}\left[ \left( f(\theta_{\mathbb{T}V}, \phi_{h,k}) - r_{h,k} - V(s_{h+1}^k) \right) \big| s_h^k, a_h^k \right] \cdot \left( \nabla_\theta^\top f(\theta_{\mathbb{T}V}, \phi_{h,k}) - \nabla_\theta^\top f(\theta, \phi_{h,k}) \right)$$

$$= \left( (\mathcal{P}_h V)(s_h^k, a_h^k) - (\mathcal{P}_h V)(s_h^k, a_h^k) \right) \cdot \left( \nabla_\theta^\top f(\theta_{\mathbb{T}V}, \phi_{h,k}) - \nabla_\theta^\top f(\theta, \phi_{h,k}) \right) = 0$$

Therefore by Vector Hoeffding's inequality (Lemma L.2) (also note the condition for boundedness $\left\| \left( f(\theta_{\mathbb{T}V}, \phi_{h,k}) - r_{h,k} - V(s_{h+1}^k) \right) \cdot (\nabla_\theta f(\theta_{\mathbb{T}V}, \phi_{h,k}) - \nabla_\theta f(\theta, \phi_{h,k})) \right\|_2 \leq H\kappa_2 \cdot \|\theta_{\mathbb{T}V} - \theta\|_2 \leq H\kappa_2 \sqrt{\frac{36H^2(\log(H/\delta) + C_{d,\log K}) + 2\lambda C_\Theta^2}{\kappa K}}$) with probability $1 - \delta/2$,

$$\left\| \frac{1}{K} \sum_{k=1}^K \left( f(\theta_{\mathbb{T}V}, \phi_{h,k}) - r_{h,k} - V(s_{h+1}^k) \right) \cdot (\nabla_\theta f(\theta_{\mathbb{T}V}, \phi_{h,k}) - \nabla_\theta f(\theta, \phi_{h,k})) \right\|_2$$

$$\leq \sqrt{\frac{4d \left( H\kappa_2 \sqrt{\frac{36H^2(\log(H/\delta) + C_{d,\log K}) + 2\lambda C_\Theta^2}{\kappa K}} \right)^2 \log(\frac{d}{\delta})}{K}}$$

$$= \sqrt{\frac{(144dH^2\kappa_2^2 \left( H^2 \log(H/\delta) + C_{d,\log K} \right) + 8dH^2\kappa_2^2 \lambda C_\Theta^2) \log(d/\delta)}{\kappa}} \cdot \frac{1}{K}$$

and this implies with probability $1 - \delta/2$,

$$\left\| \sum_{k=1}^K \left( f(\theta_{\mathbb{T}V}, \phi_{h,k}) - r_{h,k} - V(s_{h+1}^k) \right) \cdot (\nabla_\theta f(\theta_{\mathbb{T}V}, \phi_{h,k}) - \nabla_\theta f(\theta, \phi_{h,k})) \right\|_2$$

$$\leq \sqrt{\frac{(144dH^2\kappa_2^2 \left( H^2 \log(H/\delta) + C_{d,\log K} \right) + 8dH^2\kappa_2^2 \lambda C_\Theta^2) \log(d/\delta)}{\kappa}}$$

choose $u = \sum_{k=1}^K \left( f(\theta_{\mathbb{T}V}, \phi_{h,k}) - r_{h,k} - V(s_{h+1}^k) \right) \cdot (\nabla_\theta f(\theta_{\mathbb{T}V}, \phi_{h,k}) - \nabla_\theta f(\theta, \phi_{h,k}))$ in Lemma L.5, by a union bound we obtain with probability $1 - \delta$

$$|\widetilde{I}_3| = \left\| \nabla f(\widehat{\theta}_h, \phi(s,a))^\top \Sigma_h^{-1} \left[ \sum_{k=1}^K \left( f(\theta_{\mathbb{T}V}, \phi_{h,k}) - r_{h,k} - V(s_{h+1}^k) \right) \cdot (\nabla_\theta f(\theta_{\mathbb{T}V}, \phi_{h,k}) - \nabla_\theta f(\theta, \phi_{h,k})) \right] \right\|$$

$$\leq \left( \frac{2\kappa_1}{\sqrt{\kappa K}} + O(\frac{1}{K}) \right) \left\| \sum_{k=1}^K \left( f(\theta_{\mathbb{T}V}, \phi_{h,k}) - r_{h,k} - V(s_{h+1}^k) \right) \cdot (\nabla_\theta f(\theta_{\mathbb{T}V}, \phi_{h,k}) - \nabla_\theta f(\theta, \phi_{h,k})) \right\|_{\Sigma_h^{-1}}$$

$$\leq \left( \frac{2\kappa_1}{\sqrt{\kappa K}} + O(\frac{1}{K}) \right) \left( 2\sqrt{\frac{(144dH^2\kappa_2^2 \left( H^2 \log(H/\delta) + C_{d,\log K} \right) + 8dH^2\kappa_2^2 \lambda C_\Theta^2) \log(d/\delta)}{\kappa^2 K}} + O(\frac{1}{K}) \right)$$

$$= 4\kappa_1 \sqrt{\frac{(144dH^2\kappa_2^2 \left( H^2 \log(H/\delta) + C_{d,\log K} \right) + 8dH^2\kappa_2^2 \lambda C_\Theta^2) \log(d/\delta)}{\kappa^3}} \frac{1}{K} + O(\frac{1}{K^{3/2}}).$$

$\square$

**Lemma E.5.** *Under the same condition as Lemma E.4. With probability $1 - \delta$,*

$$|I_3| \le 4\kappa_1 \sqrt{\frac{(144 dH^2 \kappa_2^2 \left(H^2 \log(H/\delta) + D_{d,\log K} + C_{d,\log K}\right) + 8 dH^2 \kappa_2^2 \lambda C_\Theta^2)(\log(d/\delta) + D_{d,\log K})}{\kappa^3}} \frac{1}{K} + O(\frac{1}{K^{3/2}}).$$

*Here* $D_{d,\log K}$ $:=$ $d \cdot \log(1 + 6C_\Theta(2\kappa_1^2 + H\kappa_2)K) + d\log(1 + 6C_\Theta H\kappa_2 K) + d\log\left(1 + 288 C_\Theta \kappa_1^2 (\kappa_1 \sqrt{C_\Theta} + 2\sqrt{B\kappa_1\kappa_2})^2 K^2\right) + d^2 \log\left(1 + 288\sqrt{d}B\kappa_1^4 K^2\right)$ $=$ $\widetilde{O}(d^2)$ *with* $\widetilde{O}$ *absorbs Polylog terms.*

*Proof of Lemma E.5.* Define

$$h(V, \widetilde{\theta}, \theta) = \sum_{k=1}^{K} \left( f(\widetilde{\theta}, \phi_{h,k}) - r_{h,k} - V(s_{h+1}^k) \right) \cdot \left( \nabla_\theta f(\widetilde{\theta}, \phi_{h,k}) - \nabla_\theta f(\theta, \phi_{h,k}) \right),$$

then

$$|h(V_1, \widetilde{\theta}_1, \theta_1) - h(V_2, \widetilde{\theta}_2, \theta_2)|$$

$$\le \left| \sum_{k=1}^{K} \left( [f(\widetilde{\theta}_1, \phi_{h,k}) - V_1(s_{h+1}^k)] - [f(\widetilde{\theta}_2, \phi_{h,k}) - V_2(s_{h+1}^k)] \right) \cdot \left( \nabla_\theta f(\widetilde{\theta}_1, \phi_{h,k}) - \nabla_\theta f(\theta_1, \phi_{h,k}) \right) \right|$$

$$+ \left| \sum_{k=1}^{K} \left( f(\widetilde{\theta}_2, \phi_{h,k}) - r_{h,k} - V_2(s_{h+1}^k) \right) \cdot \left( [\nabla_\theta f(\widetilde{\theta}_1, \phi_{h,k}) - \nabla_\theta f(\theta_1, \phi_{h,k})] - [\nabla_\theta f(\widetilde{\theta}_2, \phi_{h,k}) - \nabla_\theta f(\theta_2, \phi_{h,k})] \right) \right|$$

$$\le K \sup_{s,a,s'} \left| [f(\widetilde{\theta}_1, \phi(s,a)) - f(\widetilde{\theta}_2, \phi(s,a))] - [V_1(s') - V_2(s')] \right|_2 \cdot 2\kappa_1$$

$$+ KH \cdot \sup_{s,a} \left\| [\nabla_\theta f(\widetilde{\theta}_1, \phi(s,a)) - \nabla_\theta f(\theta_1, \phi(s,a))] - [\nabla_\theta f(\widetilde{\theta}_2, \phi(s,a)) - \nabla_\theta f(\theta_2, \phi(s,a))] \right\|_2$$

$$\le K 2\kappa_1^2 \left\| \widetilde{\theta}_1 - \widetilde{\theta}_2 \right\|_2 + 2K\kappa_1 \|V_1 - V_2\|_\infty + HK\kappa_2 \left\| \widetilde{\theta}_1 - \widetilde{\theta}_2 \right\|_2 + HK\kappa_2 \|\theta_1 - \theta_2\|_2$$

$$= (2\kappa_1^2 + H\kappa_2)K \left\| \widetilde{\theta}_1 - \widetilde{\theta}_2 \right\|_2 + 2\kappa_1 K \|V_1 - V_2\|_\infty + HK\kappa_2 \|\theta_1 - \theta_2\|_2.$$

Let $\mathcal{C}_a$ be the $\frac{\epsilon/3}{(2\kappa_1^2 + H\kappa_2)K}$-covering net of $\{\theta : \|\theta\|_2 \le C_\Theta\}$, $\mathcal{C}_V$ be the $\frac{\epsilon}{6\kappa_1 K}$-covering net of $\mathcal{V}$ defined in Lemma L.9 and $\mathcal{C}_b$ be the $\frac{\epsilon}{3H\kappa_2 K}$-covering net of $\{\theta : \|\theta\|_2 \le C_\Theta\}$, then by Lemma L.8 and Lemma L.9,

$$\log |\mathcal{C}_a| \le d \cdot \log(1 + \frac{6C_\Theta(2\kappa_1^2 + H\kappa_2)K}{\epsilon}), \quad \log |\mathcal{C}_b| \le d\log(1 + \frac{6C_\Theta H\kappa_2 K}{\epsilon})$$

$$\log \mathcal{C}_V \le d\log\left(1 + \frac{288 C_\Theta \kappa_1^2 (\kappa_1 \sqrt{C_\Theta} + 2\sqrt{B\kappa_1\kappa_2})^2 K^2}{\epsilon^2}\right) + d^2 \log\left(1 + \frac{288\sqrt{d}B\kappa_1^4 K^2}{\epsilon^2}\right).$$

Further notice with probability $1 - \delta/2$ (by Lemma L.5), for all fixed sets of parameters $\theta, V$ satisfies $\|\theta_{\mathbb{T}V} - \theta\|_2 \le \sqrt{\frac{36H^2(\log(2H/\delta) + C_{d,\log K}) + 2\lambda C_\Theta^2}{\kappa K}}$ simultaneously,

$$|I_3 - \widetilde{I}_3| \le \left\| \nabla f(\widehat{\theta}_h, \phi(s,a)) \right\|_{\Sigma_h^{-1}} \cdot \left\| h(\widehat{V}_{h+1}, \theta_{\mathbb{T}\widehat{V}_{h+1}}, \widehat{\theta}_h) - h(V, \theta_{\mathbb{T}V}, \theta) \right\|_{\Sigma_h^{-1}}$$

$$\le \left( \frac{2\kappa_1}{\sqrt{\kappa K}} + O(\frac{1}{K}) \right) \cdot \left\| h(\widehat{V}_{h+1}, \theta_{\mathbb{T}\widehat{V}_{h+1}}, \widehat{\theta}_h) - h(V, \theta_{\mathbb{T}V}, \theta) \right\|_{\Sigma_h^{-1}}$$

and $\left\| \theta_{\mathbb{T}\widehat{V}_{h+1}} - \widehat{\theta}_h \right\|_2 \le \sqrt{\frac{36H^2(\log(2H/\delta) + C_{d,\log K}) + 2\lambda C_\Theta^2}{\kappa K}}$ with probability $1 - \delta/2$ by Theorem G.2.
Now, choosing $\epsilon = O(1/K^2)$ and by Lemma E.4 and union bound over covering instances, we obtain with probability $1 - \delta$

$$|I_3| \le 4\kappa_1 \sqrt{\frac{(144 dH^2 \kappa_2^2 \left(H^2 \log(H/\delta) + D_{d,\log K} + C_{d,\log K}\right) + 8 dH^2 \kappa_2^2 \lambda C_\Theta^2)(\log(d/\delta) + D_{d,\log K})}{\kappa^3}} \frac{1}{K} + O(\frac{1}{K^{3/2}}).$$

$\square$

### E.3 BOUNDING THE SECOND TERM $I_2$

In this section, we bound the term

$$I_2 := \nabla f(\widehat{\theta}_h, \phi(s,a)) \Sigma_h^{-1} [\sum_{k=1}^K \Big( f(\theta_{\mathbb{T}\widehat{V}_{h+1}}, \phi_{h,k}) - f(\theta_{\mathbb{T}V_{h+1}^\star}, \phi_{h,k}) - \widehat{V}_{h+1}(s_{h+1}^k) + V_{h+1}^\star(s_{h+1}^k) \Big) \cdot \nabla_\theta^\top f(\widehat{\theta}_h, \phi_{h,k})].$$

The following Lemma shows that $I_2$ is a higher-order error term with rate $\widetilde{O}(\frac{1}{K})$.

**Lemma E.6** (Bounding $I_2$). *If $K$ satisfies $K \geq 512 \frac{\kappa_1^4}{\kappa^2} \left( \log(\frac{2d}{\delta}) + d \log(1 + \frac{4\kappa_1^3 \kappa_2 C_\Theta K}{\lambda^2}) \right)$, and $K \geq 4\lambda/\kappa$, then with probability $1 - \delta$*

$$|I_2| \leq \widetilde{O}(\frac{\kappa_1^2 H^2 d^2}{\kappa K}) + \widetilde{O}(\frac{1}{K^{3/2}}).$$

*Here $\widetilde{O}$ absorbs constants and Polylog terms.*

*Proof of Lemma E.6.* **Step1.** Define $\eta_k(V) := f(\theta_{\mathbb{T}V}, \phi_{h,k}) - f(\theta_{\mathbb{T}V_{h+1}^\star}, \phi_{h,k}) - V(s_{h+1}^k) + V_{h+1}^\star(s_{h+1}^k)$ and let $\|V(\cdot)\|_\infty \leq H$ be any fixed function such that $\sup_{s_h^k, a_h^k, s_{h+1}^k} |\eta_k(V)| \leq \widetilde{O}(\kappa_1 H^2 \sqrt{\frac{d^2}{\kappa K}})$, *i.e.* arbitrary fixed $V$ function in the neighborhood (measured by $\eta_k$) of $V_{h+1}^\star$. Then by definition of $\mathbb{T}$ it holds $\mathbb{E}[\eta_k(V, \theta)|s_h^k, a_h^k] = 0$. Let the fixed $\theta \in \Theta$ be arbitrary and define $x_k(\theta) = \nabla_\theta f(\theta, \phi_{h,k})$. Next, define $G_h(\theta) = \sum_{k=1}^K \nabla f(\theta, \phi(s_h^k, a_h^k)) \cdot \nabla f(\theta, \phi(s_h^k, a_h^k))^\top + \lambda I_d$, since $\|x_k\|_2 \leq \kappa_1$ and $|\eta_k| \leq \widetilde{O}(\kappa_1 H^2 \sqrt{\frac{d^2}{\kappa K}})$, by self-normalized Hoeffding's inequality (Lemma L.3), with probability $1 - \delta$ (recall $t := K$ in Lemma L.3),

$$\left\| \sum_{k=1}^K x_k(\theta) \eta_k(V) \right\|_{G_h(\theta)^{-1}} \leq \widetilde{O}(\kappa_1 H^2 \sqrt{\frac{d^2}{\kappa K}}) \sqrt{d \log \left( \frac{\lambda + K\kappa_1}{\lambda \delta} \right)}.$$

**Step2.** Define $h(V, \theta) := \sum_{k=1}^K x_k(\theta) \eta_k(V)$ and $H(V, \theta) := \left\| \sum_{k=1}^K x_k(\theta) \eta_k(V) \right\|_{G_h(\theta)^{-1}}$, then note by definition $|\eta_k(V)| \leq 2H$, which implies $\|h(V, \theta)\|_2 \leq 2KH\kappa_1$ and

$$|\eta_k(V_1) - \eta_k(V_2)| \leq |\mathcal{P}_h V_1 - \mathcal{P}_h V_2| + \|V_1 - V_2\|_\infty \leq 2 \|V_1 - V_2\|_\infty$$

and

$$\|h(V_1, \theta_1) - h(V_2, \theta_2)\|_2 \leq K \max_k \left( 2H \|x_k(\theta_1) - x_k(\theta_2)\|_2 + \kappa_1 |\eta_k(V_1) - \eta_k(V_2)| \right)$$

$$\leq K(2H\kappa_2 \|\theta_1 - \theta_2\|_2 + 2\kappa_1 \|V_1 - V_2\|_\infty).$$

Furthermore,

$$\left\| G_h(\theta_1)^{-1} - G_h(\theta_2)^{-1} \right\|_2 \leq \left\| G_h(\theta_1)^{-1} \right\|_2 \left\| G_h(\theta_1) - G_h(\theta_2) \right\|_2 \left\| G_h(\theta_2)^{-1} \right\|_2$$

$$\leq \frac{1}{\lambda^2} K \sup_k \left\| \nabla f(\theta_1, \phi_{h,k}) \cdot \nabla f(\theta_1, \phi_{h,k})^\top - \nabla f(\theta_2, \phi_{h,k}) \cdot \nabla f(\theta_2, \phi_{h,k})^\top \right\|_2$$

$$\leq \frac{1}{\lambda^2} K \sup_k \left[ \left\| (\nabla f(\theta_1, \phi_{h,k}) - \nabla f(\theta_2, \phi_{h,k})) \cdot \nabla f(\theta_1, \phi_{h,k})^\top \right\|_2 + \left\| \nabla f(\theta_2, \phi_{h,k}) \cdot (\nabla f(\theta_1, \phi_{h,k})^\top - \nabla f(\theta_2, \phi_{h,k})^\top) \right\|_2 \right]$$

$$\leq \frac{2\kappa_1 K}{\lambda^2} \kappa_2 \|\theta_1 - \theta_2\|_2 = \frac{2\kappa_1 \kappa_2 K}{\lambda^2} \|\theta_1 - \theta_2\|_2.$$

All the above imply

$$|H(V_1, \theta_1) - H(V_2, \theta_2)| \leq \sqrt{|h(V_1, \theta_1)^\top G_h(\theta_1)^{-1} h(V_1, \theta_1) - h(V_2, \theta_2)^\top G_h(\theta_2)^{-1} h(V_2, \theta_2)|}$$

$$\leq \sqrt{\|h(V_1, \theta_1) - h(V_2, \theta_2)\|_2 \cdot \frac{1}{\lambda} \cdot 2KH\kappa_1} + \sqrt{2KH\kappa_1 \cdot \|G_h(\theta_1)^{-1} - G_h(\theta_2)^{-1}\|_2 \cdot 2KH\kappa_1}$$

$$+ \sqrt{2KH\kappa_1 \cdot \frac{1}{\lambda} \cdot \|h(V_1, \theta_1) - h(V_2, \theta_2)\|_2}$$

$$\leq 2\sqrt{K(2H\kappa_2 \|\theta_1 - \theta_2\|_2 + 2\kappa_1 \|V_1 - V_2\|_\infty) \cdot \frac{1}{\lambda} \cdot 2KH\kappa_1} + \sqrt{2KH\kappa_1 \cdot \frac{2\kappa_1 \kappa_2 K}{\lambda^2} \|\theta_1 - \theta_2\|_2 \cdot 2KH\kappa_1}$$

$$\leq \left( 4\sqrt{K^3 H^2 \kappa_1 \kappa_2 \frac{1}{\lambda}} + \sqrt{8K^3 H^2 \kappa_1^3 \kappa_2 \frac{1}{\lambda^2}} \right) \sqrt{\|\theta_1 - \theta_2\|_2} + 4\sqrt{K^3 \kappa_1^2 H \frac{1}{\lambda}} \|V_1 - V_2\|_\infty$$

Then a $\epsilon$-covering net of $\{H(V,\theta)\}$ can be constructed by the union of $\frac{\epsilon^2}{4\left(4\sqrt{K^3H^2\kappa_1\kappa_2\frac{1}{\lambda}}+\sqrt{8K^3H^2\kappa_1^3\kappa_2\frac{1}{\lambda^2}}\right)^2}$-covering net of $\{\theta \in \Theta\}$ and $\frac{\epsilon^2}{4(4\sqrt{K^3\kappa_1^2H\frac{1}{\lambda}})^2}$-covering net of $\mathcal{V}$ in Lemma L.9. The covering number $\mathcal{N}_\epsilon$ satisfies

$$\log\mathcal{N}_\epsilon \leq d\log\left(1 + \frac{8C_\Theta\left(4\sqrt{K^3H^2\kappa_1\kappa_2\frac{1}{\lambda}}+\sqrt{8K^3H^2\kappa_1^3\kappa_2\frac{1}{\lambda^2}}\right)^2}{\epsilon^2}\right)$$

$$+ d\log\left(1 + \frac{8C_\Theta(\kappa_1\sqrt{C_\Theta}+2\sqrt{B\kappa_1\kappa_2})^2}{\frac{\epsilon^4}{16(4\sqrt{K^3\kappa_1^2H\frac{1}{\lambda}})^4}}\right) + d^2\log\left(1 + \frac{8\sqrt{d}B\kappa_1^2}{\frac{\epsilon^4}{16(4\sqrt{K^3\kappa_1^2H\frac{1}{\lambda}})^4}}\right).$$

**Step3.** First note by definition in Step2

$$\left\|\sum_{k=1}^K\left(f(\theta_{\mathbb{T}\widehat{V}_{h+1}},\phi_{h,k}) - f(\theta_{\mathbb{T}V_{h+1}^\star},\phi_{h,k}) - \widehat{V}_{h+1}(s_{h+1}^k) + V_{h+1}^\star(s_{h+1}^k)\right)\cdot\nabla_\theta^\top f(\widehat{\theta}_h,\phi_{h,k})\right\|_{\Sigma_h^{-1}} = H(\widehat{V}_{h+1},\widehat{\theta}_h)$$

and with probability $1 - \delta$

$$|\eta_k(\widehat{V}_{h+1})| = |f(\theta_{\mathbb{T}\widehat{V}_{h+1}},\phi_{h,k}) - f(\theta_{\mathbb{T}V_{h+1}^\star},\phi_{h,k}) - \widehat{V}_{h+1}(s_{h+1}^k) + V_{h+1}^\star(s_{h+1}^k)|$$

$$\leq \kappa_1\cdot\left\|\theta_{\mathbb{T}\widehat{V}_{h+1}} - \theta_h^\star\right\|_2 + \left\|\widehat{V}_{h+1} - V_{h+1}^\star\right\|_\infty$$

$$\leq \kappa_1\sqrt{\frac{36H^2(\log(H/\delta) + C_{d,\log K}) + 2\lambda C_\Theta^2}{\kappa K}} + C\left(\kappa_1H^2\sqrt{\frac{d^2}{\kappa K}}\right) = \widetilde{O}\left(\kappa_1H^2\sqrt{\frac{d^2}{\kappa K}}\right)$$

(9)

where the second inequality uses $\theta_{\mathbb{T}V_{h+1}^\star} = \theta_h^\star$ and the third inequality uses Theorem G.2 and Theorem G.3. The last equal sign is due to $C_{d,\log K} \leq \widetilde{O}(d^2)$ (recall Lemma G.1).

Now choosing $\epsilon = O(1/K)$ in Step2 and union bound over both equation 9 and covering number in Step2, we obtain with probability $1 - \delta$,

$$H(\widehat{V}_{h+1},\widehat{\theta}_h) = \left\|\sum_{k=1}^K x_k(\widehat{\theta}_h)\eta_k(\widehat{V}_{h+1})\right\|_{G_h(\widehat{\theta}_h)^{-1}} \leq \widetilde{O}(\kappa_1H^2\sqrt{\frac{d^2}{\kappa K}})\sqrt{d + d^2} = \widetilde{O}(\frac{\kappa_1H^2d^2}{\sqrt{\kappa K}})$$

(10)

where we absorb all the Polylog terms. Meanwhile, by Lemma L.5 with probability $1 - \delta$,

$$\left\|\nabla f(\widehat{\theta}_h,\phi_{s,a})\right\|_{\Sigma_h^{-1}} \leq \frac{2\kappa_1}{\sqrt{\kappa K}} + O(\frac{1}{K}).$$

(11)

Finally, by equation 10 and equation 11 and a union bound, we have with probability $1 - \delta$,

$$|I_2| := \left|\nabla f(\widehat{\theta}_h,\phi(s,a))\Sigma_h^{-1}[\sum_{k=1}^K\left(f(\theta_{\mathbb{T}\widehat{V}_{h+1}},\phi_{h,k}) - f(\theta_{\mathbb{T}V_{h+1}^\star},\phi_{h,k}) - \widehat{V}_{h+1}(s_{h+1}^k) + V_{h+1}^\star(s_{h+1}^k)\right)\cdot\nabla_\theta^\top f(\widehat{\theta}_h,\phi_{h,k})]\right|$$

$$\leq \left\|\nabla f(\widehat{\theta}_h,\phi_{s,a})\right\|_{\Sigma_h^{-1}}\left\|\sum_{k=1}^K\left(f(\theta_{\mathbb{T}\widehat{V}_{h+1}},\phi_{h,k}) - f(\theta_{\mathbb{T}V_{h+1}^\star},\phi_{h,k}) - \widehat{V}_{h+1}(s_{h+1}^k) + V_{h+1}^\star(s_{h+1}^k)\right)\cdot\nabla_\theta^\top f(\widehat{\theta}_h,\phi_{h,k})\right\|_{\Sigma_h^{-1}}$$

$$= \left\|\nabla f(\widehat{\theta}_h,\phi_{s,a})\right\|_{\Sigma_h^{-1}}\cdot H(\widehat{V}_{h+1},\widehat{\theta}_h) \leq \left(\frac{2\kappa_1}{\sqrt{\kappa K}} + O(\frac{1}{K})\right)\cdot\widetilde{O}(\frac{\kappa_1H^2d^2}{\sqrt{\kappa K}}) = \widetilde{O}(\frac{\kappa_1^2H^2d^2}{\kappa K}) + \widetilde{O}(\frac{1}{K^{3/2}})$$

where the first inequality is CauchySchwarz inequality. $\square$

### E.4 BOUNDING THE MAIN TERM $I_1$

In this section, we bound the dominate term

$$I_1 := \nabla f(\widehat{\theta}_h,\phi(s,a))\Sigma_h^{-1}[\sum_{k=1}^K\left(f(\theta_{\mathbb{T}V_{h+1}^\star},\phi_{h,k}) - r_{h,k} - V_{h+1}^\star(s_{h+1}^k)\right)\cdot\nabla_\theta^\top f(\widehat{\theta}_h,\phi_{h,k})].$$

First of all, by CauchySchwarz inequality, we have

$$|I_1| \leq \left\| \nabla f(\widehat{\theta}_h, \phi(s,a)) \right\|_{\Sigma_h^{-1}} \cdot \left\| \sum_{k=1}^{K} \left( f(\theta_{\mathbb{T}V_{h+1}^\star}, \phi_{h,k}) - r_{h,k} - V_{h+1}^\star(s_{h+1}^k) \right) \cdot \nabla_\theta^\top f(\widehat{\theta}_h, \phi_{h,k}) \right\|_{\Sigma_h^{-1}}.$$

(12)

Then we have the following Lemma to bound $I_1$.

**Lemma E.7.** *With probability $1 - \delta$,*

$$|I_1| \leq 4Hd \left\| \nabla f(\widehat{\theta}_h, \phi(s,a)) \right\|_{\Sigma_h^{-1}} \cdot C_{\delta, \log K} + \widetilde{O}(\frac{\kappa_1}{\sqrt{\kappa}K}),$$

*where $C_{\delta, \log K}$ only contains Polylog terms.*

*Proof of Lemma E.7.* **Step1.** Let the fixed $\theta \in \Theta$ be arbitrary and define $x_k(\theta) = \nabla_\theta f(\theta, \phi_{h,k})$. Next, define $G_h(\theta) = \sum_{k=1}^{K} \nabla f(\theta, \phi(s_h^k, a_h^k)) \cdot \nabla f(\theta, \phi(s_h^k, a_h^k))^\top + \lambda I_d$, then $\|x_k\|_2 \leq \kappa_1$. Also denote $\eta_k := f(\theta_{\mathbb{T}V_{h+1}^\star}, \phi_{h,k}) - r_{h,k} - V_{h+1}^\star(s_{h+1}^k)$, then $\mathbb{E}[\eta_k | s_h^k, a_h^k] = 0$ and $|\eta_k| \leq H$. Now by self-normalized Hoeffding's inequality (Lemma L.3), with probability $1 - \delta$ (recall $t := K$ in Lemma L.3),

$$\left\| \sum_{k=1}^{K} x_k(\theta)\eta_k \right\|_{G_h(\theta)^{-1}} \leq 2H\sqrt{d \log\left(\frac{\lambda + K\kappa_1}{\lambda\delta}\right)}.$$

**Step2.** Define $h(\theta) := \sum_{k=1}^{K} x_k(\theta)\eta_k$ and $H(\theta) := \left\| \sum_{k=1}^{K} x_k(\theta)\eta_k \right\|_{G_h(\theta)^{-1}}$, then note by definition $|\eta_k| \leq H$, which implies $\|h(\theta)\|_2 \leq KH\kappa_1$ and by $x_k(\theta_1) - x_k(\theta_2) = \nabla_{\theta\theta}^2 f(\xi, \phi) \cdot (\theta_1 - \theta_2)$,

$$\|h(\theta_1) - h(\theta_2)\|_2 \leq K \max_k \left(H \|x_k(\theta_1) - x_k(\theta_2)\|_2\right) \leq HK\kappa_2 \|\theta_1 - \theta_2\|_2.$$

Furthermore,

$$\left\| G_h(\theta_1)^{-1} - G_h(\theta_2)^{-1} \right\|_2 \leq \left\| G_h(\theta_1)^{-1} \right\|_2 \left\| G_h(\theta_1) - G_h(\theta_2) \right\|_2 \left\| G_h(\theta_2)^{-1} \right\|_2$$

$$\leq \frac{1}{\lambda^2} K \sup_k \left\| \nabla f(\theta_1, \phi_{h,k}) \cdot \nabla f(\theta_1, \phi_{h,k})^\top - \nabla f(\theta_2, \phi_{h,k}) \cdot \nabla f(\theta_2, \phi_{h,k})^\top \right\|_2$$

$$\leq \frac{2\kappa_1 K}{\lambda^2} \kappa_2 \|\theta_1 - \theta_2\|_2 = \frac{2\kappa_1\kappa_2 K}{\lambda^2} \|\theta_1 - \theta_2\|_2.$$

All the above imply

$$|H(\theta_1) - H(\theta_2)| \leq \sqrt{|h(\theta_1)^\top G_h(\theta_1)^{-1} h(\theta_1) - h(\theta_2)^\top G_h(\theta_2)^{-1} h(\theta_2)|}$$

$$\leq \sqrt{\|h(\theta_1) - h(\theta_2)\|_2 \cdot \frac{1}{\lambda} \cdot KH\kappa_1} + \sqrt{KH\kappa_1 \cdot \left\| G_h(\theta_1)^{-1} - G_h(\theta_2)^{-1} \right\|_2 \cdot KH\kappa_1}$$

$$+ \sqrt{KH\kappa_1 \cdot \frac{1}{\lambda} \cdot \|h(\theta_1) - h(\theta_2)\|_2}$$

$$\leq 2\sqrt{KH\kappa_2 \|\theta_1 - \theta_2\|_2 \cdot \frac{1}{\lambda} \cdot KH\kappa_1} + \sqrt{KH\kappa_1 \cdot \frac{2\kappa_1\kappa_2 K}{\lambda^2} \|\theta_1 - \theta_2\|_2 \cdot KH\kappa_1}$$

$$\leq \left( \sqrt{4K^2 H^2 \kappa_1 \kappa_2/\lambda} + \sqrt{2K^3 H^2 \kappa_1^3 \kappa_2/\lambda^2} \right) \sqrt{\|\theta_1 - \theta_2\|_2}$$

Then a $\epsilon$-covering net of $\{H(\theta)\}$ can be constructed by the union of $\frac{\epsilon^2}{\left(\sqrt{4K^2 H^2 \kappa_1 \kappa_2/\lambda} + \sqrt{2K^3 H^2 \kappa_1^3 \kappa_2/\lambda^2}\right)^2}$-covering net of $\{\theta \in \Theta\}$. By Lemma L.8, the covering number $\mathcal{N}_\epsilon$ satisfies

$$\log \mathcal{N}_\epsilon \leq d \log \left( 1 + \frac{2C_\Theta \left( \sqrt{4K^2 H^2 \kappa_1 \kappa_2/\lambda} + \sqrt{2K^3 H^2 \kappa_1^3 \kappa_2/\lambda^2} \right)^2}{\epsilon^2} \right) = \widetilde{O}(d)$$

**Step3.** First note by definition in Step2

$$\left\| \sum_{k=1}^{K} \left( f(\theta_{\mathbb{T}V_{h+1}^\star}, \phi_{h,k}) - r_{h,k} - V_{h+1}^\star(s_{h+1}^k) \right) \cdot \nabla_\theta^\top f(\widehat{\theta}_h, \phi_{h,k}) \right\|_{\Sigma_h^{-1}} = H(\widehat{\theta}_h)$$

Now choosing $\epsilon = O(1/K)$ in Step2 and union bound over the covering number in Step2, we obtain with probability $1 - \delta$,

$$H(\widehat{\theta}_h) = \left\| \sum_{k=1}^{K} x_k(\widehat{\theta}_h)\eta_k \right\|_{G_h(\widehat{\theta}_h)^{-1}} \leq 2H\sqrt{d \left[ \log\left( \frac{\lambda + K\kappa_1}{\lambda\delta} \right) + \widetilde{O}(d) \right]} + O(\frac{1}{K}). \tag{13}$$

where we absorb all the Polylog terms. Combing above with equation 12, we obtain with probability $1 - \delta$,

$$|I_1| \leq \left\| \nabla f(\widehat{\theta}_h, \phi(s,a)) \right\|_{\Sigma_h^{-1}} \cdot \left\| \sum_{k=1}^{K} \left( f(\theta_{\mathbb{T}V_{h+1}^\star}, \phi_{h,k}) - r_{h,k} - V_{h+1}^\star(s_{h+1}^k) \right) \cdot \nabla_\theta^\top f(\widehat{\theta}_h, \phi_{h,k}) \right\|_{\Sigma_h^{-1}}$$

$$\leq \left\| \nabla f(\widehat{\theta}_h, \phi(s,a)) \right\|_{\Sigma_h^{-1}} \cdot \left( 2H\sqrt{d \left[ \log\left( \frac{\lambda + K\kappa_1}{\lambda\delta} \right) + \widetilde{O}(d) \right]} + O(\frac{1}{K}) \right)$$

$$\leq 4Hd \left\| \nabla f(\widehat{\theta}_h, \phi(s,a)) \right\|_{\Sigma_h^{-1}} \cdot C_{\delta,\log K} + \widetilde{O}(\frac{\kappa_1}{\sqrt{\kappa}K}),$$

where $C_{\delta,\log K}$ only contains Polylog terms. $\qquad\square$

### E.5 ANALYZING $\mathrm{Hot}_2$ IN EQUATION 8

**Lemma E.8.** *Recall* $\mathrm{Hot}_2 := \nabla f(\widehat{\theta}_h, \phi(s,a))\Sigma_h^{-1} \left[ \widetilde{R}_K(\theta_{\mathbb{T}\widehat{V}_{h+1}}) + \lambda\theta_{\mathbb{T}\widehat{V}_{h+1}} \right]$. *If the number of episode $K$ satisfies* $K \geq \max\left\{ 512\frac{\kappa_1^4}{\kappa^2}\left( \log(\frac{2d}{\delta}) + d\log(1 + \frac{4\kappa_1^3\kappa_2 C_\Theta K^3}{\kappa\lambda^2}) \right), \frac{4\lambda}{\kappa} \right\}$, *then with probability $1 - \delta$,*

$$\left| \nabla f(\widehat{\theta}_h, \phi(s,a))\Sigma_h^{-1} \left[ \widetilde{R}_K(\theta_{\mathbb{T}\widehat{V}_{h+1}}) + \lambda\theta_{\mathbb{T}\widehat{V}_{h+1}} \right] \right| \leq \widetilde{O}\left( \frac{\kappa_2 \max(\frac{\kappa_1}{\kappa}, \frac{1}{\sqrt{\kappa}})d^2 H^2 + \frac{d^2 H^3\kappa_3 + \lambda\kappa_1 C_\Theta}{\kappa}}{K} \right)$$

*where $\widetilde{O}$ absorbs all the constants and Polylog terms.*

*Proof of Lemma E.8.* **Step1:** we first show with probability $1 - \delta$

$$\left| \nabla f(\widehat{\theta}_h, \phi(s,a))\Sigma_h^{-1}\widetilde{R}_K(\theta_{\mathbb{T}\widehat{V}_{h+1}}) \right| \leq \widetilde{O}(\frac{1}{K}).$$

Recall by plug in $\theta_{\mathbb{T}\widehat{V}_{h+1}}$ in equation 4, we have

$$Z_h(\theta_{\mathbb{T}\widehat{V}_{h+1}}|\widehat{V}_{h+1}) - Z_h(\widehat{\theta}_h|\widehat{V}_{h+1}) = \frac{\partial}{\partial\theta} Z_h(\widehat{\theta}_h|\widehat{V}_{h+1})(\theta_{\mathbb{T}\widehat{V}_{h+1}} - \widehat{\theta}_h) + R_K(\theta_{\mathbb{T}\widehat{V}_{h+1}}), \tag{14}$$

and by second-order Taylor's Theorem we have

$$\left\| R_K(\theta_{\mathbb{T}\widehat{V}_{h+1}}) \right\|_2 = \left\| Z_h(\theta_{\mathbb{T}\widehat{V}_{h+1}}|\widehat{V}_{h+1}) - Z_h(\widehat{\theta}_h|\widehat{V}_{h+1}) - \frac{\partial}{\partial\theta} Z_h(\widehat{\theta}_h|\widehat{V}_{h+1})(\theta_{\mathbb{T}\widehat{V}_{h+1}} - \widehat{\theta}_h) \right\|_2$$

$$= \frac{1}{2} \left\| (\theta_{\mathbb{T}\widehat{V}_{h+1}} - \widehat{\theta}_h)^\top \frac{\partial^2}{\partial\theta\partial\theta} Z_h(\xi|\widehat{V}_{h+1})(\theta_{\mathbb{T}\widehat{V}_{h+1}} - \widehat{\theta}_h) \right\|_2 \tag{15}$$

$$\leq \frac{1}{2}\kappa_{z_2} \left\| \theta_{\mathbb{T}\widehat{V}_{h+1}} - \widehat{\theta}_h \right\|_2^2$$

Note

$$\frac{\partial^2}{\partial\theta\theta}Z_h(\theta|\widehat{V}_{h+1})\Big|_{\theta=\xi} = \frac{\partial}{\partial\theta}\Sigma_h^s = \sum_{k=1}^K \frac{\partial}{\partial\theta}\left\{\left(f(\xi,\phi_{h,k}) - r_{h,k} - \widehat{V}_{h+1}(s_{h+1}^k)\right)\cdot\nabla_{\theta\theta}^2 f(\xi,\phi_{h,k})\right\}$$

$$+ \sum_{k=1}^K \frac{\partial}{\partial\theta}\left(\nabla_\theta f(\xi,\phi_{h,k})\nabla_\theta^\top f(\xi,\phi_{h,k}) + \lambda I_d\right)$$

(16)

Therefore, we can bound $\kappa_{z_2}$ with $\kappa_{z_2} \le (H\kappa_3 + 3\kappa_1\kappa_2)K$ and this implies with probability $1-\delta/2$,

$$\left\|R_K(\theta_{\mathbb{T}\widehat{V}_{h+1}})\right\|_2 \le \frac{1}{2}\kappa_{z_2}\left\|\theta_{\mathbb{T}\widehat{V}_{h+1}} - \widehat{\theta}_h\right\|_2^2 \le \frac{1}{2}(H\kappa_3 + 3\kappa_1\kappa_2)K\cdot\left\|\theta_{\mathbb{T}\widehat{V}_{h+1}} - \widehat{\theta}_h\right\|_2^2$$

$$\le \frac{1}{2}(H\kappa_3 + 3\kappa_1\kappa_2)K\cdot\frac{36H^2(\log(H/\delta) + C_{d,\log K}) + 2\lambda C_\Theta^2}{\kappa K}$$

$$\le \widetilde{O}((H\kappa_3 + 3\kappa_1\kappa_2)H^2 d^2/\kappa).$$

And by Corollary E.3 with probability $1-\delta/2$,

$$\left\|\Delta_{\Sigma_h^s}(\widehat{\theta}_h - \theta_{\mathbb{T}\widehat{V}_{h+1}})\right\|_2 \le \widetilde{O}(1),$$

Therefore, by Lemma L.5 and a union bound with probability $1-\delta$,

$$|\nabla f(\widehat{\theta}_h,\phi(s,a))^\top\Sigma_h^{-1}\widetilde{R}_K(\theta_{\mathbb{T}\widehat{V}_{h+1}})| = \left|\nabla f(\widehat{\theta}_h,\phi(s,a))^\top\Sigma_h^{-1}\left(\Delta_{\Sigma_h^s}(\widehat{\theta}_h - \theta_{\mathbb{T}\widehat{V}_{h+1}}) + R_K(\theta_{\mathbb{T}\widehat{V}_{h+1}})\right)\right|$$

$$\le \left\|\nabla f(\widehat{\theta}_h,\phi(s,a))\right\|_{\Sigma_h^{-1}}\left\|\Delta_{\Sigma_h^s}(\widehat{\theta}_h - \theta_{\mathbb{T}\widehat{V}_{h+1}}) + R_K(\theta_{\mathbb{T}\widehat{V}_{h+1}})\right\|_{\Sigma_h^{-1}}$$

$$\le \left(\frac{2\kappa_1}{\sqrt{\kappa K}} + O(\frac{1}{K})\right)\left\|\Delta_{\Sigma_h^s}(\widehat{\theta}_h - \theta_{\mathbb{T}\widehat{V}_{h+1}}) + R_K(\theta_{\mathbb{T}\widehat{V}_{h+1}})\right\|_{\Sigma_h^{-1}}$$

$$\le \left(\frac{2\kappa_1}{\sqrt{\kappa K}} + O(\frac{1}{K})\right)\left(\frac{C}{\sqrt{K}} + O(\frac{1}{K})\right) = \widetilde{O}\left(\frac{\kappa_2\max(\frac{\kappa_1}{\kappa},\frac{1}{\sqrt{\kappa}})d^2 H^2 + \frac{d^2 H^3\kappa_3}{\kappa}}{K}\right)$$

where $\widetilde{O}$ absorbs all the constants and Polylog terms. Here the last inequality uses bound for $\left\|R_K(\theta_{\mathbb{T}\widehat{V}_{h+1}})\right\|_2$ and $\left\|\Delta_{\Sigma_h^s}(\widehat{\theta}_h - \theta_{\mathbb{T}\widehat{V}_{h+1}})\right\|_2$.

**Step2:** By Lemma L.5, with probability $1-\delta$,

$$\left|\nabla f(\widehat{\theta}_h,\phi(s,a))\Sigma_h^{-1}\lambda\theta_{\mathbb{T}\widehat{V}_{h+1}}\right| \le \lambda\left\|\nabla f(\widehat{\theta}_h,\phi(s,a))\right\|_{\Sigma_h^{-1}}\left\|\theta_{\mathbb{T}\widehat{V}_{h+1}}\right\|_{\Sigma_h^{-1}}$$

$$\le \lambda\left(\frac{2\kappa_1}{\sqrt{\kappa K}} + O(\frac{1}{K})\right)\cdot\left(\frac{2C_\Theta}{\sqrt{\kappa K}} + O(\frac{1}{K})\right) = \frac{4\lambda\kappa_1 C_\Theta}{\kappa K} + O(\frac{1}{K^{\frac{3}{2}}})$$

□

## F    PROOF OF THEOREM 3.2

Now we are ready to prove Theorem 3.2. In particular, we prove the first part. Also, recall that we consider the exact Bellman completeness ($\epsilon_\mathcal{F} = 0$).

### F.1    THE FIRST PART

*Proof of Theorem 3.2 (first part).* First of all, from the previous calculation (3), (8), we have

$$\left|\mathcal{P}_h\widehat{V}_{h+1}(s,a) - \widehat{\mathcal{P}}_h\widehat{V}_{h+1}(s,a)\right| \le \left|\nabla f(\widehat{\theta}_h,\phi(s,a))\left(\theta_{\mathbb{T}\widehat{V}_{h+1}} - \widehat{\theta}_h\right)\right| + |\text{Hot}_{h,1}|$$

$$\le |I_1| + |I_2| + |I_3| + |\text{Hot}_{h,2}| + |\text{Hot}_{h,1}|$$

Now by Lemma E.5, Lemma E.6, Lemma E.7, Lemma E.8 and Lemma E.1 (and a union bound), with probability $1 - \delta$,

$$|I_3| \leq \widetilde{O}(\sqrt{\frac{d^3 H^2 \kappa_2^2 \kappa_1^2}{\kappa^3}}) \frac{1}{K},$$

$$|I_2| \leq \widetilde{O}(\frac{\kappa_1^2 H^2 d^2}{\kappa K}) + \widetilde{O}(\frac{1}{K^{3/2}}),$$

$$|I_1| \leq 4Hd \left\| \nabla f(\widehat{\theta}_h, \phi(s, a)) \right\|_{\Sigma_h^{-1}} \cdot C_{\delta, \log K} + \widetilde{O}(\frac{\kappa_1}{\sqrt{\kappa} K}),$$

$$|\text{Hot}_{2,h}| \leq \widetilde{O} \left( \frac{\kappa_2 \max(\frac{\kappa_1}{\kappa}, \frac{1}{\sqrt{\kappa}}) d^2 H^2 + \frac{d^2 H^3 \kappa_3 + \lambda \kappa_1 C_\Theta}{\kappa}}{K} \right),$$

$$|\text{Hot}_{1,h}| \leq \widetilde{O}(\frac{H^2 \kappa_2 d^2}{\kappa}) \frac{1}{K}.$$

Finally, Plug the above into Lemma D.3, by a union bound over all $h \in [H]$, we have with probability $1 - \delta$, for any policy $\pi$,

$$v^\pi - v^{\widehat{\pi}} \leq \sum_{h=1}^{H} 2 \cdot \mathbb{E}_\pi \left[ |I_1| + |I_2| + |I_3| + |\text{Hot}_{h,2}| + |\text{Hot}_{h,1}| \right]$$

$$\leq \sum_{h=1}^{H} 8dH \mathbb{E}_\pi \left[ \sqrt{\nabla^\top f(\widehat{\theta}_h, \phi(s_h, a_h)) \Sigma_h^{-1} \nabla f(\widehat{\theta}_h, \phi(s_h, a_h))} \right] \cdot \iota + \widetilde{O}(\frac{C_{\text{hot}}}{K}).$$

where $\iota = C_{\delta, \log K}$ only contains Polylog terms and

$$C_{\text{hot}} = \frac{\kappa_1 H}{\sqrt{\kappa}} + \frac{\kappa_1^2 H^3 d^2}{\kappa} + \sqrt{\frac{d^3 H^4 \kappa_2^2 \kappa_1^2}{\kappa^3}} + \kappa_2 \max(\frac{\kappa_1}{\kappa}, \frac{1}{\sqrt{\kappa}}) d^2 H^3 + \frac{d^2 H^4 \kappa_3 + \lambda \kappa_1 C_\Theta}{\kappa} + \frac{H^3 \kappa_2 d^2}{\kappa}$$

$$\square$$

## F.2 THE SECOND PART

Next we prove the second part of Theorem 3.2.

*Proof of Theorem 3.2 (second part).* **Step1.** Choose $\pi = \pi^\star$ in the first part, we have

$$0 \leq v^{\pi^\star} - v^{\widehat{\pi}} \leq \sum_{h=1}^{H} 8dH \cdot \mathbb{E}_{\pi^\star} \left[ \sqrt{\nabla_\theta^\top f(\widehat{\theta}_h, \phi(s_h, a_h)) \Sigma_h^{-1} \nabla_\theta f(\widehat{\theta}_h, \phi(s_h, a_h))} \right] \cdot \iota + \widetilde{O}(\frac{C_{\text{hot}}}{K}),$$

Next, by the triangular inequality of the norm to obtain

$$\left| \left\| \nabla_\theta f(\widehat{\theta}_h, \phi(s_h, a_h)) \right\|_{\Sigma_h^{-1}} - \left\| \nabla_\theta f(\theta_h^\star, \phi(s_h, a_h)) \right\|_{\Sigma_h^{-1}} \right|$$

$$\leq \left\| \nabla_\theta f(\widehat{\theta}_h, \phi(s_h, a_h)) - \nabla_\theta f(\theta_h^\star, \phi(s_h, a_h)) \right\|_{\Sigma_h^{-1}}$$

$$= \left\| \nabla_{\theta\theta}^2 f(\xi, \phi(s_h, a_h)) \cdot \left( \widehat{\theta}_h - \theta_h^\star \right) \right\|_{\Sigma_h^{-1}},$$

since with probability $1 - \delta$,

$$\left\| \nabla_{\theta\theta}^2 f(\xi, \phi(s_h, a_h)) \cdot \left( \widehat{\theta}_h - \theta_h^\star \right) \right\|_2 \leq \kappa_2 \left\| \widehat{\theta}_h - \theta_h^\star \right\|_2 \leq \widetilde{O} \left( \frac{\kappa_1 \kappa_2 H^2 d}{\kappa} \sqrt{\frac{1}{K}} \right),$$

where the last inequality uses part three of Theorem G.3. Then by a union bound and Lemma L.5,

$$\left\| \nabla_{\theta\theta}^2 f(\xi, \phi(s_h, a_h)) \cdot \left( \widehat{\theta}_h - \theta_h^\star \right) \right\|_{\Sigma_h^{-1}} \leq \widetilde{O} \left( \frac{\kappa_1 \kappa_2 H^2 d}{\kappa^{3/2}} \cdot \frac{1}{K} \right).$$

**Step2.** Next, we show with probability $1 - \delta$,

$$\|\nabla_\theta f(\theta_h^\star, \phi(s_h, a_h))\|_{\Sigma_h^{-1}} \le 2 \|\nabla_\theta f(\theta_h^\star, \phi(s_h, a_h))\|_{\Sigma_h^{\star-1}}.$$

First of all,

$$\left\| \frac{1}{K}\Sigma_h - \frac{1}{K}\Sigma_h^\star \right\|_2 = \left\| \frac{1}{K} \left( \sum_{k=1}^K \nabla f(\widehat{\theta}_h, \phi(s,a))\nabla f(\widehat{\theta}_h, \phi(s,a))^\top - \nabla f(\theta_h^\star, \phi(s,a))\nabla f(\theta_h^\star, \phi(s,a))^\top \right) \right\|_2$$

$$\le \sup_{s,a} \left( \left\| \left( \nabla f(\widehat{\theta}_h, \phi(s,a)) - \nabla f(\theta_h^\star, \phi(s,a)) \right) \nabla f(\widehat{\theta}_h, \phi(s,a)) \right\|_2 \right.$$

$$\left. + \left\| \left( \nabla f(\widehat{\theta}_h, \phi(s,a)) - \nabla f(\theta_h^\star, \phi(s,a)) \right) \nabla f(\widehat{\theta}_h, \phi(s,a)) \right\|_2 \right)$$

$$\le 2\kappa_2\kappa_1 \left\| \widehat{\theta}_h - \theta_h^\star \right\|_2 \le \widetilde{O}\left( \frac{\kappa_2\kappa_1^2 H^2 d}{\kappa} \sqrt{\frac{1}{K}} \right)$$

Second, by Lemma L.6 with probability $1 - \delta$

$$\left\| \frac{\Sigma_h^\star}{K} - \mathbb{E}_\mu[\nabla_\theta f(\theta_h^\star, \phi)\nabla_\theta f(\theta_h^\star, \phi)^\top] - \frac{\lambda}{K} \right\| \le \frac{4\sqrt{2}\kappa_1^2}{\sqrt{K}} \left( \log \frac{2d}{\delta} \right)^{1/2}$$

This implies

$$\left\| \frac{\Sigma_h^\star}{K} \right\| \le \left\| \mathbb{E}_\mu[\nabla_\theta f(\theta_h^\star, \phi)\nabla_\theta f(\theta_h^\star, \phi)^\top] \right\| + \frac{\lambda}{K} + \frac{4\sqrt{2}\kappa_1^2}{\sqrt{K}} \left( \log \frac{2d}{\delta} \right)^{1/2}$$

$$\le \kappa_1^2 + \lambda + 4\sqrt{2}\kappa_1^2 \left( \log \frac{2d}{\delta} \right)^{1/2}$$

and also by *Weyl's spectrum theorem* and under the condition $K \ge \frac{128\kappa_1^4 \log(2d/\delta)}{\kappa^2}$, with probability $1 - \delta$

$$\lambda_{\min}(\frac{\Sigma_h^\star}{K}) \ge \lambda_{\min}\left( \mathbb{E}_\mu[\nabla_\theta f(\theta_h^\star, \phi)\nabla_\theta f(\theta_h^\star, \phi)^\top] \right) + \frac{\lambda}{K} - \frac{4\sqrt{2}\kappa_1^2}{\sqrt{K}} \left( \log \frac{2d}{\delta} \right)^{1/2}$$

$$\ge \kappa + \frac{\lambda}{K} - \frac{4\sqrt{2}\kappa_1^2}{\sqrt{K}} \left( \log \frac{2d}{\delta} \right)^{1/2} \ge \frac{\kappa}{2}$$

then $\left\| (\frac{\Sigma_h^\star}{K})^{-1} \right\| \le \frac{2}{\kappa}$. Similarly, with probability $1 - \delta$, $\left\| (\frac{\Sigma_h}{K})^{-1} \right\| \le \frac{2}{\kappa}$. Then by Lemma L.7,

$$\|\nabla_\theta f(\theta_h^\star, \phi(s,a))\|_{K\Sigma_h^{-1}} \le \left[ 1 + \sqrt{\|K\Sigma_h^{\star-1}\| \|\Sigma_h^\star/K\| \cdot \|K\Sigma_h^{-1}\| \cdot \|\Sigma_h/K - \Sigma_h^\star/K\|} \right] \cdot \|\nabla_\theta f(\theta_h^\star, \phi(s,a))\|_{K\Sigma_h^{\star-1}}$$

$$\le \left[ 1 + \sqrt{\frac{4}{\kappa^2}O(\kappa_1^2 + \lambda)\widetilde{O}\left( \frac{\kappa_2\kappa_1^2 H^2 d}{\kappa} \sqrt{\frac{1}{K}} \right)} \right] \cdot \|\nabla_\theta f(\theta_h^\star, \phi(s,a))\|_{K\Sigma_h^{\star-1}}$$

$$\le 2 \|\nabla_\theta f(\theta_h^\star, \phi(s,a))\|_{K\Sigma_h^{\star-1}}$$

as long as $K \ge \widetilde{O}(\frac{(\kappa_1^2+\lambda)^2\kappa_2^2\kappa_1^2 H^4 d^2}{\kappa^6})$. The above is equivalently to

$$\|\nabla_\theta f(\theta_h^\star, \phi(s_h, a_h))\|_{\Sigma_h^{-1}} \le 2 \|\nabla_\theta f(\theta_h^\star, \phi(s_h, a_h))\|_{\Sigma_h^{\star-1}}.$$

Combining Step1, Step2 and a union bound, we have with probability $1 - \delta$,

$$
\begin{aligned}
0 \leq v^{\pi^\star} - v^{\widehat{\pi}} &\leq \sum_{h=1}^{H} 8dH \cdot \mathbb{E}_{\pi^\star} \left[ \sqrt{\nabla_\theta^\top f(\widehat{\theta}_h, \phi(s_h, a_h)) \Sigma_h^{-1} \nabla_\theta f(\widehat{\theta}_h, \phi(s_h, a_h))} \right] \cdot \iota + \widetilde{O}(\frac{C_{\text{hot}}}{K}) \\
&\leq \sum_{h=1}^{H} 8dH \cdot \mathbb{E}_{\pi^\star} \left[ \sqrt{\nabla_\theta^\top f(\theta_h^\star, \phi(s_h, a_h)) \Sigma_h^{-1} \nabla_\theta f(\theta_h^\star, \phi(s_h, a_h))} \right] \cdot \iota + \widetilde{O}(\frac{C_{\text{hot}}}{K}) + \widetilde{O}\left( \frac{\kappa_1 \kappa_2 H^4 d^2}{\kappa^{3/2}} \cdot \frac{1}{K} \right) \\
&\leq \sum_{h=1}^{H} 16dH \cdot \mathbb{E}_{\pi^\star} \left[ \sqrt{\nabla_\theta^\top f(\theta_h^\star, \phi(s_h, a_h)) \Sigma_h^{\star -1} \nabla_\theta f(\theta_h^\star, \phi(s_h, a_h))} \right] \cdot \iota + \widetilde{O}(\frac{C'_{\text{hot}}}{K})
\end{aligned}
$$

where $C'_{\text{hot}} = C_{\text{hot}} + \frac{\kappa_1 \kappa_2 H^4 d^2}{\kappa^{3/2}}$.

$\square$

## G  PROVABLE EFFICIENCY BY REDUCTION TO GENERAL FUNCTION APPROXIMATION

In this section, we bound the accuracy of the parameter difference $\left\| \widehat{\theta}_h - \theta_{\mathbb{T}\widehat{V}_{h+1}} \right\|_2$ via a reduction to General Function Approximation scheme in Chen and Jiang (2019).

Recall the objective

$$
\ell_h(\theta) := \frac{1}{K} \sum_{k=1}^{K} \left[ f\left( \theta, \phi(s_h^k, a_h^k) \right) - r(s_h^k, a_h^k) - \widehat{V}_{h+1}\left( s_{h+1}^k \right) \right]^2 + \frac{\lambda}{K} \cdot \|\theta\|_2^2 \tag{17}
$$

Then by definition, $\widehat{\theta}_h := \arg\min_{\theta \in \Theta} \ell_h(\theta)$ and $\theta_{\mathbb{T}\widehat{V}_{h+1}}$ satisfies $f(\theta_{\mathbb{T}\widehat{V}_{h+1}}, \phi) = \mathcal{P}_h \widehat{V}_{h+1} + \delta_{\widehat{V}_{h+1}}$. Therefore, in this case, we have the following lemma:

**Lemma G.1.** *Fix $h \in [H]$. With probability $1 - \delta$,*

$$
\mathbb{E}_\mu[\ell_h(\widehat{\theta}_h)] - \mathbb{E}_\mu[\ell_h(\theta_{\mathbb{T}\widehat{V}_{h+1}})] \leq \frac{36H^2(\log(1/\delta) + C_{d,\log K}) + \lambda C_\Theta^2}{K} + \sqrt{\frac{16H^3 \epsilon_\mathcal{F}(\log(1/\delta) + C_{d,\log K})}{K}} + 4H\epsilon_\mathcal{F}.
$$

*where the expectation over $\mu$ is taken w.r.t. $(s_h^k, a_h^k, s_{h+1}^k)$ $k = 1, ..., K$ only (i.e., first compute $\mathbb{E}_\mu[\ell_h(\theta)]$ for a fixed $\theta$, then plug-in either $\widehat{\theta}_{h+1}$ or $\theta_{\mathbb{T}\widehat{V}_{h+1}}$). Here $C_{d,\log(K)} := d\log(1 + 24C_\Theta(H+1)\kappa_1 K) + d\log\left( 1 + 288H^2 C_\Theta(\kappa_1\sqrt{C_\Theta} + 2\sqrt{\kappa_1\kappa_2/\lambda})^2 K^2 \right) + d^2 \log\left( 1 + 288H^2\sqrt{d}\kappa_1^2 K^2/\lambda \right).$*

*Proof of Lemma G.1.* **Step1:** we first prove the case where $\lambda = 0$.

Indeed, fix $h \in [H]$ and any function $V(\cdot) \in \mathbb{R}^\mathcal{S}$. Similarly, define $f_V(s,a) := f(\theta_{\mathbb{T}V}, \phi) = \mathcal{P}_h V + \delta_V$. For any fixed $\theta \in \Theta$, denote $g(s,a) = f(\theta, \phi(s,a))$. Then define[10]

$$
X(g, V, f_V) := (g(s,a) - r - V(s'))^2 - (f_V(s,a) - r - V(s'))^2.
$$

Since all episodes are independent of each other, $X_k(g, V, f_V) := X(g(s_h^k, a_h^k), V(s_{h+1}^k), f_V(s_h^k, a_h^k))$ are independent r.v.s and it holds

$$
\frac{1}{K} \sum_{k=1}^{K} X_k(g, V, f_V) = \ell(g) - \ell(f_V). \tag{18}
$$

---

[10]We abuse the notation here to use either $X(g, V, f_V)$ or $X(\theta, V, f_V)$. They mean the same quantity.

Next, the variance of $X$ is bounded by:

$$\text{Var}[X(g, V, f_V)] \leq \mathbb{E}_\mu[X(g, f, f_V)^2]$$

$$=\mathbb{E}_\mu\left[\left((g(s_h, a_h) - r_h - V(s_{h+1}))^2 - (f_V(s_h, a_h) - r_h - V(s_{h+1}))^2\right)^2\right]$$

$$=\mathbb{E}_\mu\left[(g(s_h, a_h) - f_V(s_h, a_h))^2(g(s_h, a_h) + f_V(s_h, a_h) - 2r_h - 2V(s_{h+1}))^2\right]$$

$$\leq 4H^2 \cdot \mathbb{E}_\mu[(g(s_h, a_h) - f_V(s_h, a_h))^2]$$

$$\leq 4H^2 \cdot \mathbb{E}_\mu\left[(g(s_h, a_h) - r_h - V(s_{h+1}))^2 - (f_V(s_h, a_h) - r_h - V(s_{h+1}))^2\right] + 8H^3\epsilon_\mathcal{F} \quad (*)$$

$$=4H^2 \cdot \mathbb{E}_\mu[X(g, f, f_V)] + 8H^3\epsilon_\mathcal{F}$$

where the step $(*)$ comes from

$$\mathbb{E}_\mu\left[(g(s_h, a_h) - r_h - V(s_{h+1}))^2 - (f_V(s_h, a_h) - r_h - V(s_{h+1}))^2\right]$$

$$=\mathbb{E}_\mu\left[(g(s_h, a_h) - f_V(s_h, a_h)) \cdot (g(s_h, a_h) + f_V(s_h, a_h) - 2r_h - 2V(s_{h+1}))\right]$$

$$=\mathbb{E}_\mu\left[(g(s_h, a_h) - f_V(s_h, a_h)) \cdot (g(s_h, a_h) - f_V(s_h, a_h) + 2f_V(s_h, a_h) - 2r_h - 2V(s_{h+1}))\right]$$

$$=\mathbb{E}_\mu\left[(g(s_h, a_h) - f_V(s_h, a_h))^2\right] + \mathbb{E}_\mu\left[2(g(s_h, a_h) - f_V(s_h, a_h))\mathbb{E}_{P_h}[f_V(s_h, a_h) - r_h - V(s_{h+1}) \mid s_h, a_h]\right]$$

$$\geq \mathbb{E}_\mu\left[(g(s_h, a_h) - f_V(s_h, a_h))^2\right] - 2H\|\delta_V\|_\infty \geq \mathbb{E}_\mu\left[(g(s_h, a_h) - f_V(s_h, a_h))^2\right] - 2H\epsilon_\mathcal{F}$$

$$(19)$$

where the last step uses law of total expectation and the definition of $f_V$.

Therefore, by Bernstein inequality, with probability $1 - \delta$,

$$\mathbb{E}_\mu[X(g, f, f_V)] - \frac{1}{K}\sum_{k=1}^K X_k(g, f, f_V)$$

$$\leq \sqrt{\frac{2\text{Var}[X(g, f, f_V)]\log(1/\delta)}{K}} + \frac{4H^2\log(1/\delta)}{3K}$$

$$\leq \sqrt{\frac{8H^2\mathbb{E}_\mu[X(g, f, f_V)]\log(1/\delta)}{K}} + \sqrt{\frac{16H^3\epsilon_\mathcal{F}\log(1/\delta)}{K}} + \frac{4H^2\log(1/\delta)}{3K}.$$

Now, if we choose $g(s, a) := f(\widehat{\theta}_h, \phi(s, a))$, then $\widehat{\theta}_h$ minimizes $\ell_h(\theta)$, therefore, it also minimizes $\frac{1}{K}\sum_{k=1}^K X_i(\theta, \widehat{V}_{h+1}, f_{\widehat{V}_{h+1}})$ and this implies

$$\frac{1}{K}\sum_{k=1}^K X_k(\widehat{\theta}_h, \widehat{V}_{h+1}, f_{\widehat{V}_{h+1}}) \leq \frac{1}{K}\sum_{k=1}^K X_k(\theta_{\mathbb{T}\widehat{V}_{h+1}}, \widehat{V}_{h+1}, f_{\widehat{V}_{h+1}}) = 0.$$

Therefore, we obtain

$$\mathbb{E}_\mu[X(\widehat{\theta}_h, \widehat{V}_{h+1}, f_{\widehat{V}_{h+1}})] \leq \sqrt{\frac{8H^2 \cdot \mathbb{E}_\mu[X(\widehat{\theta}_h, \widehat{V}_{h+1}, f_{\widehat{V}_{h+1}})]\log(1/\delta)}{K}} + \sqrt{\frac{16H^3\epsilon_\mathcal{F}\log(1/\delta)}{K}} + \frac{4H^2\log(1/\delta)}{3K}.$$

However, the above does not hold with probability $1 - \delta$ since $\widehat{\theta}_h$ and $\widehat{V}_{h+1} :=$ $\min\{\max_a f(\widehat{\theta}_{h+1}, \phi(\cdot, a)) - \sqrt{\nabla f(\widehat{\theta}_{h+1}, \phi(\cdot, a))^\top A \cdot \nabla f(\theta, \phi(\cdot, a))}, H\}$ (where $A$ is certain symmetric matrix with bounded norm) depend on $\widehat{\theta}_h$ and $\widehat{\theta}_{h+1}$ which are data-dependent. Therefore, we need to further apply covering Lemma L.10 and choose $\epsilon = O(1/K)$ and a union bound to obtain with probability $1 - \delta$,

$$\mathbb{E}_\mu[X(\widehat{\theta}_h, \widehat{V}_{h+1}, f_{\widehat{V}_{h+1}})] \leq \sqrt{\frac{8H^2 \cdot \mathbb{E}_\mu[X(\widehat{\theta}_h, \widehat{V}_{h+1}, f_{\widehat{V}_{h+1}})](\log(1/\delta) + C_{d,\log K})}{K}} + \frac{7H^2(\log(1/\delta) + C_{d,\log K})}{3K}$$

$$+ \sqrt{\frac{16H^3\epsilon_\mathcal{F}(\log(1/\delta) + C_{d,\log K})}{K}} + 4H\epsilon_\mathcal{F}$$

where $C_{d,\log(K)} := \log(1+24C_\Theta(H+1)\kappa_1 K)+d\log\left(1+288H^2C_\Theta(\kappa_1\sqrt{C_\Theta}+2\sqrt{\kappa_1\kappa_2/\lambda})^2K^2\right)+$
$d^2\log\left(1+288H^2\sqrt{d}\kappa_1^2K^2/\lambda\right)$.[11] Solving this quadratic equation to obtain with probability $1-\delta$,

$$\mathbb{E}_\mu[X(\widehat{\theta}_h,\widehat{V}_{h+1},f_{\widehat{V}_{h+1}})] \le \frac{36H^2(\log(1/\delta)+C_{d,\log K})}{K}+\sqrt{\frac{16H^3\epsilon_\mathcal{F}(\log(1/\delta)+C_{d,\log K})}{K}}+4H\epsilon_\mathcal{F}$$

Now according to equation 18, by definition we finally have with probability $1-\delta$ (recall the expectation over $\mu$ is taken w.r.t. $(s_h^k, a_h^k, s_{h+1}^k)$ $k = 1, ..., K$ only)

$$\begin{aligned}\mathbb{E}_\mu[\ell_h(\widehat{\theta}_{h+1})] - \mathbb{E}_\mu[\ell_h(\theta_{\mathbb{T}\widehat{V}_{h+1}})] &= \mathbb{E}_\mu[X(\widehat{\theta}_h,\widehat{V}_{h+1},f_{\widehat{V}_{h+1}})]\\ &\le \frac{36H^2(\log(1/\delta)+C_{d,\log K})}{K} + \sqrt{\frac{16H^3\epsilon_\mathcal{F}(\log(1/\delta)+C_{d,\log K})}{K}} + 4H\epsilon_\mathcal{F}.\end{aligned} \quad (20)$$

**Step2.** If $\lambda > 0$, there is only extra term $\frac{\lambda}{K}\left(\left\|\widehat{\theta}_h\right\|_2 - \left\|\theta_{\mathbb{T}\widehat{V}_{h+1}}\right\|_2\right) \le \frac{\lambda}{K}\left\|\widehat{\theta}_h\right\|_2 \le \frac{\lambda C_\Theta^2}{K}$ in addition to above. This finishes the proof.

$\square$

**Theorem G.2** (Provable efficiency (Part I)). *Let $C_{d,\log K}$ be the same as Lemma G.1. Then denote*
$b_{d,K,\epsilon_\mathcal{F}} := \sqrt{\frac{16H^3\epsilon_\mathcal{F}(\log(1/\delta)+C_{d,\log K})}{K}} + 4H\epsilon_\mathcal{F}$, *with probability $1-\delta$*

$$\left\|\widehat{\theta}_h - \theta_{\mathbb{T}\widehat{V}_{h+1}}\right\|_2 \le \sqrt{\frac{36H^2(\log(H/\delta)+C_{d,\log K})+2\lambda C_\Theta^2}{\kappa K}} + \sqrt{\frac{b_{d,K,\epsilon_\mathcal{F}}}{\kappa}} + \sqrt{\frac{2H\epsilon_\mathcal{F}}{\kappa}}, \ \forall h \in [H].$$

*Proof of Theorem G.2.* Apply a union bound in Lemma G.1, we have with probability $1-\delta$,

$$\mathbb{E}_\mu[\ell_h(\widehat{\theta}_h)] - \mathbb{E}_\mu[\ell_h(\theta_{\mathbb{T}\widehat{V}_{h+1}})] \le \frac{36H^2(\log(H/\delta)+C_{d,\log K})+\lambda C_\Theta^2}{K} + b_{d,K,\epsilon_\mathcal{F}}, \ \forall h \in [H]$$

$$\Rightarrow \mathbb{E}_\mu[\ell_h(\widehat{\theta}_h) - \frac{\lambda}{K}\left\|\widehat{\theta}_h\right\|_2^2] - \mathbb{E}_\mu[\ell_h(\theta_{\mathbb{T}\widehat{V}_{h+1}}) - \frac{\lambda}{K}\left\|\theta_{\mathbb{T}\widehat{V}_{h+1}}\right\|_2^2] \le \frac{36H^2(\log(H/\delta)+C_{d,\log K})+2\lambda C_\Theta^2}{K} + b_{d,K,\epsilon_\mathcal{F}} \quad (21)$$

Now we prove for all $h \in [H]$,

$$\mathbb{E}_\mu\left[\left(f(\widehat{\theta}_h,\phi(\cdot,\cdot)) - f(\theta_{\mathbb{T}\widehat{V}_{h+1}},\phi(\cdot,\cdot))\right)^2\right] \le \mathbb{E}_\mu\left[\ell_h(\widehat{\theta}_h) - \frac{\lambda\left\|\widehat{\theta}_h\right\|_2^2}{K}\right] - \mathbb{E}_\mu\left[\ell_h(\theta_{\mathbb{T}\widehat{V}_{h+1}}) - \frac{\lambda\left\|\theta_{\mathbb{T}\widehat{V}_{h+1}}\right\|_2^2}{K}\right] + 2H\epsilon_\mathcal{F}. \quad (22)$$

Indeed, similar to equation 20, by definition we have

$$\mathbb{E}_\mu\left[\ell_h(\widehat{\theta}_h) - \frac{\lambda\left\|\widehat{\theta}_h\right\|_2^2}{K}\right] - \mathbb{E}_\mu\left[\ell_h(\theta_{\mathbb{T}\widehat{V}_{h+1}}) - \frac{\lambda\left\|\theta_{\mathbb{T}\widehat{V}_{h+1}}\right\|_2^2}{K}\right] = \mathbb{E}_\mu[X(\widehat{\theta}_h,\widehat{V}_{h+1},f_{\widehat{V}_{h+1}})]$$

$$=\mathbb{E}_\mu\left(\left[f\left(\widehat{\theta}_h,\phi(s_h,a_h)\right) - r_h - \widehat{V}_{h+1}(s_{h+1})\right]^2 - \left[f\left(\theta_{\mathbb{T}\widehat{V}_{h+1}},\phi(s_h,a_h)\right) - r_h - \widehat{V}_{h+1}(s_{h+1})\right]^2\right)$$

$$=\mathbb{E}_\mu\left[\left(f(\widehat{\theta}_h,\phi(\cdot,\cdot)) - f(\theta_{\mathbb{T}\widehat{V}_{h+1}},\phi(\cdot,\cdot))\right)^2\right]$$
$$+\mathbb{E}_\mu\left[\left(f(\widehat{\theta}_h,\phi(s_h,a_h)) - f(\theta_{\mathbb{T}\widehat{V}_{h+1}},\phi(s_h,a_h))\right) \cdot \left(f\left(\theta_{\mathbb{T}\widehat{V}_{h+1}},\phi(s_h,a_h)\right) - r_h - \widehat{V}_{h+1}(s_{h+1})\right)\right]$$

$$=\mathbb{E}_\mu\left[\left(f(\widehat{\theta}_h,\phi(\cdot,\cdot)) - f(\theta_{\mathbb{T}\widehat{V}_{h+1}},\phi(\cdot,\cdot))\right)^2\right]$$
$$+\mathbb{E}_\mu\left[\left(f(\widehat{\theta}_h,\phi(s_h,a_h)) - f(\theta_{\mathbb{T}\widehat{V}_{h+1}},\phi(s_h,a_h))\right) \cdot \mathbb{E}\left(f\left(\theta_{\mathbb{T}\widehat{V}_{h+1}},\phi(s_h,a_h)\right) - r_h - \widehat{V}_{h+1}(s_{h+1})\Big|s_h,a_h\right)\right]$$

$$\ge\mathbb{E}_\mu\left[\left(f(\widehat{\theta}_h,\phi(\cdot,\cdot)) - f(\theta_{\mathbb{T}\widehat{V}_{h+1}},\phi(\cdot,\cdot))\right)^2\right] - 2H\epsilon_\mathcal{F}$$

---

[11]Here in our realization of Lemma L.9, we set $B = 1/\lambda$ (since $\left\|\Sigma_h^{-1}\right\|_2 \le 1/\lambda$).

where the third identity uses $\mu$ is taken w.r.t. $s_h, a_h, s_{h+1}$ (recall Lemma G.1) and law of total expectation. The first inequality uses the definition of $\theta_{\mathbb{T}\widehat{V}_{h+1}}$.

Now apply Assumption 2.3, we have

$$\mathbb{E}_\mu \left[ \left( f(\widehat{\theta}_h, \phi(\cdot, \cdot)) - f(\theta_{\mathbb{T}\widehat{V}_{h+1}}, \phi(\cdot, \cdot)) \right)^2 \right] \geq \kappa \left\| \widehat{\theta}_h - \theta_{\mathbb{T}\widehat{V}_{h+1}} \right\|_2^2,$$

Combine the above with equation 21 and equation 22, we obtain the stated result. $\qquad\square$

**Theorem G.3** (Provable efficiency (Part II)). *Let $C_{d,\log K}$ be the same as Lemma G.1 and suppose $\epsilon_\mathcal{F} = 0$. Furthermore, suppose $\lambda \leq 1/2C_\Theta^2$ and $K \geq \max\left\{ 512\frac{\kappa_1^4}{\kappa^2}\left( \log(\frac{2d}{\delta}) + d\log(1 + \frac{4\kappa_1^3\kappa_2 C_\Theta K^3}{\lambda^2}) \right), \frac{4\lambda}{\kappa} \right\}$. Then, with probability $1 - \delta$, $\forall h \in [H]$,*

$$\sup_{s,a} \left| f(\widehat{\theta}_h, \phi(s,a)) - f(\theta_h^\star, \phi(s,a)) \right| \leq \left( \kappa_1 H \sqrt{\frac{36H^2(\log(H^2/\delta) + C_{d,\log K}) + 2\lambda C_\Theta^2}{\kappa}} + \frac{2H^2 d\kappa_1}{\sqrt{\kappa}} \right) \sqrt{\frac{1}{K}} + O(\frac{1}{K}).$$

*Furthermore, we have with probability $1 - \delta$,*

$$\sup_h \left\| \widehat{V}_h - V_h^\star \right\|_\infty \leq \left( \kappa_1 H \sqrt{\frac{36H^2(\log(H^2/\delta) + C_{d,\log K}) + 2\lambda C_\Theta^2}{\kappa}} + \frac{2H^2 d\kappa_1}{\sqrt{\kappa}} \right) \sqrt{\frac{1}{K}} + O(\frac{1}{K})$$

$$= \widetilde{O} \left( \kappa_1 H^2 \sqrt{\frac{d^2}{\kappa}} \sqrt{\frac{1}{K}} \right)$$

*where $\widetilde{O}$ absorbs Polylog terms and higher order terms. Lastly, it also holds for all $h \in [H]$, w.p. $1 - \delta$*

$$\left\| \widehat{\theta}_h - \theta_h^\star \right\|_2 \leq \left( \kappa_1 H \frac{\sqrt{72H^2(\log(H^2/\delta) + C_{d,\log K}) + 4\lambda C_\Theta^2}}{\kappa} + \frac{4H^2 d\kappa_1}{\kappa} \right) \sqrt{\frac{1}{K}} + O(\frac{1}{K})$$

$$= \widetilde{O} \left( \frac{\kappa_1 H^2 d}{\kappa} \sqrt{\frac{1}{K}} \right)$$

*Proof of Theorem G.3.* **Step1:** we show the first result.

We prove this by backward induction. When $h = H + 1$, by convention $f(\widehat{\theta}_h, \phi(s,a)) = f(\theta_h^\star, \phi(s,a)) = 0$ so the base case holds. Suppose for $h + 1$, with probability $1 - (H - h)\delta$, it holds true that $\sup_{s,a} \left| f(\widehat{\theta}_{h+1}, \phi(s,a)) - f(\theta_{h+1}^\star, \phi(s,a)) \right| \leq C_{h+1}\sqrt{\frac{1}{K}} + a(h+1)$, we next consider the case for $t = h$.

On one hand, by Theorem G.2, we have with probability $1 - \delta/2$,

$$\sup_{s,a} \left| f(\widehat{\theta}_h, \phi(s,a)) - f(\theta_h^\star, \phi(s,a)) \right|$$

$$\leq \sup_{s,a} \left| f(\widehat{\theta}_h, \phi(s,a)) - f(\theta_{\mathbb{T}\widehat{V}_{h+1}}, \phi(s,a)) \right| + \sup_{s,a} \left| f(\theta_{\mathbb{T}\widehat{V}_{h+1}}, \phi(s,a)) - f(\theta_h^\star, \phi(s,a)) \right|$$

$$= \sup_{s,a} \left| \nabla f(\xi, \phi(s,a))^\top (\widehat{\theta}_h - \theta_{\mathbb{T}\widehat{V}_{h+1}}) \right| + \sup_{s,a} \left| f(\theta_{\mathbb{T}\widehat{V}_{h+1}}, \phi(s,a)) - f(\theta_{\mathbb{T}V_{h+1}^\star}, \phi(s,a)) \right|$$

$$\leq \kappa_1 \cdot \left\| \widehat{\theta}_h - \theta_{\mathbb{T}\widehat{V}_{h+1}} \right\|_2 + \sup_{s,a} \left| \mathcal{P}_{h,s,a}\widehat{V}_{h+1} - \mathcal{P}_{h,s,a}V_{h+1}^\star \right|$$

$$\leq \kappa_1 \sqrt{\frac{36H^2(\log(H/\delta) + C_{d,\log K}) + 2\lambda C_\Theta^2}{\kappa K}} + \left\| \widehat{V}_{h+1} - V_{h+1}^\star \right\|_\infty,$$

Recall $\widehat{V}_{h+1}(\cdot) := \min\{\max_a f(\widehat{\theta}_{h+1}, \phi(\cdot, a)) - \Gamma_h(\cdot, a), H\}$ and $V_{h+1}^\star(\cdot) = \max_a f(\theta_{h+1}^\star, \phi(\cdot, a)) = \min\{\max_a f(\theta_{h+1}^\star, \phi(\cdot, a)), H\}$, we obtain

$$\left\| \widehat{V}_{h+1} - V_{h+1}^\star \right\|_\infty \leq \sup_{s,a} \left| f(\widehat{\theta}_{h+1}, \phi(s,a)) - f(\theta_{h+1}^\star, \phi(s,a)) \right| + \sup_{h,s,a} \Gamma_h(s,a) \qquad (23)$$

Note the above holds true for any generic $\Gamma_h(s,a)$. In particular, according to Algorithm 1, we specify

$$\Gamma_h(\cdot,\cdot) = dH\sqrt{\nabla_\theta f(\widehat{\theta}_h, \phi(\cdot,\cdot))^\top \Sigma_h^{-1} \nabla_\theta f(\widehat{\theta}_h, \phi(\cdot,\cdot))} \left(+\widetilde{O}(\frac{1}{K})\right)$$

and by Lemma L.5, with probability $1 - \delta$,

$$\Gamma_h \leq \frac{2dH\kappa_1}{\sqrt{\kappa K}} + \widetilde{O}(\frac{1}{K})$$

and by a union bound this implies with probability $1 - (H - h + 1)\delta$,

$$\sup_{s,a}\left|f(\widehat{\theta}_h, \phi(s,a)) - f(\theta_h^\star, \phi(s,a))\right|$$

$$\leq C_{h+1}\sqrt{\frac{1}{K}} + \kappa_1\sqrt{\frac{36H^2(\log(H/\delta) + C_{d,\log K}) + 2\lambda C_\Theta^2}{\kappa K}} + \frac{2dH\kappa_1}{\sqrt{\kappa K}} + \widetilde{O}(\frac{1}{K}) := C_h\sqrt{\frac{1}{K}} + \widetilde{O}(\frac{1}{K})$$

Solving for $C_h$, we obtain $C_h \leq \kappa_1 H\sqrt{\frac{36H^2(\log(H/\delta) + C_{d,\log K}) + 2\lambda C_\Theta^2}{\kappa}} + H\frac{2dH\kappa_1}{\sqrt{\kappa}}$ for all $H$. By a union bound (replacing $\delta$ by $\delta/H$), we obtain the stated result.

**Step2:** Utilizing the intermediate result equation 23, we directly have with probability $1 - \delta$,

$$\sup_h\left\|\widehat{V}_h - V_h^\star\right\|_\infty \leq \sup_{s,a}\left|f(\widehat{\theta}_h, \phi(s,a)) - f(\theta_h^\star, \phi(s,a))\right| + \frac{2dH\kappa_1}{\sqrt{\kappa K}} + O(\frac{1}{K}),$$

where $\sup_{s,a}\left|f(\widehat{\theta}_h, \phi(s,a)) - f(\theta_h^\star, \phi(s,a))\right|$ can be bounded using Step1.

**Step3:** Denote $M := \left(\kappa_1 H\sqrt{\frac{36H^2(\log(H^2/\delta) + C_{d,\log K}) + 2\lambda C_\Theta^2}{\kappa}} + \frac{2H^2 d\kappa_1}{\sqrt{\kappa}}\right)\sqrt{\frac{1}{K}} + O(\frac{1}{K})$, then by Step1 we have with probability $1 - \delta$ (here $\xi$ is some point between $\widehat{\theta}_h$ and $\theta_h^\star$) for all $h \in [H]$

$$M^2 \geq \sup_{s,a}\left|f(\widehat{\theta}_h, \phi(s,a)) - f(\theta_h^\star, \phi(s,a))\right|^2$$

$$\geq \mathbb{E}_{\mu,h}[(f(\widehat{\theta}_h, \phi(s,a)) - f(\theta_h^\star, \phi(s,a)))^2] \geq \kappa\left\|\widehat{\theta}_h - \theta_h^\star\right\|_2^2$$

where the last inequality is by Assumption 2.3. Solve this to obtain the stated result.

$\square$

## H WITH POSITIVE BELLMAN COMPLETENESS COEFFICIENT $\epsilon_\mathcal{F} > 0$

In Theorem 3.2, we consider the case where $\epsilon_\mathcal{F} = 0$. If $\epsilon_\mathcal{F} > 0$, similar guarantee can be achieved with the measurement of model misspecification. For instance, the additional error $\sqrt{\frac{16H^3\epsilon_\mathcal{F}(\log(1/\delta) + C_{d,\log K})}{K}} + 4H\epsilon_\mathcal{F}$ will show up in Lemma G.1 (as stated in the current version), $\sqrt{\frac{b_{d,K,\epsilon_\mathcal{F}}}{\kappa}} + \sqrt{\frac{2H\epsilon_\mathcal{F}}{\kappa}}$ will show up in Lemma G.2. Then the decomposition in equation 3 will incur the extra $\delta_{\widehat{V}_{h+1}}$ term with $\delta_{\widehat{V}_{h+1}}$ might not be 0. The analysis with positive $\epsilon_\mathcal{F} > 0$ will make the proofs more intricate but incurs no additional technical challenge. Since the inclusion of this quantity is not our major focus, as a result, we only provide the proof for the case where $\epsilon_\mathcal{F} = 0$ so the readers can focus on the more critical components that characterize the hardness of differentiable function class.

## I VFQL AND ITS ANALYSIS

We present the *vanilla fitted Q-learning* (VFQL) Algorithm 2 as follows. For VFQL, no pessimism is used and we assume $\widehat{\theta}_h \in \Theta$ without loss of generality.

---

**Algorithm 2** Vanilla Fitted Q-Learning (VFQL)

---

1: **Input:** Offline Dataset $\mathcal{D} = \left\{ \left( s_h^k, a_h^k, r_h^k, s_{h+1}^k \right) \right\}_{k,h=1}^{K,H}$. Denote $\phi_{h,k} := \phi(s_h^k, a_h^k)$.

2: **Initialization:** Set $\widehat{V}_{H+1}(\cdot) \leftarrow 0$ and $\lambda > 0$.

3: **for** $h = H, H-1, \ldots, 1$ **do**

4:      Set $\widehat{\theta}_h \leftarrow \arg\min_{\theta \in \Theta} \left\{ \sum_{k=1}^K \left[ f\left(\theta, \phi_{h,k}\right) - r_{h,k} - \widehat{V}_{H+1}(s_{h+1}^k) \right]^2 + \lambda \cdot \|\theta\|_2^2 \right\}$

5:      Set $\widehat{Q}_h(\cdot,\cdot) \leftarrow \min \left\{ f(\widehat{\theta}_h, \phi(\cdot,\cdot)), H-h+1 \right\}^+$

6:      Set $\widehat{\pi}_h(\cdot \mid \cdot) \leftarrow \arg\max_{\pi_h} \left\langle \widehat{Q}_h(\cdot,\cdot), \pi_h(\cdot \mid \cdot) \right\rangle_{\mathcal{A}}$, $\widehat{V}_h(\cdot) \leftarrow \max_{\pi_h} \left\langle \widehat{Q}_h(\cdot,\cdot), \pi_h(\cdot \mid \cdot) \right\rangle_{\mathcal{A}}$

7: **end for**

8: **Output:** $\{\widehat{\pi}_h\}_{h=1}^H$.

---

## I.1 ANALYSIS FOR VFQL (THEOREM 3.1)

Recall $\iota_h(s,a) := \mathcal{P}_h \widehat{V}_{h+1}(s,a) - \widehat{Q}_h(s,a)$ and the definition of Bellman operator D.1. Note $\min\{\cdot, H-h+1\}^+$ is a non-expansive operator, therefore we have

$$|\iota_h(s,a)| = |\mathcal{P}_h \widehat{V}_{h+1}(s,a) - \widehat{Q}_h(s,a)| = \left| \min\left\{ \mathcal{P}_h \widehat{V}_{h+1}(s,a), H-h+1 \right\}^+ - \min\left\{ f(\widehat{\theta}_h, \phi(\cdot,\cdot)), H-h+1 \right\}^+ \right|$$

$$\leq \left| \mathcal{P}_h \widehat{V}_{h+1}(s,a) - f(\widehat{\theta}_h, \phi(\cdot,\cdot)) \right| \leq \left| f(\theta_{\mathbb{T}\widehat{V}_{h+1}}) - f(\widehat{\theta}_h, \phi(\cdot,\cdot)) \right| + \epsilon_{\mathcal{F}}.$$

By Lemma D.2, we have for any $\pi$,

$$v^\pi - v^{\widehat{\pi}} = -\sum_{h=1}^H E_{\widehat{\pi}}[\iota_h(s_h, a_h)] + \sum_{h=1}^H E_\pi[\iota_h(s_h, a_h)] \leq \sum_{h=1}^H E_{\widehat{\pi}}[|\iota_h(s_h, a_h)|] + \sum_{h=1}^H E_\pi[|\iota_h(s_h, a_h)|]$$

$$\leq \sum_{h=1}^H \mathbb{E}_{\widehat{\pi}}[|f(\theta_{\mathbb{T}\widehat{V}_{h+1}}, \phi(\cdot,\cdot)) - f(\widehat{\theta}_h, \phi(\cdot,\cdot))|] + \sum_{h=1}^H \mathbb{E}_\pi[|f(\theta_{\mathbb{T}\widehat{V}_{h+1}}, \phi(\cdot,\cdot)) - f(\widehat{\theta}_h, \phi(\cdot,\cdot))|] + 2H\epsilon_{\mathcal{F}}$$

$$\leq \sum_{h=1}^H \sqrt{\mathbb{E}_{\widehat{\pi}}[|f(\theta_{\mathbb{T}\widehat{V}_{h+1}}, \phi(\cdot,\cdot)) - f(\widehat{\theta}_h, \phi(\cdot,\cdot))|^2]} + \sum_{h=1}^H \sqrt{\mathbb{E}_\pi[|f(\theta_{\mathbb{T}\widehat{V}_{h+1}}, \phi(\cdot,\cdot)) - f(\widehat{\theta}_h, \phi(\cdot,\cdot))|^2]} + 2H\epsilon_{\mathcal{F}}$$

$$\leq 2\sqrt{C_{\text{eff}}} \sum_{h=1}^H \sqrt{\mathbb{E}_{\mu,h}[|f(\theta_{\mathbb{T}\widehat{V}_{h+1}}, \phi(\cdot,\cdot)) - f(\widehat{\theta}_h, \phi(\cdot,\cdot))|^2]} + 2H\epsilon_{\mathcal{F}}$$

$$\tag{24}$$

where the second inequality uses Cauchy inequality and the third one uses the definition of concentrability coefficient 2.2.

Next, for VFQL, there is no pessimism therefore the quantity $B$ in Lemma L.10 is zero, hence the covering number applied in Lemma G.1 is bounded by $C_{d,\log(K)} \leq \widetilde{O}(d)$ and

$$\mathbb{E}_\mu[\ell_h(\widehat{\theta}_h)] - \mathbb{E}_\mu[\ell_h(\theta_{\mathbb{T}\widehat{V}_{h+1}})] \leq \frac{36H^2(\log(1/\delta) + C_{d,\log K}) + \lambda C_\Theta^2}{K} + \sqrt{\frac{16H^3 \epsilon_{\mathcal{F}}(\log(1/\delta) + C_{d,\log K})}{K}} + 4H\epsilon_{\mathcal{F}}.$$

Now leveraging equation 21 and equation 22 in Theorem G.2 to obtain

$$\mathbb{E}_\mu\left[ \left( f(\widehat{\theta}_h, \phi(\cdot,\cdot)) - f(\theta_{\mathbb{T}\widehat{V}_{h+1}}, \phi(\cdot,\cdot)) \right)^2 \right] \leq \mathbb{E}_\mu\left[ \ell_h(\widehat{\theta}_h) - \frac{\lambda \left\| \widehat{\theta}_h \right\|_2^2}{K} \right] - \mathbb{E}_\mu\left[ \ell_h(\theta_{\mathbb{T}\widehat{V}_{h+1}}) - \frac{\lambda \left\| \theta_{\mathbb{T}\widehat{V}_{h+1}} \right\|_2^2}{K} \right] + 2H\epsilon_{\mathcal{F}}$$

$$\leq \frac{36H^2(\log(H/\delta) + C_{d,\log K}) + 2\lambda C_\Theta^2}{K} + b_{d,K,\epsilon_{\mathcal{F}}} + 2H\epsilon_{\mathcal{F}}$$

Plug the above into equation 24, we obtain with probability $1 - \delta$, for all policy $\pi$,

$$v^\pi - v^{\widehat{\pi}} \leq 2\sqrt{C_{\text{eff}}}H\sqrt{\frac{36H^2(\log(H/\delta) + C_{d,\log K}) + 2\lambda C_\Theta^2}{K} + b_{d,K,\epsilon_{\mathcal{F}}}} + 2H\epsilon_{\mathcal{F}} + 2H\epsilon_{\mathcal{F}}$$

$$=2\sqrt{C_{\text{eff}}}H\sqrt{\frac{36H^2(\log(H/\delta) + C_{d,\log K}) + 2\lambda C_\Theta^2}{K}} + \sqrt{\frac{16H^3\epsilon_{\mathcal{F}}(\log(1/\delta) + C_{d,\log K})}{K}} + 6H\epsilon_{\mathcal{F}} + 2H\epsilon_{\mathcal{F}}$$

$$=\sqrt{C_{\text{eff}}}H \cdot \widetilde{O}\left(\sqrt{\frac{H^2d + \lambda C_\Theta^2}{K}} + \sqrt[4]{\frac{H^3 d\epsilon_{\mathcal{F}}}{K}}\right) + O(\sqrt{C_{\text{eff}}H^3\epsilon_{\mathcal{F}}} + H\epsilon_{\mathcal{F}})$$

This finishes the proof of Theorem 3.1.

## J  PROOFS FOR VAFQL

In this section, we present the analysis for *variance-aware fitted Q learning* (VAFQL). Throughout the whole section, we assume $\epsilon_{\mathcal{F}} = 0$, *i.e.* the exact Bellman-Completeness holds. The algorithm is presented in the following. Before giving the proofs of Theorem 3, we first prove some useful lemmas.

---

**Algorithm 3** Variance-Aware Fitted Q Learning (VAFQL)

---

1: **Input:** Split dataset $\mathcal{D} = \left\{\left(s_h^k, a_h^k, r_h^k\right)\right\}_{k,h=1}^{K,H} \mathcal{D}' = \left\{\left(\bar{s}_h^k, \bar{a}_h^k, \bar{r}_h^k\right)\right\}_{k,h=1}^{K,H}$. Require $\beta$.
2: **Initialization:** Set $\widehat{V}_{H+1}(\cdot) \leftarrow 0$. Denote $\phi_{h,k} := \phi(s_h^k, a_h^k), \bar{\phi}_{h,k} := \phi(\bar{s}_h^k, \bar{a}_h^k)$
3: **for** $h = H, H-1, \ldots, 1$ **do**
4:     Set $\boldsymbol{u}_h \leftarrow \arg\min_{\theta \in \Theta} \left\{\sum_{k=1}^K \left[f\left(\theta, \bar{\phi}_{h,k}\right) - \widehat{V}_{h+1}(\bar{s}_{h+1}^k)\right]^2 + \lambda \cdot \|\theta\|_2^2\right\}$
5:     Set $\boldsymbol{v}_h \leftarrow \arg\min_{\theta \in \Theta} \left\{\sum_{k=1}^K \left[f\left(\theta, \bar{\phi}_{h,k}\right) - \widehat{V}_{h+1}^2(\bar{s}_{h+1}^k)\right]^2 + \lambda \cdot \|\theta\|_2^2\right\}$
6:     Set $\left[\widehat{\text{Var}}_h \widehat{V}_{h+1}\right](\cdot,\cdot) = f(\boldsymbol{v}_h, \phi(\cdot,\cdot))_{[0,(H-h+1)^2]} - \left[f(\boldsymbol{u}_h, \phi(\cdot,\cdot))_{[0,H-h+1]}\right]^2$
7:     Set $\widehat{\sigma}_h(\cdot,\cdot)^2 \leftarrow \max\{1, \widehat{\text{Var}}_{P_h}\widehat{V}_{h+1}(\cdot,\cdot)\}$
8:     Set $\widehat{\theta}_h \leftarrow \arg\min_{\theta \in \Theta} \left\{\sum_{k=1}^K \left[f\left(\theta, \phi_{h,k}\right) - r_{h,k} - \widehat{V}_{h+1}(s_{h+1}^k)\right]^2 / \widehat{\sigma}_h^2(s_h^k, a_h^k) + \lambda \cdot \|\theta\|_2^2\right\}$
9:     Set $\Lambda_h \leftarrow \sum_{k=1}^K \nabla f(\widehat{\theta}_h, \phi_{h,k})\nabla f(\widehat{\theta}_h, \phi_{h,k})^\top / \widehat{\sigma}^2(s_h^k, a_h^k) + \lambda \cdot I$,
10:     Set $\Gamma_h(\cdot,\cdot) \leftarrow \beta\sqrt{\nabla_\theta f(\widehat{\theta}_h, \phi(\cdot,\cdot))^\top \Lambda_h^{-1} \nabla_\theta f(\widehat{\theta}_h, \phi(\cdot,\cdot))} \left(+\widetilde{O}(\frac{1}{K})\right)$
11:     Set $\bar{Q}_h(\cdot,\cdot) \leftarrow f(\widehat{\theta}_h, \phi(\cdot,\cdot)) - \Gamma_h(\cdot,\cdot), \widehat{Q}_h(\cdot,\cdot) \leftarrow \min\left\{\bar{Q}_h(\cdot,\cdot), H - h + 1\right\}^+$
12:     Set $\widehat{\pi}_h(\cdot \mid \cdot) \leftarrow \arg\max_{\pi_h} \left\langle\widehat{Q}_h(\cdot,\cdot), \pi_h(\cdot \mid \cdot)\right\rangle_{\mathcal{A}}, \widehat{V}_h(\cdot) \leftarrow \max_{\pi_h} \left\langle\widehat{Q}_h(\cdot,\cdot), \pi_h(\cdot \mid \cdot)\right\rangle_{\mathcal{A}}$
13: **end for**
14: **Output:** $\{\widehat{\pi}_h\}_{h=1}^H$.

---

### J.1  PROVABLE EFFICIENCY FOR VARIANCE-AWARE FITTED Q LEARNING

Recall the objective

$$\ell_h(\theta) := \frac{1}{K}\sum_{k=1}^K \left[f\left(\theta, \phi(s_h^k, a_h^k)\right) - r(s_h^k, a_h^k) - \widehat{V}_{h+1}(s_{h+1}^k)\right]^2 / \widehat{\sigma}_h^2(s_h^k, a_h^k) + \frac{\lambda}{K} \cdot \|\theta\|_2^2$$

Then by definition, $\widehat{\theta}_h := \arg\min_{\theta \in \Theta} \ell_h(\theta)$ and $\theta_{\mathbb{T}\widehat{V}_{h+1}}$ satisfies $f(\theta_{\mathbb{T}\widehat{V}_{h+1}}, \phi) = \mathcal{P}_h\widehat{V}_{h+1}(s_{h+1}^k)$ (recall $\epsilon_{\mathcal{F}} = 0$). Therefore, in this case, we have the following lemma:

**Lemma J.1.** *Fix $h \in [H]$. With probability $1 - \delta$,*

$$\mathbb{E}_\mu[\ell_h(\widehat{\theta}_h)] - \mathbb{E}_\mu[\ell_h(\theta_{\mathbb{T}\widehat{V}_{h+1}})] \leq \frac{36H^2(\log(1/\delta) + C_{d,\log K}) + \lambda C_\Theta^2}{K}$$

*where the expectation over $\mu$ is taken w.r.t. $(s_h^k, a_h^k, s_{h+1}^k)$ $k = 1, ..., K$ only (i.e., first compute $\mathbb{E}_\mu[\ell_h(\theta)]$ for a fixed $\theta$, then plug-in either $\widehat{\theta}_{h+1}$ or $\theta_{\mathbb{T}\widehat{V}_{h+1}}$). Here*

$$C_{d,\log(K)} := d\log(1+24C_\Theta(H+1)\kappa_1 K) + d\log\left(1+288H^2 C_\Theta(\kappa_1\sqrt{C_\Theta}+2\sqrt{\kappa_1\kappa_2/\lambda})^2 K^2\right) +$$
$$d^2\log\left(1+288H^2\sqrt{d}\kappa_1^2 K^2/\lambda\right) + d\log(1+16C_\Theta H^2\kappa_1 K) + d\log(1+32C_\Theta H^3\kappa_1 K).$$

*Proof of Lemma J.1.* **Step1:** Consider the case where $\lambda = 0$. Indeed, fix $h \in [H]$ and any function $V(\cdot) \in \mathbb{R}^{\mathcal{S}}$. Similarly, define $f_V(s,a) := f(\theta_{\mathbb{T}V}, \phi) = \mathcal{P}_h V$. For any fixed $\theta \in \Theta$, denote $g(s,a) = f(\theta, \phi(s,a))$. Moreover, for any $u,v \in \Theta$, define

$$\sigma_{u,v}^2(\cdot,\cdot) := \max\{1, f(v,\phi(\cdot,\cdot))_{[0,(H-h+1)^2]} - \left[f(u,\phi(\cdot,\cdot))_{[0,H-h+1]}\right]^2\}$$

Then define (we omit the subscript $u,v$ of $\sigma_{u,v}^2$ for the illustration purpose when there is no ambiguity)

$$X(g,V,f_V,\sigma^2) := \frac{(g(s,a)-r-V(s'))^2 - (f_V(s,a)-r-V(s'))^2}{\sigma_{u,v}^2(s,a)}.$$

Since all episodes are independent of each other, $X_k(g,V,f_V) :=$ $X(g(s_h^k,a_h^k), V(s_{h+1}^k), f_V(s_h^k,a_h^k), \sigma^2(s_h^k,a_h^k))$ are independent r.v.s and it holds

$$\frac{1}{K}\sum_{k=1}^{K} X_k(g,V,f_V,\sigma^2) = \ell(g) - \ell(f_V). \tag{25}$$

Next, the variance of $X$ is bounded by

$$\mathrm{Var}[X(g,V,f_V,\sigma^2)] \le \mathbb{E}_\mu[X(g,f,f_V,\sigma^2)^2]$$
$$=\mathbb{E}_\mu\left[\left((g(s_h,a_h)-r_h-V(s_{h+1}))^2 - (f_V(s_h,a_h)-r_h-V(s_{h+1}))^2\right)^2 / \sigma^2(s_h,a_h)^2\right]$$
$$=\mathbb{E}_\mu\left[\frac{(g(s_h,a_h)-f_V(s_h,a_h))^2}{\sigma^2(s_h,a_h)} \cdot \frac{(g(s_h,a_h)+f_V(s_h,a_h)-2r_h-2V(s_{h+1}))^2}{\sigma^2(s_h,a_h)}\right]$$
$$\le 4H^2 \cdot \mathbb{E}_\mu[\frac{(g(s_h,a_h)-f_V(s_h,a_h))^2}{\sigma^2(s_h,a_h)}]$$
$$=4H^2 \cdot \mathbb{E}_\mu\left[\frac{(g(s_h,a_h)-r_h-V(s_{h+1}))^2 - (f_V(s_h,a_h)-r_h-V(s_{h+1}))^2}{\sigma^2(s_h,a_h)}\right] \quad (*)$$
$$=4H^2 \cdot \mathbb{E}_\mu[X(g,f,f_V,\sigma^2)]$$

$(*)$ follows from that

$$\mathbb{E}_\mu\left[\frac{f(\widehat{\theta}_h, \phi(s_h,a_h)) - f(\theta_{\mathbb{T}\widehat{V}_{h+1}}, \phi(s_h,a_h))}{\sigma^2(s_h,a_h)} \cdot \mathbb{E}\left(f\left(\theta_{\mathbb{T}\widehat{V}_{h+1}}, \phi(s_h,a_h)\right) - r_h - \widehat{V}_{h+1}(s_{h+1})\Big|s_h,a_h\right)\right] = 0.$$

Therefore, by Bernstein inequality, with probability $1-\delta$,

$$\mathbb{E}_\mu[X(g,f,f_V,\sigma^2)] - \frac{1}{K}\sum_{k=1}^{K}X_k(g,f,f_V,\sigma^2)$$
$$\le \sqrt{\frac{2\mathrm{Var}[X(g,f,f_V,\sigma^2)]\log(1/\delta)}{K}} + \frac{4H^2\log(1/\delta)}{3K}$$
$$\le \sqrt{\frac{8H^2\mathbb{E}_\mu[X(g,f,f_V,\sigma^2)]\log(1/\delta)}{K}} + \frac{4H^2\log(1/\delta)}{3K}.$$

Now, if we choose $g(s,a) := f(\widehat{\theta}_h, \phi(s,a))$ and $u = \boldsymbol{u}_h, v = \boldsymbol{v}_h$ from Algorithm 3, then $\widehat{\theta}_h$ minimizes $\ell_h(\theta)$, therefore, it also minimizes $\frac{1}{K}\sum_{k=1}^{K}X_i(\theta,\widehat{V}_{h+1}, f_{\widehat{V}_{h+1}}, \widehat{\sigma}_h^2)$ and this implies

$$\frac{1}{K}\sum_{k=1}^{K}X_k(\widehat{\theta}_h, \widehat{V}_{h+1}, f_{\widehat{V}_{h+1}}, \widehat{\sigma}_h^2) \le \frac{1}{K}\sum_{k=1}^{K}X_k(\theta_{\mathbb{T}\widehat{V}_{h+1}}, \widehat{V}_{h+1}, f_{\widehat{V}_{h+1}}, \widehat{\sigma}_h^2) = 0.$$

Thus, we obtain

$$\mathbb{E}_\mu[X(\widehat{\theta}_h, \widehat{V}_{h+1}, f_{\widehat{V}_{h+1}}, \widehat{\sigma}_h^2)] \le \sqrt{\frac{8H^2 \cdot \mathbb{E}_\mu[X(\widehat{\theta}_h, \widehat{V}_{h+1}, f_{\widehat{V}_{h+1}}, \widehat{\sigma}_h^2)] \log(1/\delta)}{K}} + \frac{4H^2 \log(1/\delta)}{3K}.$$

However, the above does not hold with probability $1 - \delta$ since $\widehat{\theta}_h$, $\widehat{\sigma}_h^2$ and $\widehat{V}_{h+1} :=$ $\min\{\max_a f(\widehat{\theta}_{h+1}, \phi(\cdot, a)) - \sqrt{\nabla f(\widehat{\theta}_{h+1}, \phi(\cdot, a))^\top A \cdot \nabla f(\theta, \phi(\cdot, a))}, H\}$ (where $A$ is certain symmetric matrix with bounded norm) depend on $\widehat{\theta}_h$, $\widehat{\theta}_{h+1}$ which are data-dependent. Therefore, we need to further apply covering Lemma L.11 and choose $\epsilon = O(1/K)$ and a union bound to obtain with probability $1 - \delta$,

$$\mathbb{E}_\mu[X(\widehat{\theta}_h, \widehat{V}_{h+1}, f_{\widehat{V}_{h+1}}, \widehat{\sigma}_h^2)] \le \sqrt{\frac{8H^2 \cdot \mathbb{E}_\mu[X(\widehat{\theta}_h, \widehat{V}_{h+1}, f_{\widehat{V}_{h+1}}, \widehat{\sigma}_h^2)](\log(1/\delta) + C_{d,\log K})}{K}} + \frac{4H^2(\log(1/\delta) + C_{d,\log K})}{3K}.$$

where $C_{d,\log(K)} := d \log(1 + 24C_\Theta(H+1)\kappa_1 K) + d \log\left(1 + 288H^2 C_\Theta(\kappa_1\sqrt{C_\Theta} + 2\sqrt{\kappa_1 \kappa_2/\lambda})^2 K^2\right) + d^2 \log\left(1 + 288H^2\sqrt{d}\kappa_1^2 K^2/\lambda\right) + d \log(1 + 16C_\Theta H^2 \kappa_1 K) + d \log(1 + 32C_\Theta H^3 \kappa_1 K)$ (where we let $B = 1/\lambda$ since $\|\Lambda_h^{-1}\|_2 \le 1/\lambda$). Solving this quadratic equation to obtain with probability $1 - \delta$,

$$\mathbb{E}_\mu[X(\widehat{\theta}_h, \widehat{V}_{h+1}, f_{\widehat{V}_{h+1}})] \le \frac{36H^2(\log(1/\delta) + C_{d,\log K})}{K}.$$

Now according to equation 25, by definition we finally have with probability $1 - \delta$ (recall the expectation over $\mu$ is taken w.r.t. $(s_h^k, a_h^k, s_{h+1}^k)$ $k = 1, ..., K$ only)

$$\mathbb{E}_\mu[\ell_h(\widehat{\theta}_{h+1})] - \mathbb{E}_\mu[\ell_h(\theta_{\mathbb{T}\widehat{V}_{h+1}})] = \mathbb{E}_\mu[X(\widehat{\theta}_h, \widehat{V}_{h+1}, f_{\widehat{V}_{h+1}})] \le \frac{36H^2(\log(1/\delta) + C_{d,\log K})}{K} \quad (26)$$

where we used $f(\theta_{\mathbb{T}\widehat{V}_{h+1}}, \phi) = \mathcal{P}_h \widehat{V}_{h+1} = f_{\widehat{V}_{h+1}}$.

**Step2.** If $\lambda > 0$, there is only extra term $\frac{\lambda}{K}\left(\left\|\widehat{\theta}_h\right\|_2 - \left\|\theta_{\mathbb{T}\widehat{V}_{h+1}}\right\|_2\right) \le \frac{\lambda}{K}\left\|\widehat{\theta}_h\right\|_2 \le \frac{\lambda C_\Theta^2}{K}$ in addition to above. This finishes the proof.

$\square$

**Theorem J.2** (Provable efficiency for VAFQL). *Let $C_{d,\log K}$ be the same as Lemma J.1. Then, with probability $1 - \delta$*

$$\left\|\widehat{\theta}_h - \theta_{\mathbb{T}\widehat{V}_{h+1}}\right\|_2 \le \sqrt{\frac{36H^4(\log(H/\delta) + C_{d,\log K}) + 2\lambda C_\Theta^2}{\kappa K}}, \ \forall h \in [H].$$

*Proof of Theorem J.2.* Apply a union bound in Lemma J.1, we have with probability $1 - \delta$,

$$\mathbb{E}_\mu[\ell_h(\widehat{\theta}_h)] - \mathbb{E}_\mu[\ell_h(\theta_{\mathbb{T}\widehat{V}_{h+1}})] \le \frac{36H^2(\log(H/\delta) + C_{d,\log K}) + \lambda C_\Theta^2}{K}, \ \forall h \in [H]$$

$$\Rightarrow \mathbb{E}_\mu[\ell_h(\widehat{\theta}_h) - \frac{\lambda}{K}\left\|\widehat{\theta}_h\right\|_2^2] - \mathbb{E}_\mu[\ell_h(\theta_{\mathbb{T}\widehat{V}_{h+1}}) - \frac{\lambda}{K}\left\|\theta_{\mathbb{T}\widehat{V}_{h+1}}\right\|_2^2] \le \frac{36H^2(\log(H/\delta) + C_{d,\log K}) + 2\lambda C_\Theta^2}{K}$$

$$(27)$$

Now we prove for all $h \in [H]$,

$$\mathbb{E}_\mu\left[\left(f(\widehat{\theta}_h, \phi(\cdot, \cdot)) - f(\theta_{\mathbb{T}\widehat{V}_{h+1}}, \phi(\cdot, \cdot))\right)^2\right] = \mathbb{E}_\mu\left[\ell_h(\widehat{\theta}_h) - \frac{\lambda\left\|\widehat{\theta}_h\right\|_2^2}{K}\right] - \mathbb{E}_\mu\left[\ell_h(\theta_{\mathbb{T}\widehat{V}_{h+1}}) - \frac{\lambda\left\|\theta_{\mathbb{T}\widehat{V}_{h+1}}\right\|_2^2}{K}\right].$$

$$(28)$$

Indeed, identical to equation 26,

$$
\mathbb{E}_\mu \left[ \ell_h(\widehat{\theta}_h) - \frac{\lambda \left\| \widehat{\theta}_h \right\|_2^2}{K} \right] - \mathbb{E}_\mu \left[ \ell_h(\theta_{\mathbb{T}\widehat{V}_{h+1}}) - \frac{\lambda \left\| \theta_{\mathbb{T}\widehat{V}_{h+1}} \right\|_2^2}{K} \right] = \mathbb{E}_\mu[X(\widehat{\theta}_h, \widehat{V}_{h+1}, f_{\widehat{V}_{h+1}})]
$$

$$
= \mathbb{E}_\mu \left( \left[ f\left( \widehat{\theta}_h, \phi(s_h, a_h) \right) - r_h - \widehat{V}_{h+1}(s_{h+1}) \right]^2 / \widehat{\sigma}_h^2(s_h, a_h) - \left[ f\left( \theta_{\mathbb{T}\widehat{V}_{h+1}}, \phi(s_h, a_h) \right) - r_h - \widehat{V}_{h+1}(s_{h+1}) \right]^2 / \widehat{\sigma}_h^2(s_h, a_h) \right)
$$

$$
= \mathbb{E}_\mu \left[ \left( f(\widehat{\theta}_h, \phi(\cdot, \cdot)) - f(\theta_{\mathbb{T}\widehat{V}_{h+1}}, \phi(\cdot, \cdot)) \right)^2 / \widehat{\sigma}_h^2(\cdot, \cdot) \right]
$$

$$
+ \mathbb{E}_\mu \left[ \left( f(\widehat{\theta}_h, \phi(s_h, a_h)) - f(\theta_{\mathbb{T}\widehat{V}_{h+1}}, \phi(s_h, a_h)) \right) \cdot \left( f\left( \theta_{\mathbb{T}\widehat{V}_{h+1}}, \phi(s_h, a_h) \right) - r_h - \widehat{V}_{h+1}(s_{h+1}) \right) / \widehat{\sigma}_h^2(s_h, a_h) \right]
$$

$$
= \mathbb{E}_\mu \left[ \left( f(\widehat{\theta}_h, \phi(\cdot, \cdot)) - f(\theta_{\mathbb{T}\widehat{V}_{h+1}}, \phi(\cdot, \cdot)) \right)^2 / \widehat{\sigma}_h^2(\cdot, \cdot) \right]
$$

$$
+ \mathbb{E}_\mu \left[ \left( f(\widehat{\theta}_h, \phi(s_h, a_h)) - f(\theta_{\mathbb{T}\widehat{V}_{h+1}}, \phi(s_h, a_h)) \right) \cdot \mathbb{E} \left( f\left( \theta_{\mathbb{T}\widehat{V}_{h+1}}, \phi(s_h, a_h) \right) - r_h - \widehat{V}_{h+1}(s_{h+1}) \Big| s_h, a_h \right) / \widehat{\sigma}_h^2(s_h, a_h) \right]
$$

$$
= \mathbb{E}_\mu \left[ \left( f(\widehat{\theta}_h, \phi(\cdot, \cdot)) - f(\theta_{\mathbb{T}\widehat{V}_{h+1}}, \phi(\cdot, \cdot)) \right)^2 / \widehat{\sigma}_h^2(\cdot, \cdot) \right]
$$

where the third identity uses law of total expectation and that $\mu$ is taken w.r.t. $s_h, a_h, s_{h+1}$ only (recall Lemma J.1) so the $\widehat{\sigma}_h^2$ can be move outside of the conditional expectation.[12] The fourth identity uses the definition of $\theta_{\mathbb{T}\widehat{V}_{h+1}}$ since $f(\theta_{\mathbb{T}\widehat{V}_{h+1}}, \phi(s, a)) = \mathcal{P}_{h,s,a}\widehat{V}_{h+1}$.

Then we have

$$
\mathbb{E}_\mu \left[ \left( f(\widehat{\theta}_h, \phi(\cdot, \cdot)) - f(\theta_{\mathbb{T}\widehat{V}_{h+1}}, \phi(\cdot, \cdot)) \right)^2 / \widehat{\sigma}_h^2(\cdot, \cdot) \right]
$$

$$
\geq \mathbb{E}_\mu \left[ \left( f(\widehat{\theta}_h, \phi(\cdot, \cdot)) - f(\theta_{\mathbb{T}\widehat{V}_{h+1}}, \phi(\cdot, \cdot)) \right)^2 \right] / H^2 \geq \frac{\kappa}{H^2} \left\| \widehat{\theta}_h - \theta_{\mathbb{T}\widehat{V}_{h+1}} \right\|_2^2,
$$

where the third identity uses $\mu$ is over $s_h, a_h$ only and the last one uses $\widehat{\sigma}_h^2(\cdot, \cdot) \leq H^2$. Combine the above with equation 27 and equation 28, we obtain the stated result. $\qquad\square$

**Theorem J.3** (Provable efficiency of VAFQL (Part II)). *Let $C_{d,\log K}$ be the same as Lemma J.1. Furthermore, suppose $\lambda \leq 1/2C_\Theta^2$ and $K \geq \max \left\{ 512 \frac{\kappa_1^4}{\kappa^2} \left( \log(\frac{2d}{\delta}) + d\log(1 + \frac{4\kappa_1^3\kappa_2 C_\Theta K^3}{\lambda^2}) \right), \frac{4\lambda}{\kappa} \right\}$. Then, with probability $1 - \delta$, $\forall h \in [H]$*

$$
\sup_{s,a} \left| f(\widehat{\theta}_h, \phi(s, a)) - f(\theta_h^\star, \phi(s, a)) \right| \leq \left( \kappa_1 H \sqrt{\frac{36H^4(\log(H/\delta) + C_{d,\log K}) + 2\lambda C_\Theta^2}{\kappa}} + \frac{2dH^3\kappa_1}{\sqrt{\kappa}} \right) \sqrt{\frac{1}{K}} + O(\frac{1}{K}),
$$

*Furthermore, we have with probability $1 - \delta$,*

$$
\sup_h \left\| \widehat{V}_h - V_h^\star \right\|_\infty \leq \left( \kappa_1 H \sqrt{\frac{36H^4(\log(H/\delta) + C_{d,\log K}) + 2\lambda C_\Theta^2}{\kappa}} + \frac{2dH^3\kappa_1}{\sqrt{\kappa}} \right) \sqrt{\frac{1}{K}} + O(\frac{1}{K})
$$

$$
= \widetilde{O} \left( \kappa_1 H^3 \sqrt{\frac{d^2}{\kappa}} \sqrt{\frac{1}{K}} \right)
$$

*where $\widetilde{O}$ absorbs Polylog terms and higher order terms. Lastly, it also holds for all $h \in [H]$, w.p. $1 - \delta$*

$$
\left\| \widehat{\theta}_h - \theta_h^\star \right\|_2 \leq \left( \kappa_1 H \frac{\sqrt{72H^4(\log(H^2/\delta) + C_{d,\log K}) + 4\lambda C_\Theta^2}}{\kappa} + \frac{4H^3 d\kappa_1}{\kappa} \right) \sqrt{\frac{1}{K}} + O(\frac{1}{K})
$$

$$
= \widetilde{O} \left( \frac{\kappa_1 H^3 d}{\kappa} \sqrt{\frac{1}{K}} \right)
$$

---

[12]Recall $\widehat{\sigma}_h^2$ computed in Algorithm 3 uses an independent copy $\mathcal{D}'$.

*Proof of Theorem J.3.* **Step1:** we show the first result.

We prove this by backward induction. When $h = H + 1$, by convention $f(\widehat{\theta}_h, \phi(s,a)) = f(\theta_h^\star, \phi(s,a)) = 0$ so the base case holds. Suppose for $h + 1$, with probability $1 - (H - h)\delta$, $\sup_{s,a} \left| f(\widehat{\theta}_h, \phi(s,a)) - f(\theta_h^\star, \phi(s,a)) \right| \leq C_{h+1}\sqrt{\frac{1}{K}}$, we next consider the case for $t = h$.

On one hand, by Theorem J.2, we have with probability $1 - \delta/2$,

$$
\sup_{s,a} \left| f(\widehat{\theta}_h, \phi(s,a)) - f(\theta_h^\star, \phi(s,a)) \right|
$$

$$
\leq \sup_{s,a} \left| f(\widehat{\theta}_h, \phi(s,a)) - f(\theta_{\mathbb{T}\widehat{V}_{h+1}}, \phi(s,a)) \right| + \sup_{s,a} \left| f(\theta_{\mathbb{T}\widehat{V}_{h+1}}, \phi(s,a)) - f(\theta_h^\star, \phi(s,a)) \right|
$$

$$
= \sup_{s,a} \left| \nabla f(\xi, \phi(s,a))^\top (\widehat{\theta}_h - \theta_{\mathbb{T}\widehat{V}_{h+1}}) \right| + \sup_{s,a} \left| f(\theta_{\mathbb{T}\widehat{V}_{h+1}}, \phi(s,a)) - f(\theta_{\mathbb{T}V_{h+1}^\star}, \phi(s,a)) \right|
$$

$$
\leq \kappa_1 \cdot \left\| \widehat{\theta}_h - \theta_{\mathbb{T}\widehat{V}_{h+1}} \right\|_2 + \sup_{s,a} \left| \mathcal{P}_{h,s,a}\widehat{V}_{h+1} - \mathcal{P}_{h,s,a}V_{h+1}^\star \right|
$$

$$
\leq \kappa_1 \sqrt{\frac{36H^4(\log(H/\delta) + C_{d,\log K}) + 2\lambda C_\Theta^2}{\kappa K}} + \left\| \widehat{V}_{h+1} - V_{h+1}^\star \right\|_\infty,
$$

Recall we have the form $\widehat{V}_{h+1}(\cdot) := \min\{\max_a f(\widehat{\theta}_{h+1}, \phi(\cdot, a)) - \Gamma_h(\cdot, a), H\}$ and $V_{h+1}^\star(\cdot) = \max_a f(\theta_{h+1}^\star, \phi(\cdot, a)) = \min\{\max_a f(\theta_{h+1}^\star, \phi(\cdot, a)), H\}$, we obtain

$$
\left\| \widehat{V}_{h+1} - V_{h+1}^\star \right\|_\infty \leq \sup_{s,a} \left| f(\widehat{\theta}_{h+1}, \phi(s,a)) - f(\theta_{h+1}^\star, \phi(s,a)) \right| + \sup_{h,s,a} \Gamma_h(s,a) \qquad (29)
$$

Note the above holds true for any generic $\Gamma_h(s,a)$. In particular, according to Algorithm 3, we specify

$$
\Gamma_h(\cdot, \cdot) = d\sqrt{\nabla_\theta f(\widehat{\theta}_h, \phi(\cdot, \cdot))^\top \Lambda_h^{-1} \nabla_\theta f(\widehat{\theta}_h, \phi(\cdot, \cdot))} \left( + \widetilde{O}(\frac{1}{K}) \right)
$$

and by Lemma L.5, with probability $1 - \delta$ (note here $\Sigma_h^{-1}$ is replaced by $\Lambda_h^{-1}$ and $\left\| \Lambda_h^{-1} \right\|_2 \leq H^2/\kappa$),

$$
\Gamma_h \leq \frac{2dH^2\kappa_1}{\sqrt{\kappa K}} + O(\frac{1}{K})
$$

and by a union bound this implies with probability $1 - (H - h + 1)\delta$,

$$
\sup_{s,a} \left| f(\widehat{\theta}_h, \phi(s,a)) - f(\theta_h^\star, \phi(s,a)) \right|
$$

$$
\leq C_{h+1}\sqrt{\frac{1}{K}} + \kappa_1 \sqrt{\frac{36H^4(\log(H/\delta) + C_{d,\log K}) + 2\lambda C_\Theta^2}{\kappa K}} + \frac{2dH^2\kappa_1}{\sqrt{\kappa K}} + O(\frac{1}{K}) := C_h\sqrt{\frac{1}{K}}.
$$

Solving for $C_h$, we obtain $C_h \leq \kappa_1 H \sqrt{\frac{36H^4(\log(H/\delta) + C_{d,\log K}) + 2\lambda C_\Theta^2}{\kappa}} + H\frac{2dH^2\kappa_1}{\sqrt{\kappa}}$ for all $H$. By a union bound (replacing $\delta$ by $\delta/H$), we obtain the stated result.

**Step2:** Utilizing the intermediate result equation 29, we directly have with probability $1 - \delta$,

$$
\sup_h \left\| \widehat{V}_h - V_h^\star \right\|_\infty \leq \sup_{s,a} \left| f(\widehat{\theta}_h, \phi(s,a)) - f(\theta_h^\star, \phi(s,a)) \right| + \frac{2dH^2\kappa_1}{\sqrt{\kappa K}} + O(\frac{1}{K}),
$$

where $\sup_{s,a} \left| f(\widehat{\theta}_h, \phi(s,a)) - f(\theta_h^\star, \phi(s,a)) \right|$ can be bounded using Step1.

**Step3:** Denote $M := \left( \kappa_1 H \sqrt{\frac{36H^4(\log(H^2/\delta) + C_{d,\log K}) + 2\lambda C_\Theta^2}{\kappa}} + \frac{2H^3 d\kappa_1}{\sqrt{\kappa}} \right) \sqrt{\frac{1}{K}} + O(\frac{1}{K})$, then by Step1 we have with probability $1 - \delta$ (here $\xi$ is some point between $\widehat{\theta}_h$ and $\theta_h^\star$) for all $h \in [H]$

$$
M^2 \geq \sup_{s,a} \left| f(\widehat{\theta}_h, \phi(s,a)) - f(\theta_h^\star, \phi(s,a)) \right|^2
$$

$$
\geq \mathbb{E}_\mu[\left( f(\widehat{\theta}_h, \phi(s,a)) - f(\theta_h^\star, \phi(s,a)) \right)^2] \geq \kappa \left\| \widehat{\theta}_h - \theta_h^\star \right\|_2^2
$$

where the last step is by Assumption 2.3. Solving this to obtain the stated result.

$\square$

## J.2 BOUNDING $|\widehat{\sigma}_h^2 - \sigma_h^{\star 2}|$

Recall the definition $\sigma_h^{\star 2}(\cdot,\cdot) = \max\{1, [\mathrm{Var}_{P_h} V_{h+1}^\star](\cdot,\cdot)\}$. In this section, we bound the term $|\widehat{\sigma}_h^2 - \sigma_h^{\star 2}| := \left\|\widehat{\sigma}_h^2(\cdot,\cdot) - \sigma_h^{\star 2}(\cdot,\cdot)\right\|_\infty$ and

$$
\begin{aligned}
\boldsymbol{u}_h &= \underset{\theta\in\Theta}{\arg\min}\left\{\frac{1}{K}\sum_{k=1}^{K}\left[f\left(\theta,\bar{\phi}_{h,k}\right) - \widehat{V}_{h+1}(\bar{s}_{h+1}^k)\right]^2 + \frac{\lambda}{K}\cdot\|\theta\|_2^2\right\}\\
\boldsymbol{v}_h &= \underset{\theta\in\Theta}{\arg\min}\left\{\frac{1}{K}\sum_{k=1}^{K}\left[f\left(\theta,\bar{\phi}_{h,k}\right) - \widehat{V}_{h+1}^2(\bar{s}_{h+1}^k)\right]^2 + \frac{\lambda}{K}\cdot\|\theta\|_2^2\right\}
\end{aligned}
\tag{30}
$$

where

$$
\widehat{\sigma}_h^2(\cdot,\cdot) := \max\{1, f(\boldsymbol{v}_h,\phi(\cdot,\cdot))_{[0,(H-h+1)^2]} - \left[f(\boldsymbol{u}_h,\phi(\cdot,\cdot))_{[0,H-h+1]}\right]^2\}
$$

and true parameters $\boldsymbol{u}_h^\star, \boldsymbol{v}_h^\star$ satisfy $f(\boldsymbol{u}_h^\star, \phi(\cdot,\cdot)) = \mathbb{E}_{P(s'|\cdot,\cdot)}[V_h^\star(s')], f(\boldsymbol{v}_h^\star,\phi) = \mathbb{E}_{P(s'|\cdot,\cdot)}[V_h^{\star 2}(s')]$. Furthermore, we define

$$
\sigma_{\widehat{V}_{h+1}}^2(\cdot,\cdot) := \max\{1, [\mathrm{Var}_{P_h}\widehat{V}_{h+1}](\cdot,\cdot)\}
$$

and the *parameter Expectation operator* $\mathbb{J}: V\in\mathbb{R}^{\mathcal{S}}\to\theta_{\mathbb{J}V}\in\Theta$ such that:

$$
f(\theta_{\mathbb{J}V},\phi) = \mathbb{E}_{P_h}[V(s')], \quad\forall\|V\|_2 \le \mathcal{B}_{\mathcal{F}}.
$$

Note $\theta_{\mathbb{J}V}\in\Theta$ by Bellman completeness, reward $r$ is constant and differentiability (Definition 1.1) is an additive closed property. By definition,

$$
\begin{aligned}
|\widehat{\sigma}_h^2 - \sigma_{\widehat{V}_{h+1}}^2| &\le |f(\boldsymbol{v}_h,\phi) - f(\theta_{\mathbb{J}\widehat{V}_{h+1}^2},\phi)| + |f(\boldsymbol{u}_h,\phi)^2 - f(\theta_{\mathbb{J}\widehat{V}_{h+1}},\phi)^2|\\
&\le |f(\boldsymbol{v}_h,\phi) - f(\theta_{\mathbb{J}\widehat{V}_{h+1}^2},\phi)| + 2H\cdot|f(\boldsymbol{u}_h,\phi) - f(\theta_{\mathbb{J}\widehat{V}_{h+1}},\phi)|
\end{aligned}
$$

and

$$
\begin{aligned}
|\sigma_h^{\star 2} - \widehat{\sigma}_h^2| &\le |f(\boldsymbol{v}_h^\star,\phi) - f(\boldsymbol{v}_h,\phi)| + |f(\boldsymbol{u}_h^\star,\phi)^2 - f(\boldsymbol{v}_h,\phi)^2|\\
&\le |f(\boldsymbol{v}_h^\star,\phi) - f(\boldsymbol{v}_h,\phi)| + 2H\cdot|f(\boldsymbol{u}_h^\star,\phi) - f(\boldsymbol{v}_h,\phi)|
\end{aligned}
$$

We first give the following result.

**Lemma J.4.** *Suppose* $\lambda\le 1/2C_\Theta^2$ *and* $K\ge\max\left\{512\frac{\kappa_1^4}{\kappa^2}\left(\log(\frac{2d}{\delta}) + d\log(1 + \frac{4\kappa_1^3\kappa_2 C_\Theta K^3}{\lambda^2})\right), \frac{4\lambda}{\kappa}\right\}$.
*Then, with probability* $1-\delta$, $\forall h\in[H]$,

$$
\left\|\boldsymbol{u}_h - \theta_{\mathbb{J}\widehat{V}_{h+1}}\right\|_2 \le \sqrt{\frac{36H^2(\log(H/\delta) + \widetilde{O}(d^2)) + 2\lambda C_\Theta^2}{\kappa K}}, \quad\forall h\in[H],
$$

$$
\left\|\boldsymbol{v}_h - \theta_{\mathbb{J}\widehat{V}_{h+1}^2}\right\|_2 \le \sqrt{\frac{36H^4(\log(H/\delta) + \widetilde{O}(d^2)) + 2\lambda C_\Theta^2}{\kappa K}}, \quad\forall h\in[H].
$$

*and*

$$
\sup_{s,a}|f(\boldsymbol{u}_h,\phi(s,a)) - f(\boldsymbol{u}_h^\star,\phi(s,a))| \le \left(\kappa_1 H\sqrt{\frac{36H^2(\log(H^2/\delta) + \widetilde{O}(d^2)) + 2\lambda C_\Theta^2}{\kappa}} + \frac{2H^2 d\kappa_1}{\sqrt{\kappa}}\right)\sqrt{\frac{1}{K}} + O(\frac{1}{K}),
$$

$$
\sup_{s,a}|f(\boldsymbol{v}_h,\phi(s,a)) - f(\boldsymbol{v}_h^\star,\phi(s,a))| \le \left(\kappa_1 H\sqrt{\frac{36H^4(\log(H^2/\delta) + \widetilde{O}(d^2)) + 2\lambda C_\Theta^2}{\kappa}} + \frac{2H^3 d\kappa_1}{\sqrt{\kappa}}\right)\sqrt{\frac{1}{K}} + O(\frac{1}{K}).
$$

*The above directly implies for all* $h\in[H]$, *with probability* $1-\delta$,

$$
|\sigma_h^{\star 2} - \widehat{\sigma}_h^2| \le \left(3\kappa_1 H^2\sqrt{\frac{36H^4(\log(H^2/\delta) + \widetilde{O}(d^2)) + 2\lambda C_\Theta^2}{\kappa}} + \frac{6H^4 d\kappa_1}{\sqrt{\kappa}}\right)\sqrt{\frac{1}{K}} + O(\frac{1}{K})
$$

$$
|\widehat{\sigma}_h^2 - \sigma_{\widehat{V}_{h+1}}^2| \le 3H\kappa_1\sqrt{\frac{36H^4(\log(H/\delta) + \widetilde{O}(d^2)) + 2\lambda C_\Theta^2}{\kappa K}}.
$$

*Proof of Lemma J.4.* In fact, the proof follows a reduction from the provable efficiency procedure conducted in Section G. This is due to the regression procedure in equation 30 is the same as the procedure equation 17 except the *parameter Bellman operator* $\mathbb{T}$ is replaced by the *parameter Expectation operator* $\mathbb{J}$ (recall here $\bar{\phi}_{h,k}$ uses the independent copy $\mathcal{D}'$ and $\widetilde{O}(d^2)$ comes from the covering argument.). Concretely, the $X(g, V, f_V)$ used in Lemma G.1 will be modified to $X(g, V, f_V) = (g(s, a) - V(s'))^2 - (f(\theta_{\mathbb{J}V}, \phi(s, a)) - V(s'))^2$ by removing reward information and the decomposition

$$\mathbb{E}\mu \left[ (g(s_h, a_h) - V(s_{h+1}))^2 - (f(\theta_{\mathbb{J}V}, \phi(s_h, a_h)) - V(s_{h+1}))^2 \right] = \mathbb{E}_\mu \left[ (g(s_h, a_h) - f(\theta_{\mathbb{J}V}, \phi(s_h, a_h)))^2 \right]$$

holds true. Then with probability $1 - \delta$,

$$
\begin{aligned}
|\sigma_h^{\star 2} - \widehat{\sigma}_h^2| \leq & |f(\boldsymbol{v}_h^\star, \phi) - f(\boldsymbol{v}_h, \phi)| + 2H \cdot |f(\boldsymbol{u}_h^\star, \phi) - f(\boldsymbol{v}_h, \phi)| \\
\leq & \left( 3\kappa_1 H^2 \sqrt{\frac{36H^4(\log(H^2/\delta) + \widetilde{O}(d^2)) + 2\lambda C_\Theta^2}{\kappa}} + \frac{6H^4 d\kappa_1}{\sqrt{\kappa}} \right) \sqrt{\frac{1}{K}} + O(\frac{1}{K}).
\end{aligned}
$$

and

$$
\begin{aligned}
|\widehat{\sigma}_h^2 - \sigma_{\widehat{V}_{h+1}}^2| \leq & |f(\boldsymbol{v}_h, \phi) - f(\theta_{\mathbb{J}\widehat{V}_{h+1}^2}, \phi)| + 2H \cdot |f(\boldsymbol{u}_h, \phi) - f(\theta_{\mathbb{J}\widehat{V}_{h+1}}, \phi)| \\
\leq & \kappa_1 \left\| \boldsymbol{v}_h - \theta_{\mathbb{J}\widehat{V}_{h+1}^2} \right\|_2 + 2H\kappa_1 \left\| \boldsymbol{u}_h - \theta_{\mathbb{J}\widehat{V}_{h+1}} \right\|_2 \\
\leq & 3H\kappa_1 \sqrt{\frac{36H^4(\log(H/\delta) + \widetilde{O}(d^2)) + 2\lambda C_\Theta^2}{\kappa K}}.
\end{aligned}
$$

$\square$

## J.3 PROOF OF THEOREM 4.1

In this section, we sketch the proof of Theorem 4.1 since the most components are identical to Theorem 3.2. We will focus on highlighting the difference for obtaining the tighter bound.

First of all, Recall in the first-order condition, we have

$$
\nabla_\theta \left\{ \sum_{k=1}^K \frac{\left[ f(\theta, \phi_{h,k}) - r_{h,k} - \widehat{V}_{h+1}(s_{h+1}^k) \right]^2}{\widehat{\sigma}_h^2(s_h^k, a_h^k)} + \lambda \cdot \|\theta\|_2^2 \right\} \Bigg|_{\theta = \widehat{\theta}_h} = 0, \ \ \forall h \in [H].
$$

Therefore, if we define the quantity $Z_h(\cdot, \cdot) \in \mathbb{R}^d$ as

$$
Z_h(\theta | V, \sigma^2) = \sum_{k=1}^K \frac{\left[ f(\theta, \phi_{h,k}) - r_{h,k} - V(s_{h+1}^k) \right]}{\sigma(s_h^k, a_h^k)} \frac{\nabla f(\theta, \phi_{h,k})}{\sigma(s_h^k, a_h^k)} + \lambda \cdot \theta, \ \ \forall \theta \in \Theta, \|V\|_2 \leq H,
$$

then we have

$$
Z_h(\widehat{\theta}_h | \widehat{V}_{h+1}, \widehat{\sigma}_h^2) = 0.
$$

According to the regression oracle (Line 8 of Algorithm 3), the estimated Bellman operator $\widehat{\mathcal{P}}_h$ maps $\widehat{V}_{h+1}$ to $\widehat{\theta}_h$, *i.e.* $\widehat{\mathcal{P}}_h \widehat{V}_{h+1} = f(\widehat{\theta}_h, \phi)$. Therefore (recall Definition D.1)

$$
\begin{aligned}
& \mathcal{P}_h \widehat{V}_{h+1}(s, a) - \widehat{\mathcal{P}}_h \widehat{V}_{h+1}(s, a) = \mathcal{P}_h \widehat{V}_{h+1}(s, a) - f(\widehat{\theta}_h, \phi(s, a)) \\
= & f(\theta_{\mathbb{T}\widehat{V}_{h+1}}, \phi(s, a)) - f(\widehat{\theta}_h, \phi(s, a)) \\
= & \nabla f(\widehat{\theta}_h, \phi(s, a)) \left( \theta_{\mathbb{T}\widehat{V}_{h+1}} - \widehat{\theta}_h \right) + \text{Hot}_{h,1},
\end{aligned}
\tag{31}
$$

where we apply the first-order Taylor expansion for the differentiable function $f$ at point $\widehat{\theta}_h$ and $\text{Hot}_{h,1}$ is a higher-order term. Indeed, the following Lemma E.1 bounds the $\text{Hot}_{h,1}$ term with $\widetilde{O}(\frac{1}{K})$.

**Lemma J.5.** *Recall the definition (from the above decomposition)* $\mathrm{Hot}_{h,1} := f(\theta_{\mathbb{T}\widehat{V}_{h+1}}, \phi(s,a)) - f(\widehat{\theta}_h, \phi(s,a)) - \nabla f(\widehat{\theta}_h, \phi(s,a)) \left(\theta_{\mathbb{T}\widehat{V}_{h+1}} - \widehat{\theta}_h\right)$, *then with probability* $1 - \delta$,

$$|\mathrm{Hot}_{h,1}| \leq \widetilde{O}(\frac{1}{K}), \quad \forall h \in [H].$$

*Proof.* The proof is identical to that of Lemma E.1 but with the help of Lemma J.2. $\qquad\square$

Next, according to the expansion of $Z_h(\theta|\widehat{V}_{h+1}, \widehat{\sigma}_h^2)$, we have

$$\nabla f(\widehat{\theta}_h, \phi(s,a)) \left(\theta_{\mathbb{T}\widehat{V}_{h+1}} - \widehat{\theta}_h\right) = I_1 + I_2 + I_3 + \mathrm{Hot}_2, \tag{32}$$

where

$$\mathrm{Hot}_2 := \nabla f(\widehat{\theta}_h, \phi(s,a))\Lambda_h^{-1}\left[\widetilde{R}_K(\theta_{\mathbb{T}\widehat{V}_{h+1}}) + \lambda\theta_{\mathbb{T}\widehat{V}_{h+1}}\right]$$

$$\Delta_{\Lambda_h^s} = \sum_{k=1}^K \frac{\left(f(\widehat{\theta}_h, \phi_{h,k}) - r_{h,k} - \widehat{V}_{h+1}(s_{h+1}^k)\right) \cdot \nabla_{\theta\theta}^2 f(\widehat{\theta}_h, \phi_{h,k})}{\widehat{\sigma}^2(s_h^k, a_h^k)}$$

$$\Lambda_h = \sum_{k=1}^K \frac{\nabla_\theta f(\widehat{\theta}_h, \phi_{h,k})\nabla_\theta^\top f(\widehat{\theta}_{h,k}, \phi_{h,k})}{\widehat{\sigma}^2(s_h^k, a_h^k)} + \lambda I_d$$

$$\widetilde{R}_K(\theta_{\mathbb{T}\widehat{V}_{h+1}}) = \Delta_{\Lambda_h^s}(\widehat{\theta}_h - \theta_{\mathbb{T}\widehat{V}_{h+1}}) + R_K(\theta_{\mathbb{T}\widehat{V}_{h+1}})$$

where $R_K(\theta_{\mathbb{T}\widehat{V}_{h+1}})$ is the second order residual that is bounded by $\widetilde{O}(1/K)$ and

$$I_1 = \nabla f(\widehat{\theta}_h, \phi(s,a))\Lambda_h^{-1}\sum_{k=1}^K \frac{\left(f(\theta_{\mathbb{T}V_{h+1}^\star}, \phi_{h,k}) - r_{h,k} - V_{h+1}^\star(s_{h+1}^k)\right) \cdot \nabla_\theta^\top f(\widehat{\theta}_h, \phi_{h,k})}{\widehat{\sigma}_h^2(s_h^k, a_h^k)}$$

$$I_2 = \nabla f(\widehat{\theta}_h, \phi(s,a))\Lambda_h^{-1}\sum_{k=1}^K \frac{\left(f(\theta_{\mathbb{T}\widehat{V}_{h+1}}, \phi_{h,k}) - f(\theta_{\mathbb{T}V_{h+1}^\star}, \phi_{h,k}) - \widehat{V}_{h+1}(s_{h+1}^k) + V_{h+1}^\star(s_{h+1}^k)\right) \cdot \nabla_\theta^\top f(\widehat{\theta}_h, \phi_{h,k})}{\widehat{\sigma}_h^2(s_h^k, a_h^k)}$$

$$I_3 = \nabla f(\widehat{\theta}_h, \phi(s,a))\Lambda_h^{-1}\sum_{k=1}^K \frac{\left(f(\theta_{\mathbb{T}\widehat{V}_{h+1}}, \phi_{h,k}) - r_{h,k} - \widehat{V}_{h+1}(s_{h+1}^k)\right) \cdot \left(\nabla_\theta^\top f(\theta_{\mathbb{T}\widehat{V}_{h+1}}, \phi_{h,k}) - \nabla_\theta^\top f(\widehat{\theta}_h, \phi_{h,k})\right)}{\widehat{\sigma}_h^2(s_h^k, a_h^k)}$$

Similar to the PFQL case, $I_2, I_3, \mathrm{Hot}_2$ can be bounded to have order $O(1/K)$ via provably efficiency theorems in Section J.1 and in particular, the inclusion of $\sigma_{u,v}^2$ will not cause additional order in $d$.[13] Now we prove the result for the dominate term $I_1$.

**Lemma J.6.** *With probability* $1 - \delta$,

$$|I_1| \leq 4Hd\left\|\nabla f(\widehat{\theta}_h, \phi(s,a))\right\|_{\Sigma_h^{-1}} \cdot C_{\delta,\log K} + \widetilde{O}(\frac{\kappa_1}{\sqrt{\kappa}K}),$$

*where $C_{\delta,\log K}$ only contains Polylog terms.*

*Proof of Lemma J.6.* First of all, by CauchySchwarz inequality, we have

$$|I_1| \leq \left\|\nabla f(\widehat{\theta}_h, \phi(s,a))\right\|_{\Lambda_h^{-1}} \cdot \left\|\sum_{k=1}^K \frac{\left(f(\theta_{\mathbb{T}V_{h+1}^\star}, \phi_{h,k}) - r_{h,k} - V_{h+1}^\star(s_{h+1}^k)\right) \cdot \nabla_\theta^\top f(\widehat{\theta}_h, \phi_{h,k})}{\widehat{\sigma}_h^2(s_h^k, a_h^k)}\right\|_{\Lambda_h^{-1}}. \tag{33}$$

Recall that $\sigma_{u,v}^2(\cdot, \cdot) := \max\{1, f(v, \phi(\cdot, \cdot))_{[0,(H-h+1)^2]} - \left[f(u, \phi(\cdot, \cdot))_{[0,H-h+1]}\right]^2\}$.

---

[13]Note in Lemma L.11, we only have additive terms that has the same order has Lemma L.10.

**Step1.** Let the fixed $\theta \in \Theta$ be arbitrary and fixed $u, v$ such that $\sigma_{u,v}^2(\cdot, \cdot) \geq \frac{1}{2}\sigma_{\boldsymbol{u}_h^\star, \boldsymbol{v}_h^\star}^2(\cdot, \cdot) = \frac{1}{2}\sigma_h^{\star 2}(\cdot, \cdot)$ and define $x_k(\theta, u, v) = \nabla_\theta f(\theta, \phi_{h,k})/\sigma_{u,v}(s_h^k, a_h^k)$. Next, define $G_{u,v}(\theta) = \sum_{k=1}^K \nabla f(\theta, \phi(s_h^k, a_h^k)) \cdot \nabla f(\theta, \phi(s_h^k, a_h^k))^\top / \sigma_{u,v}^2(s_h^k, a_h^k) + \lambda I_d$, then $\|x_k\|_2 \leq \kappa_1$. Also denote $\eta_k := [f(\theta_{\mathbb{T}V_{h+1}^\star}, \phi_{h,k}) - r_{h,k} - V_{h+1}^\star(s_{h+1}^k)]/\sigma_{u,v}(s_h^k, a_h^k)$, then $\mathbb{E}[\eta_k | s_h^k, a_h^k] = 0$ and

$$\mathrm{Var}[\eta_k | s_h^k, a_h^k] = \frac{\mathrm{Var}[f(\theta_{\mathbb{T}V_{h+1}^\star}, \phi_{h,k}) - r_{h,k} - V_{h+1}^\star(s_{h+1}^k)|s_h^k, a_h^k]}{\sigma_{u,v}^2(s_h^k, a_h^k)}$$

$$\leq \frac{2\mathrm{Var}[f(\theta_{\mathbb{T}V_{h+1}^\star}, \phi_{h,k}) - r_{h,k} - V_{h+1}^\star(s_{h+1}^k)|s_h^k, a_h^k]}{\sigma_h^{\star 2}(s_h^k, a_h^k)}$$

$$= \frac{2[\mathrm{Var}_{P_h}V_{h+1}^\star](s_h^k, a_h^k)}{\sigma_h^{\star 2}(s_h^k, a_h^k)} \leq 2,$$

then by Self-normalized Bernstein's inequality (Lemma L.4), with probability $1 - \delta$,

$$\left\|\sum_{k=1}^K x_k(\theta, u, v)\eta_k\right\|_{G(\theta,u,v)^{-1}} \leq 16\sqrt{d\log\left(1 + \frac{K\kappa_1^2}{\lambda d}\right) \cdot \log\left(\frac{4K^2}{\delta}\right)} + 4\zeta\log\left(\frac{4K^2}{\delta}\right) \leq \widetilde{O}(\sqrt{d})$$

where $|\eta_k| \leq \zeta$ with $\zeta = 2\max_{s,a,s'} \frac{|f(\theta_{\mathbb{T}V_{h+1}^\star}, \phi(s,a)) - r - V_{h+1}^\star(s')|}{\sigma_h^\star(s,a)}$ and the last inequality uses $\sqrt{d} \geq \widetilde{O}(\zeta)$.

**Step2.** Define $h(\theta, u, v) := \sum_{k=1}^K x_k(\theta, u, v)\eta_k(u, v)$ and $H(\theta, u, v) := \|h(\theta, u, v)\|_{G_{u,v}(\theta)^{-1}}$,

$$\|h(\theta_1, u_1, v_1) - h(\theta_2, u_2, v_2)\|_2 \leq K \max_k \|(x_k \cdot \eta_k)(\theta_1, u_1, v_1) - (x_k \cdot \eta_k)(\theta_2, u_2, v_2)\|_2$$

$$\leq K \max_k \left\{ H \left|\frac{\nabla f(\theta_1, \phi_{h,k}) - \nabla f(\theta_2, \phi_{h,k})}{\sigma_{u_1,v_1}^2(s_h^k, a_h^k)}\right| + H\kappa_1 \left|\frac{\sigma_{u_1,v_1}^2(s_h^k, a_h^k) - \sigma_{u_2,v_2}^2(s_h^k, a_h^k)}{\sigma_{u_1,v_1}^2(s_h^k, a_h^k)\sigma_{u_2,v_2}^2(s_h^k, a_h^k)}\right| \right\}$$

$$\leq KH\kappa_1 \|\theta_1 - \theta_2\|_2 + KH\kappa_1 \|\sigma_{u_1,v_1}^2 - \sigma_{u_2,v_2}^2\|_2$$

Furthermore,

$$\left\|G_h(\theta_1, u_1, v_1)^{-1} - G_h(\theta_2, u_2, v_2)^{-1}\right\|_2 \leq \left\|G_h(\theta_1, u_1, v_1)^{-1}\right\|_2 \|G_h(\theta_1, u_1, v_1) - G_h(\theta_2, u_2, v_2)\|_2 \left\|G_h(\theta_2, u_2, v_2)^{-1}\right\|_2$$

$$\leq \frac{1}{\lambda^2} K \sup_k \left\|\frac{\nabla f(\theta_1, \phi_{h,k}) \cdot \nabla f(\theta_1, \phi_{h,k})^\top}{\sigma_{u_1,v_1}^2(s_h^k, a_h^k)} - \frac{\nabla f(\theta_2, \phi_{h,k}) \cdot \nabla f(\theta_2, \phi_{h,k})^\top}{\sigma_{u_2,v_2}^2(s_h^k, a_h^k)}\right\|_2$$

$$\leq \frac{1}{\lambda^2}\left(K\kappa_2\kappa_1 \|\theta_1 - \theta_2\|_2 + K\kappa_1^2 \|\sigma_{u_1,v_1}^2 - \sigma_{u_2,v_2}^2\|_2\right)$$

All the above imply

$$|H(\theta_1, u_1, v_1) - H(\theta_2, u_2, v_2)| \leq \sqrt{|h(\theta_1, u_1, v_1)^\top G_{u_1,v_1}(\theta_1)^{-1}h(\theta_1, u_1, v_1) - h(\theta_2, u_2, v_2)^\top G_{u_2,v_2}(\theta_2)^{-1}h(\theta_2, u_2, v_2)|}$$

$$\leq \sqrt{\|h(\theta_1, u_1, v_1) - h(\theta_2, u_2, v_2)\|_2 \cdot \frac{1}{\lambda} \cdot KH\kappa_1} + \sqrt{KH\kappa_1 \cdot \|G_{u_1,v_1}(\theta_1)^{-1} - G_{u_2,v_2}(\theta_2)^{-1}\|_2 \cdot KH\kappa_1}$$

$$+ \sqrt{(KH\kappa_1 \cdot \frac{1}{\lambda}) \cdot \|h(\theta_1, u_1, v_1) - h(\theta_2, u_2, v_2)\|_2}$$

$$\leq 2\sqrt{KH\kappa_1(\|\theta_1 - \theta_2\|_2 + \|\sigma_{u_1,v_1}^2 - \sigma_{u_2,v_2}^2\|_2) \cdot \frac{1}{\lambda} \cdot KH\kappa_1} + \sqrt{K^2H^2\kappa_1^2 \cdot \frac{K\kappa_1}{\lambda^2}\left(\kappa_2 \|\theta_1 - \theta_2\|_2 + \kappa_1 \|\sigma_{u_1,v_1}^2 - \sigma_{u_2,v_2}^2\|_2\right)}$$

$$\leq \left(\sqrt{4K^2H^2\kappa_1^2/\lambda} + \sqrt{K^3H^2\kappa_1^3\kappa_2/\lambda^2}\right)\sqrt{\|\theta_1 - \theta_2\|_2} + \left(\sqrt{4K^2H^2\kappa_1^2/\lambda} + \sqrt{K^3H^2\kappa_1^4/\lambda^2}\right)\sqrt{\|\sigma_{u_1,v_1}^2 - \sigma_{u_2,v_2}^2\|_2}$$

note

$$|\sigma_{u_1,v_1}^2(s,a) - \sigma_{u_2,v_2}^2(s,a)| \leq |f(v_1, \phi(s,a)) - f(v_2, \phi(s,a))| + 2H|f(u_1, \phi(s,a)) - f(u_2, \phi(s,a))|$$

$$\leq \kappa_1 \|v_1 - v_2\|_2 + 2H\kappa_1 \|u_1 - u_2\|_2,$$

Then a $\epsilon$-covering net of $\{H(\theta, u, v)\}$ can be constructed by the union of covering net for $\theta, u, v$ and by Lemma L.8, the covering number $\mathcal{N}_\epsilon$ satisfies (where $\widetilde{O}$ absorbs Polylog terms)

$$\log\mathcal{N}_\epsilon \leq \widetilde{O}(d)$$

**Step3.** First note by definition in Step2

$$\left\| \sum_{k=1}^{K} \frac{\left( f(\theta_{\mathbb{T}V_{h+1}^{\star}}, \phi_{h,k}) - r_{h,k} - V_{h+1}^{\star}(s_{h+1}^k) \right) \cdot \nabla_\theta^\top f(\widehat{\theta}_h, \phi_{h,k})}{\widehat{\sigma}_h^2(s_h^k, a_h^k)} \right\|_{\Lambda_h^{-1}} = H(\widehat{\theta}_h, \boldsymbol{u}_h, \boldsymbol{v}_h)$$

Now choosing $\epsilon = O(1/K)$ in Step2 and union bound over the covering number in Step2, we obtain with probability $1 - \delta$ (recall $\sqrt{d} \geq \widetilde{O}(\zeta)$),

$$H(\widehat{\theta}_h, \boldsymbol{u}_h, \boldsymbol{v}_h) \leq 16 \sqrt{d \log\left(1 + \frac{K\kappa_1^2}{\lambda d}\right) \cdot \left[\log\left(\frac{4K^2}{\delta}\right) + \widetilde{O}(d)\right] + 4\zeta[\log\left(\frac{4K^2}{\delta}\right) + \widetilde{O}(d)] + O(\frac{1}{K})}$$

$$\leq \widetilde{O}(d) + O(\frac{1}{K})$$

where we absorb all the Polylog terms. Combing above with equation 33, we obtain with probability $1 - \delta$,

$$|I_1| \leq \left\| \nabla f(\widehat{\theta}_h, \phi(s,a)) \right\|_{\Lambda_h^{-1}} \cdot H(\widehat{\theta}_h, \boldsymbol{u}_h, \boldsymbol{v}_h)$$

$$\leq \left\| \nabla f(\widehat{\theta}_h, \phi(s,a)) \right\|_{\Lambda_h^{-1}} \cdot \left[ \widetilde{O}(d) + O(\frac{1}{K}) \right]$$

$$\leq \widetilde{O}\left( d \left\| \nabla f(\widehat{\theta}_h, \phi(s,a)) \right\|_{\Lambda_h^{-1}} \right) + \widetilde{O}(\frac{\kappa_1}{\sqrt{\kappa}K}),$$

$\square$

Combing dominate term $I_1$ (via Lemma J.6) and all other higher order terms we can obtain the first result together with Lemma D.3.

The proof of the second result is also very similar to the proofs in Section F.2. Concretely, when picking $\pi = \pi^\star$, we can convert the quantity

$$\sqrt{\nabla_\theta^\top f(\widehat{\theta}_h, \phi(s_h, a_h))\Lambda_h^{-1}\nabla_\theta f(\widehat{\theta}_h, \phi(s_h, a_h))}$$

to

$$\sqrt{\nabla_\theta^\top f(\theta_h^\star, \phi(s_h, a_h))\Lambda_h^{-1}\nabla_\theta f(\theta_h^\star, \phi(s_h, a_h))}$$

using Theorem J.3, and convert

$$\sqrt{\nabla_\theta^\top f(\theta_h^\star, \phi(s_h, a_h))\Lambda_h^{-1}\nabla_\theta f(\theta_h^\star, \phi(s_h, a_h))}$$

to

$$\sqrt{\nabla_\theta^\top f(\theta_h^\star, \phi(s_h, a_h))\Lambda_h^{\star-1}\nabla_\theta f(\theta_h^\star, \phi(s_h, a_h))}$$

using Lemma J.4.

## K   THE LOWER BOUND

**Theorem K.1** (Restatement of Theorem 4.2). *Specifying the model to have linear representation $f = \langle \theta, \phi \rangle$. There exist a pair of universal constants $c, c' > 0$ such that given dimension $d$, horizon $H$ and sample size $K > c'd^3$, one can always find a family of MDP instances such that for any algorithm $\widehat{\pi}$*

$$\inf_{\widehat{\pi}} \sup_{M \in \mathcal{M}} \mathbb{E}_M \left[ v^\star - v^{\widehat{\pi}} \right] \geq c\sqrt{d} \cdot \sum_{h=1}^{H} \mathbb{E}_{\pi^\star} \left[ \sqrt{\nabla_\theta^\top f(\theta_h^\star, \phi(\cdot, \cdot))(\Lambda_h^{\star,p})^{-1}\nabla_\theta f(\theta_h^\star, \phi(\cdot, \cdot))} \right], \quad (34)$$

*where $\Lambda_h^{\star,p} = \mathbb{E}\left[ \sum_{k=1}^{K} \frac{\nabla_\theta f(\theta_h^\star, \phi(s_h^k, a_h^k)) \cdot \nabla_\theta f(\theta_h^\star, \phi(s_h^k, a_h^k))^\top}{\mathrm{Var}_h(V_{h+1}^\star)(s_h^k, a_h^k)} \right]$.*

**Remark K.2.** *Note Theorem 4.2 is a valid lower bound for comparison. This is because the upper bound result holds true for all model $f$ such that the corresponding $\mathcal{F}$ satisfies Assumption 2.1, 2.3. Therefore, for the lower bound construction it suffices to find one model $f$ such that the lower bound equation 34 holds. Here we simply choose the linear function approximation.*

### K.1 REGARDING THE PROOF OF LOWER BOUND

The proof of Theorem 4.2 can be done via a reduction to linear function approximation lower bound. In fact, it can be directly obtained from Theorem 3.5 of Yin et al. (2022), and the original proof comes from Theorem 2 of Zanette et al. (2021).

Concretely, all the proofs in Theorem 3.5 of Yin et al. (2022) follows and the only modification is to replace

$$\sqrt{\mathbb{E}_{\pi^\star}[\phi]^\top (\Lambda_h^\star)^{-1} \mathbb{E}_{\pi^\star}[\phi]} \le \frac{1}{2} \left\| \phi\left(+1, u_h\right)\right\|_{(\Lambda_h^{\star,p})^{-1}} + \frac{1}{2} \left\| \phi\left(-1, u_h\right)\right\|_{(\Lambda_h^{\star,p})^{-1}}$$

in Section E.5 by

$$\mathbb{E}_{\pi^\star}\left[ \sqrt{\phi(\cdot,\cdot)^\top (\Lambda_h^{\star,p})^{-1}\phi(\cdot,\cdot)} \right] = \frac{1}{2}\left\| \phi(+1, u_h)\right\|_{(\Lambda_h^{\star,p})^{-1}} + \frac{1}{2}\left\| \phi(-1, u_h)\right\|_{(\Lambda_h^{\star,p})^{-1}},$$

and the final result holds with $\phi(\cdot,\cdot) = \nabla_\theta f(\theta_h^\star, \phi(\cdot,\cdot))$ by the reduction $f = \langle \theta, \phi \rangle$.

## L    AUXILIARY LEMMAS

**Lemma L.1** ($k$-th Order Mean Value Form of Taylor's Expansion). *Let $k \ge 1$ be an integer and let function $f : \mathbb{R}^d \to \mathbb{R}$ be $k$ times differentiable and continuous over the compact domain $\Theta \subset \mathbb{R}^d$. Then for any $x, \theta \in \Theta$, there exists $\xi$ in the line segment of $x$ and $\theta$, such that*

$$f(x) - f(\theta) = \nabla f(\theta)^\top (x - \theta) + \frac{1}{2!}(x-\theta)^\top \nabla^2_{\theta\theta} f(\theta)(x-\theta) + \ldots + \frac{1}{(k-1)!}\nabla^{k-1} f(\theta) \left( \bigotimes (x-\theta) \right)^{k-1}$$

$$+ \frac{1}{k!}\nabla^k f(\xi) \left( \bigotimes (x-\theta) \right)^{k}.$$

*Here $\nabla^k f(\theta)$ denotes $k$-dimensional tensor and $\bigotimes$ denotes tensor product.*

**Lemma L.2** (Vector Hoeffding's Inequality). *Let $X = (X_1, \ldots, X_d)$ be $d$-dimensional vector Random Variable with $E[X] = 0$ and $\|X\|_2 \le R$. $X^{(1)}, \ldots, X^{(n)}$'s are $n$ samples. Then with probability $1 - \delta$,*

$$\left\| \frac{1}{n}\sum_{i=1}^{n} X^{(i)} \right\|_2 \le \sqrt{\frac{4dR^2}{n}\log(\frac{d}{\delta})}.$$

*Proof of Lemma L.2.* Since $\|X\|_2 \le R$ implies $|X_j| \le R$, by the univariate Hoeffding's inequality, for a fixed $j \in \{1, ..., d\}$, denote $Y_j := \frac{1}{n}\sum_{i=1}^{n} X_j^{(i)}$. Then with probability $1-\delta$ (note $|X_j^{(i)}| \le R$),

$$\mathbb{P}\left( |Y_j| \ge 2\sqrt{\frac{R^2}{n}\log(\frac{1}{\delta})} \right) \le \delta.$$

By a union bound,

$$\mathbb{P}\left( \exists i \ s.t. \ |Y_j| \ge 2\sqrt{\frac{R^2}{n}\log(\frac{1}{\delta})} \right) \le d\delta \Leftrightarrow \mathbb{P}\left( \forall i \ \ |Y_j| \le 2\sqrt{\frac{R^2}{n}\log(\frac{1}{\delta})} \right) \ge 1 - d\delta$$

$$\Leftrightarrow \mathbb{P}\left( \forall i \ \ Y_j^2 \le \frac{4R^2}{n}\log(\frac{1}{\delta}) \right) \ge 1 - d\delta \Rightarrow \mathbb{P}\left( \|Y\|_2 \le \sqrt{\frac{4dR^2}{n}\log(\frac{1}{\delta})} \right) \ge 1 - d\delta$$

$$\Leftrightarrow \mathbb{P}\left( \|Y\|_2 \le \sqrt{\frac{4dR^2}{n}\log(\frac{d}{\delta})} \right) \ge 1 - \delta.$$

$\square$

**Lemma L.3** (Hoeffding inequality for self-normalized martingales (Abbasi-Yadkori et al., 2011)). *Let $\{\eta_t\}_{t=1}^{\infty}$ be a real-valued stochastic process. Let $\{\mathcal{F}_t\}_{t=0}^{\infty}$ be a filtration, such that $\eta_t$ is $\mathcal{F}_t$-measurable. Assume $\eta_t$ also satisfies $\eta_t$ given $\mathcal{F}_{t-1}$ is zero-mean and $R$-subgaussian, i.e.*

$$\forall \lambda \in \mathbb{R}, \quad \mathbb{E}\left[ e^{\lambda \eta_t} \mid \mathcal{F}_{t-1} \right] \le e^{\lambda^2 R^2 / 2}$$

Let $\{x_t\}_{t=1}^\infty$ be an $\mathbb{R}^d$-valued stochastic process where $x_t$ is $\mathcal{F}_{t-1}$ measurable and $\|x_t\| \le L$. Let $\Lambda_t = \lambda I_d + \sum_{s=1}^t x_s x_s^\top$. Then for any $\delta > 0$, with probability $1 - \delta$, for all $t > 0$,

$$\left\| \sum_{s=1}^t x_s \eta_s \right\|_{\Lambda_t^{-1}}^2 \le 8R^2 \cdot \frac{d}{2} \log\left( \frac{\lambda + tL}{\lambda \delta} \right).$$

**Lemma L.4** (Bernstein inequality for self-normalized martingales (Zhou et al., 2021a)). *Let $\{\eta_t\}_{t=1}^\infty$ be a real-valued stochastic process. Let $\{\mathcal{F}_t\}_{t=0}^\infty$ be a filtration, such that $\eta_t$ is $\mathcal{F}_t$-measurable. Assume $\eta_t$ also satisfies*

$$|\eta_t| \le R, \mathbb{E}[\eta_t \mid \mathcal{F}_{t-1}] = 0, \mathbb{E}[\eta_t^2 \mid \mathcal{F}_{t-1}] \le \sigma^2.$$

*Let $\{x_t\}_{t=1}^\infty$ be an $\mathbb{R}^d$-valued stochastic process where $x_t$ is $\mathcal{F}_{t-1}$ measurable and $\|x_t\| \le L$. Let $\Lambda_t = \lambda I_d + \sum_{s=1}^t x_s x_s^\top$. Then for any $\delta > 0$, with probability $1 - \delta$, for all $t > 0$,*

$$\left\| \sum_{s=1}^t \mathbf{x}_s \eta_s \right\|_{\boldsymbol{\Lambda}_t^{-1}} \le 8\sigma \sqrt{ d \log\left( 1 + \frac{tL^2}{\lambda d} \right) \cdot \log\left( \frac{4t^2}{\delta} \right) } + 4R \log\left( \frac{4t^2}{\delta} \right)$$

**Lemma L.5.** *Let $\nabla f(\theta, \phi(\cdot, \cdot)) : \mathcal{S} \times \mathcal{A} \to \mathbb{R}^d$ be a bounded function s.t. $\sup_{\theta \in \Theta} \|\nabla f(\theta, \phi(\cdot, \cdot))\|_2 \le \kappa_1$. If $K$ satisfies*

$$K \ge \max\left\{ 512 \frac{\kappa_1^4}{\kappa^2} \left( \log(\frac{2d}{\delta}) + d \log(1 + \frac{4\kappa_1 B^2 \kappa_2 C_\Theta K^3}{\lambda^2}) \right), \frac{4\lambda}{\kappa} \right\}$$

*Then with probability at least $1 - \delta$, for all $\|u\|_2 \le B$ simultaneously, it holds that*

$$\|u\|_{\Sigma_h^{-1}} \le \frac{2B}{\sqrt{\kappa K}} + O(\frac{1}{K})$$

*where $\Sigma_h = \sum_{k=1}^K \nabla f(\widehat{\theta}_h, \phi(s_h^k, a_h^k)) \cdot \nabla f(\widehat{\theta}_h, \phi(s_h^k, a_h^k))^\top + \lambda I_d$.*

*Proof of Lemma L.5.* For a fixed $\theta$, define $\bar{G} = \sum_{k=1}^K \nabla f(\theta, \phi(s_h^k, a_h^k)) \cdot \nabla f(\theta, \phi(s_h^k, a_h^k))^\top + \lambda I_d$, and $G = \mathbb{E}_\mu[\nabla f(\theta, \phi(s_h, a_h)) \cdot \nabla f(\theta, \phi(s_h, a_h))^\top]$, then by Lemma H.5. of Min et al. (2021), as long as

$$K \ge \max\{512\kappa_1^4 \left\| G^{-1} \right\|^2 \log(\frac{2d}{\delta}), 4\lambda \left\| G^{-1} \right\|_2\}, \tag{35}$$

then with probability $1 - \delta$, for all $u \in \mathbb{R}^d$ simultaneously, $\|u\|_{\bar{G}^{-1}} \le \frac{2}{\sqrt{K}} \|u\|_{G^{-1}}$. As a corollary, if we constraint $u$ to the subspace $\|u\|_2 \le B$, then we have: with probability $1 - \delta$, for all $\{u \in \mathbb{R}^d : \|u\|_2 \le B\}$ simultaneously,

$$\|u\|_{\bar{G}^{-1}} \le \frac{2}{\sqrt{K}} \|u\|_{G^{-1}} = \frac{2}{\sqrt{K}} \sqrt{u^\top G^{-1} u} \le \frac{2B\sqrt{\|G^{-1}\|_2}}{\sqrt{K}}. \tag{36}$$

Next, for any $\theta$, define

$$h_u(\theta) := \|u\|_{\bar{G}^{-1}} = \sqrt{u^\top \bar{G}^{-1} u} = \sqrt{ u^\top \left( \sum_{k=1}^K \nabla f(\theta, \phi(s_h^k, a_h^k)) \cdot \nabla f(\theta, \phi(s_h^k, a_h^k))^\top + \lambda I_d \right)^{-1} u }$$

and $\bar{G}(\theta) = \sum_{k=1}^K \nabla f(\theta, \phi(s_h^k, a_h^k)) \cdot \nabla f(\theta, \phi(s_h^k, a_h^k))^\top + \lambda I_d$, we have for any $\theta_1, \theta_2$

$$\begin{aligned}
\left\| \bar{G}(\theta_1) - \bar{G}(\theta_2) \right\|_2 &\le \left\| \sum_{k=1}^K \left( \nabla f(\theta_1, \phi(s_h^k, a_h^k)) - \nabla f(\theta_2, \phi(s_h^k, a_h^k)) \right) \cdot \nabla f(\theta_1, \phi(s_h^k, a_h^k))^\top \right\| \\
&\quad + \left\| \sum_{k=1}^K \nabla f(\theta_2, \phi(s_h^k, a_h^k)) \left( \nabla f(\theta_1, \phi(s_h^k, a_h^k)) - \nabla f(\theta_2, \phi(s_h^k, a_h^k)) \right)^\top \right\| \\
&\le K\kappa_2 \kappa_1 \|\theta_1 - \theta_2\|_2 + K\kappa_2 \kappa_1 \|\theta_1 - \theta_2\|_2 \le 2K\kappa_2 \kappa_1 \|\theta_1 - \theta_2\|_2.
\end{aligned}$$

Use the basic inequality for $a, b > 0 \Rightarrow |\sqrt{a} - \sqrt{b}| \le \sqrt{|a - b|}$,

$$\sup_u |h_u(\theta_1) - h_u(\theta_2)| \le \sup_u \sqrt{\left|u^\top \left(\bar{G}(\theta_1)^{-1} - \bar{G}(\theta_2)^{-1}\right) u\right|} \le \sqrt{B^2 \cdot \left\|\bar{G}(\theta_1)^{-1} - \bar{G}(\theta_2)^{-1}\right\|_2}$$

$$\le \sqrt{B^2 \cdot \left\|\bar{G}(\theta_1)^{-1}\right\|_2 \left\|\bar{G}(\theta_1) - \bar{G}(\theta_2)\right\|_2 \left\|\bar{G}(\theta_2)^{-1}\right\|_2}$$

$$\le \sqrt{B^2 \frac{1}{\lambda} 2K\kappa_2\kappa_1 \|\theta_1 - \theta_2\|_2 \frac{1}{\lambda}} = \sqrt{\frac{2B^2 K\kappa_1\kappa_2 \|\theta_1 - \theta_2\|_2}{\lambda^2}}$$

Therefore, the $\epsilon$-covering net of $\{h(\theta) : \theta \in \Theta\}$ is implies by the $\frac{\lambda^2\epsilon^2}{2KB^2\kappa_1\kappa_2}$-covering net of $\{\theta : \theta \in \Theta\}$, so by Lemma L.8, the covering number $\mathcal{N}_\epsilon$ satisfies

$$\log \mathcal{N}_\epsilon \le d \log(1 + \frac{4B^2 K\kappa_1\kappa_2 C_\Theta}{\lambda^2\epsilon^2}).$$

Select $\theta = \widehat{\theta}_h$. Choose $\epsilon = O(1/K)$ and by a union bound over equation 36 to get with probability $1 - \delta$, for all $\|u\|_2 \le B$ (note By Assumption 2.3 $\|G^{-1}\|_2 \le 1/\kappa$),

$$\|u\|_{\Sigma_h^{-1}} \le \frac{2B}{\sqrt{\kappa K}} + O(\frac{1}{K})$$

if (union bound over the condition equation 35)

$$K \ge \max\left\{512\frac{\kappa_1^4}{\kappa^2}\left(\log(\frac{2d}{\delta}) + d\log(1 + \frac{4\kappa_1 B^2\kappa_2 C_\Theta K^3}{\lambda^2})\right), \frac{4\lambda}{\kappa}\right\}$$

where this condition is satisfied by the Lemma statement.

$\square$

**Lemma L.6.** *let* $\phi : \mathcal{S} \times \mathcal{A} \to \mathbb{R}^d$ *satisfies* $\|\phi(s, a)\| \le C$ *for all* $s, a \in \mathcal{S} \times \mathcal{A}$. *For any* $K > 0, \lambda > 0$, *define* $\bar{G}_K = \sum_{k=1}^K \phi(s_k, a_k)\phi(s_k, a_k)^\top + \lambda I_d$ *where* $(s_k, a_k)$'s *are i.i.d samples from some distribution* $\nu$. *Then with probability* $1 - \delta$,

$$\left\|\frac{\bar{G}_K}{K} - \mathbb{E}_\nu\left[\frac{\bar{G}_K}{K}\right]\right\| \le \frac{4\sqrt{2}C^2}{\sqrt{K}}\left(\log\frac{2d}{\delta}\right)^{1/2}.$$

*Proof of Lemma L.6.* See Lemma H.5 of Yin et al. (2022) or Lemma H.4 of Lemma Min et al. (2021) for details. $\square$

**Lemma L.7** (Lemma H.4 in Yin et al. (2022))**.** *Let* $\Lambda_1$ *and* $\Lambda_2 \in \mathbb{R}^{d \times d}$ *are two positive semi-definite matrices. Then:*

$$\left\|\Lambda_1^{-1}\right\| \le \left\|\Lambda_2^{-1}\right\| + \left\|\Lambda_1^{-1}\right\| \cdot \left\|\Lambda_2^{-1}\right\| \cdot \|\Lambda_1 - \Lambda_2\|$$

*and*

$$\|\phi\|_{\Lambda_1^{-1}} \le \left[1 + \sqrt{\left\|\Lambda_2^{-1}\right\| \|\Lambda_2\| \cdot \left\|\Lambda_1^{-1}\right\| \cdot \|\Lambda_1 - \Lambda_2\|}\right] \cdot \|\phi\|_{\Lambda_2^{-1}}.$$

*for all* $\phi \in \mathbb{R}^d$.

### L.1 COVERING ARGUMENTS

**Lemma L.8.** *(Covering Number of Euclidean Ball) For any* $\epsilon > 0$, *the* $\epsilon$-*covering number of the Euclidean ball in* $\mathbb{R}^d$ *with radius* $R > 0$ *is upper bounded by* $(1 + 2R/\epsilon)^d$.

**Lemma L.9.** *Define* $\mathcal{V}$ *to be the class mapping* $\mathcal{S}$ *to* $\mathbb{R}$ *with the parametric form*

$$V(\cdot) := \min\{\max_a f(\theta, \phi(\cdot, a)) - \sqrt{\nabla f(\theta, \phi(\cdot, a))^\top A \cdot \nabla f(\theta, \phi(\cdot, a))}, H\}$$

*where the parameter spaces are* $\{\theta : \|\theta\|_2 \le C_\Theta\}$ *and* $\{A : \|A\|_2 \le B\}$. *Let* $\mathcal{N}_\epsilon^\mathcal{V}$ *be the covering number of* $\epsilon$-*net with respect to* $l_\infty$ *distance, then we have*

$$\log \mathcal{N}_\epsilon^\mathcal{V} \le d \log\left(1 + \frac{8C_\Theta(\kappa_1\sqrt{C_\Theta} + 2\sqrt{B}\kappa_1\kappa_2)^2}{\epsilon^2}\right) + d^2 \log\left(1 + \frac{8\sqrt{d}B\kappa_1^2}{\epsilon^2}\right).$$

*Proof of Lemma L.9.*

$$\sup_s |V_1(s) - V_2(s)|$$

$$\leq \sup_{s,a} \left| f(\theta_1, \phi(s,a)) - \sqrt{\nabla f(\theta_1, \phi(s,a))^\top A_1 \cdot \nabla f(\theta_1, \phi(s,a))} - f(\theta_2, \phi(s,a)) + \sqrt{\nabla f(\theta_2, \phi(s,a))^\top A_2 \cdot \nabla f(\theta_2, \phi(s,a))} \right|$$

$$= \sup_{s,a} \left| \nabla f(\xi, \phi(s,a)) \cdot (\theta_1 - \theta_2) - \sqrt{\nabla f(\theta_1, \phi(s,a))^\top A_1 \cdot \nabla f(\theta_1, \phi(s,a))} + \sqrt{\nabla f(\theta_2, \phi(s,a))^\top A_2 \cdot \nabla f(\theta_2, \phi(s,a))} \right|$$

$$\leq \kappa_1 \cdot \|\theta_1 - \theta_2\|_2 + \sup_{s,a} \left| \sqrt{\nabla f(\theta_1, \phi(s,a))^\top A_1 \cdot \nabla f(\theta_1, \phi(s,a))} - \sqrt{\nabla f(\theta_2, \phi(s,a))^\top A_2 \cdot \nabla f(\theta_2, \phi(s,a))} \right|$$

$$\leq \kappa_1 \cdot \|\theta_1 - \theta_2\|_2 + \sup_{s,a} \sqrt{|[\nabla f(\theta_1, \phi(s,a)) - \nabla f(\theta_2, \phi(s,a))]^\top A_1 \cdot \nabla f(\theta_1, \phi(s,a))|}$$

$$+ \sup_{s,a} \sqrt{|\nabla f(\theta_2, \phi(s,a))^\top (A_1 - A_2) \cdot \nabla f(\theta_1, \phi(s,a))|} + \sup_{s,a} \sqrt{|\nabla f(\theta_2, \phi(s,a))^\top A_2 \cdot [\nabla f(\theta_1, \phi(s,a)) - \nabla f(\theta_2, \phi(s,a))]|}$$

$$\leq \kappa_1 \cdot \|\theta_1 - \theta_2\|_2 + 2\sup_{s,a} \sqrt{\|\nabla f(\theta_1, \phi(s,a)) - \nabla f(\theta_2, \phi(s,a))\|_2 \cdot B \cdot \kappa_1} + \sqrt{\kappa_1^2 \|A_1 - A_2\|_2}$$

$$\leq \kappa_1 \cdot \|\theta_1 - \theta_2\|_2 + 2\sup_{s,a} \sqrt{\|\nabla f(\theta_1, \phi(s,a)) - \nabla f(\theta_2, \phi(s,a))\|_2 \cdot B \cdot \kappa_1} + \sqrt{\kappa_1^2 \|A_1 - A_2\|_2}$$

$$\leq \kappa_1 \cdot \|\theta_1 - \theta_2\|_2 + 2\sup_{s,a} \sqrt{\|\nabla f(\theta_1, \phi(s,a))\|_2 \cdot \|\theta_1 - \theta_2\|_2 \cdot B \cdot \kappa_1} + \sqrt{\kappa_1^2 \|A_1 - A_2\|_2}$$

$$\leq \kappa_1 \cdot \|\theta_1 - \theta_2\|_2 + 2\sqrt{\kappa_2 \cdot \|\theta_1 - \theta_2\|_2 \cdot B \cdot \kappa_1} + \sqrt{\kappa_1^2 \|A_1 - A_2\|_2}$$

$$\leq \left( \kappa_1 \sqrt{C_\Theta} + 2\sqrt{B\kappa_1\kappa_2} \right) \sqrt{\|\theta_1 - \theta_2\|_2} + \kappa_1 \sqrt{\|A_1 - A_2\|_2} \leq \left( \kappa_1 \sqrt{C_\Theta} + 2\sqrt{B\kappa_1\kappa_2} \right) \sqrt{\|\theta_1 - \theta_2\|_2} + \kappa_1 \sqrt{\|A_1 - A_2\|_F}$$

Here $\|\cdot\|_F$ is Frobenius norm. Let $\mathcal{C}_\theta$ be the $\frac{\epsilon^2}{4(\kappa_1\sqrt{C_\Theta}+2\sqrt{B\kappa_1\kappa_2})^2}$-net of space $\{\theta : \|\theta\|_2 \leq C_\Theta\}$ and $\mathcal{C}_w$ be the $\frac{\epsilon^2}{4\kappa_1^2}$-net of the space $\{A : \|A\|_F \leq \sqrt{d}B\}$, then by Lemma L.8,

$$|\mathcal{C}_w| \leq \left( 1 + \frac{8C_\Theta(\kappa_1\sqrt{C_\Theta} + 2\sqrt{B\kappa_1\kappa_2})^2}{\epsilon^2} \right)^d, \quad |\mathcal{C}_A| \leq \left( 1 + \frac{8\sqrt{d}B\kappa_1^2}{\epsilon^2} \right)^{d^2}$$

Therefore, the covering number of space $\mathcal{V}$ satisfies

$$\log \mathcal{N}_\epsilon^\mathcal{V} \leq \log(|\mathcal{C}_w| \cdot |\mathcal{C}_A|) \leq d \log \left( 1 + \frac{8C_\Theta(\kappa_1\sqrt{C_\Theta} + 2\sqrt{B\kappa_1\kappa_2})^2}{\epsilon^2} \right) + d^2 \log \left( 1 + \frac{8\sqrt{d}B\kappa_1^2}{\epsilon^2} \right)$$

$$\square$$

**Lemma L.10** (Covering of $\mathbb{E}_\mu(X(g, V, f))$). *Define*

$$X(\theta, \theta') := (f(\theta, \phi(s,a)) - r - V_{\theta'}(s'))^2 - (f_{V_{\theta'}}(s,a) - r - V_{\theta'}(s'))^2,$$

*where $f_V := \mathcal{P}_h V + \delta_V$ and $V(s)$ has form $V_\theta(s)$ that belongs to $\mathcal{V}$ (as defined in Lemma L.9). Here $X(\theta, \theta')$ is a function of $s, a, r, s'$ as well, and we suppress the notation for conciseness only. Then the function class $\mathcal{H} = \{h(\theta, \theta') := \mathbb{E}_\mu[X(\theta, \theta')]| \|\theta\|_2 \leq C_\Theta, V_\theta \in \mathcal{V}\}$ has the covering number of $(\epsilon + 4H\epsilon_\mathcal{F})$-net bounded by*

$$d \log(1 + \frac{24C_\Theta(H+1)\kappa_1}{\epsilon}) + d \log \left( 1 + \frac{288H^2 C_\Theta(\kappa_1\sqrt{C_\Theta} + 2\sqrt{B\kappa_1\kappa_2})^2}{\epsilon^2} \right) + d^2 \log \left( 1 + \frac{288H^2\sqrt{d}B\kappa_1^2}{\epsilon^2} \right).$$

*Proof of Lemma L.10.* First of all,

$$X(\theta, \theta') = f(\theta, \phi(s,a))^2 - f_{V_{\theta'}}(s,a)^2 - 2f(\theta, \phi(s,a)) \cdot (r + V_{\theta'}(s')) + 2f_{V_{\theta'}}(s,a) \cdot (r + V_{\theta'}(s')),$$

For any $(\theta_1, \theta_1'), (\theta_2, \theta_2')$,

$$|X(\theta_1, \theta_1') - X(\theta_2, \theta_2')| \leq |f(\theta_1, \phi(s,a))^2 - f(\theta_2, \phi(s,a))^2|$$

$$+|f_{V_{\theta_1'}}(s,a)^2 - f_{V_{\theta_2'}}(s,a)^2| + 2|f_{V_{\theta_1'}}(s,a) - f_{V_{\theta_2'}}(s,a)| \cdot (r + V_{\theta_1'}(s'))$$

$$+2f_{V_{\theta_2'}}(s,a) \cdot |V_{\theta_1'}(s') - V_{\theta_2'}(s')| + 2|f(\theta_1, \phi(s,a)) - f(\theta_2, \phi(s,a))| \cdot (r + V_{\theta_1'}(s'))$$

$$+2|f(\theta_2, \phi(s,a))| \cdot |V_{\theta_1'}(s') - V_{\theta_2'}(s')|$$

$$\leq 2H \cdot |f(\theta_1, \phi(s,a)) - f(\theta_2, \phi(s,a))| + 2H \cdot |f_{V_{\theta_1'}}(s,a) - f_{V_{\theta_2'}}(s,a)|$$

$$+4H \cdot |V_{\theta_1'}(s') - V_{\theta_2'}(s')| + 4(H+1) \cdot |f(\theta_1, \phi(s,a)) - f(\theta_2, \phi(s,a))|$$

$$\leq (6H+1) \cdot |f(\theta_1, \phi(s,a)) - f(\theta_2, \phi(s,a))| + 2H \max_{s'} |V_{\theta_1'}(s') - V_{\theta_2'}(s')| + 4H\epsilon_{\mathcal{F}}$$

$$+4H \cdot |V_{\theta_1'}(s') - V_{\theta_2'}(s')|$$

$$\leq (6H+1)\|\nabla f(\xi, \phi(s,a))\|_2 \cdot \|\theta_1 - \theta_2\|_2 + 6H \|V_{\theta_1'} - V_{\theta_2'}\|_\infty + 4H\epsilon_{\mathcal{F}}$$

$$\leq (6H+1)\kappa_1 \cdot \|\theta_1 - \theta_2\|_2 + 6H \|V_{\theta_1'} - V_{\theta_2'}\|_\infty + 4H\epsilon_{\mathcal{F}}$$

where the second inequality comes from $f_V = \mathcal{P}_h V + \delta_V$. Note the above holds true for all $s, a, r, s'$, therefore it implies

$$|\mathbb{E}_\mu[X(\theta_1, \theta_1')] - \mathbb{E}_\mu[X(\theta_2, \theta_2')]| \leq \sup_{s,a,s'} |X(\theta_1, \theta_1') - X(\theta_2, \theta_2')|$$

$$\leq (6H+1)\kappa_1 \cdot \|\theta_1 - \theta_2\|_2 + 6H \|V_{\theta_1'} - V_{\theta_2'}\|_\infty + 4H\epsilon_{\mathcal{F}}$$

Now let $\mathcal{C}_1$ be the $\frac{\epsilon}{12(H+1)\kappa_1}$-net of $\{\theta : \|\theta\|_2 \leq C_\Theta\}$ and $\mathcal{C}_2$ be the $\epsilon/6H$-net of $\mathcal{V}$, applying Lemma L.8 and Lemma L.9 to obtain

$$\log|\mathcal{C}_1| \leq d\log(1 + \frac{24C_\Theta(H+1)\kappa_1}{\epsilon}), \quad \log|\mathcal{C}_2| \leq d\log\left(1 + \frac{288H^2 C_\Theta(\kappa_1\sqrt{C_\Theta} + 2\sqrt{B\kappa_1\kappa_2})^2}{\epsilon^2}\right) + d^2\log\left(1 + \frac{288H^2\sqrt{d}B\kappa_1^2}{\epsilon^2}\right)$$

which implies the covering number of $\mathcal{H}$ to be bounded by

$$\log|\mathcal{C}_1| \cdot |\mathcal{C}_2| \leq d\log(1 + \frac{24C_\Theta(H+1)\kappa_1}{\epsilon}) + d\log\left(1 + \frac{288H^2 C_\Theta(\kappa_1\sqrt{C_\Theta} + 2\sqrt{B\kappa_1\kappa_2})^2}{\epsilon^2}\right) + d^2\log\left(1 + \frac{288H^2\sqrt{d}B\kappa_1^2}{\epsilon^2}\right).$$

$\square$

**Lemma L.11.** *Denote* $\sigma_{u,v}^2(\cdot, \cdot) := \max\{1, f(v, \phi(\cdot, \cdot))_{[0, (H-h+1)^2]} - [f(u, \phi(\cdot, \cdot))_{[0, H-h+1]}]^2\}$ *and define*

$$\bar{X}(\theta, \theta', u, v) := \frac{(f(\theta, \phi(s,a)) - r - V_{\theta'}(s'))^2 - (f_{V_{\theta'}}(s,a) - r - V_{\theta'}(s'))^2}{\sigma_{u,v}^2(s,a)},$$

*where* $f_V := \mathcal{P}_h V$ *and* $V(s)$ *has form* $V_\theta(s)$ *that belongs to* $\mathcal{V}$ *(as defined in Lemma L.9). Here* $\bar{X}(\theta, \theta', u, v)$ *is a function of* $s, a, r, s'$ *as well, and we suppress the notation for conciseness only. Then the function class* $\mathcal{H} = \{h(\theta, \theta', u, v) := \mathbb{E}_\mu[\bar{X}(\theta, \theta', u, v)] \mid \|\theta\|_2 \leq C_\Theta, V_\theta \in \mathcal{V}\}$ *has the covering number of* $\epsilon$-*net bounded by*

$$d\log(1 + \frac{24C_\Theta(H+1)\kappa_1}{\epsilon}) + d\log\left(1 + \frac{288H^2 C_\Theta(\kappa_1\sqrt{C_\Theta} + 2\sqrt{B\kappa_1\kappa_2})^2}{\epsilon^2}\right) + d^2\log\left(1 + \frac{288H^2\sqrt{d}B\kappa_1^2}{\epsilon^2}\right)$$

$$+ d\log(1 + \frac{16C_\Theta H^2\kappa_1}{\epsilon}) + d\log(1 + \frac{32C_\Theta H^3\kappa_1}{\epsilon})$$

*Proof of Lemma L.11.* Recall $\sigma_{u,v}^2(\cdot, \cdot) := \max\{1, f(v, \phi(\cdot, \cdot))_{[0, (H-h+1)^2]} - [f(u, \phi(\cdot, \cdot))_{[0, H-h+1]}]^2\}$, and since $\max$, truncation are non-expansive operations, then we can achieve for any $s, a$

$$|\sigma_{u_1, v_1}^2(s,a) - \sigma_{u_2, v_2}^2(s,a)| \leq |f(v_1, \phi(s,a)) - f(v_2, \phi(s,a))| + 2H |f(u_1, \phi(s,a)) - f(u_2, \phi(s,a))|$$

$$\leq \kappa_1 \|v_1 - v_2\|_2 + 2H\kappa_1 \|u_1 - u_2\|_2,$$

Hence

$$
\left| \bar{X}(\theta_1, \theta_1', u_1, v_1) - \bar{X}(\theta_2, \theta_2', u_2, v_2) \right| = \left| \frac{X(\theta_1, \theta_1')}{\sigma_{u_1, v_1}^2} - \frac{X(\theta_2, \theta_2')}{\sigma_{u_2, v_2}^2} \right|
$$

$$
\leq \left| \frac{X(\theta_1, \theta_1') - X(\theta_2, \theta_2')}{\sigma_{u_1, v_1}^2} \right| + \left| \frac{X(\theta_2, \theta_2')}{\sigma_{u_1, v_1}^2 \sigma_{u_2, v_2}^2} \left( \sigma_{u_1, v_1}^2 - \sigma_{u_2, v_2}^2 \right) \right|
$$

$$
\leq |X(\theta_1, \theta_1') - X(\theta_2, \theta_2')| + 2H^2 \left| \sigma_{u_1, v_1}^2 - \sigma_{u_2, v_2}^2 \right|
$$

$$
\leq |X(\theta_1, \theta_1') - X(\theta_2, \theta_2')| + 2H^2 \kappa_1 \|v_1 - v_2\|_2 + 4H^3 \kappa_1 \|u_1 - u_2\|_2
$$

$$
\leq (6H + 1)\kappa_1 \cdot \|\theta_1 - \theta_2\|_2 + 6H \left\| V_{\theta_1'} - V_{\theta_2'} \right\|_\infty + 2H^2 \kappa_1 \|v_1 - v_2\|_2 + 4H^3 \kappa_1 \|u_1 - u_2\|_2
$$

Note the above holds true for all $s, a, r, s'$, therefore it implies

$$
\left| \mathbb{E}_\mu[\bar{X}(\theta_1, \theta_1', u_1, v_1)] - \mathbb{E}_\mu[\bar{X}(\theta_2, \theta_2', u_2, v_2)] \right|
$$

$$
\leq (6H + 1)\kappa_1 \cdot \|\theta_1 - \theta_2\|_2 + 6H \left\| V_{\theta_1'} - V_{\theta_2'} \right\|_\infty + 2H^2 \kappa_1 \|v_1 - v_2\|_2 + 4H^3 \kappa_1 \|u_1 - u_2\|_2
$$

and similar to Lemma L.10, the covering number of $\epsilon$-net will be bounded by

$$
d \log(1 + \frac{24 C_\Theta (H+1)\kappa_1}{\epsilon}) + d \log \left( 1 + \frac{288 H^2 C_\Theta (\kappa_1 \sqrt{C_\Theta} + 2\sqrt{B}\kappa_1 \kappa_2)^2}{\epsilon^2} \right) + d^2 \log \left( 1 + \frac{288 H^2 \sqrt{d} B \kappa_1^2}{\epsilon^2} \right)
$$

$$
+ d \log(1 + \frac{16 C_\Theta H^2 \kappa_1}{\epsilon}) + d \log(1 + \frac{32 C_\Theta H^3 \kappa_1}{\epsilon})
$$

Comparing to Lemma L.10, the last two terms are incurred by covering $u, v$ arguments.

$\square$

