# OpenReview forum: "Offline Reinforcement Learning with Differentiable Function Approximation is Provably Efficient"
_ICLR.cc/2023/Conference — ICLR 2023 poster_

### Official Review · Reviewer_tREF · 2022-10-21

**Confidence:** 3
**Correctness:** 4
**Technical Novelty And Significance:** 3
**Empirical Novelty And Significance:** 2
**Recommendation:** 6

**Clarity, Quality, Novelty And Reproducibility:**

Quality: This paper is of good quality overall.

Clarity: The results are clearly stated and related to the existing literature.

Originality: Although the FDA setting seems less studied, I have the impression that the technical novelty of this paper is limited, or at least the authors need to highlight such novelty more.

**Details Of Ethics Concerns:**

I have not ethics concerns.

**Strength And Weaknesses:**

Strength:

1. This paper provides a new algorithm and finite-sample analysis for offline RL with DFA. The theory seems sound because the main arguments look correct to me, although the analysis is too overwhelming hence I did not check all the details.

2. This paper provides a comprehensive discussion of the application of existing results to the DFA setting, including the pessimistic value iteration and its variance reduction results.

3. The authors did a good job in positing this paper in the related literature, and include detailed discussion on the implications, making this paper fruitful and dense.

Weakness:

1. All the results seem to be generalizations of existing algorithms and theoretical results. It may be helpful if the technical novelty can be discussed more thoroughly.

**Summary Of The Paper:**

This paper studies offline reinforcement learning with differential function approximation (DFA) indexed by a parameter. Leveraging the ideas in DFA and the pessimism principle, the authors propose a value-iteration-based algorithm and analyze the suboptimality of the learned policy. The theoretical results generalize previous asymptotic analysis for DFA by constructing a finite-sample uncertainty quantification for the fitted value functions. A variant based on variance reduction is also provided and analyzed. The authors also discuss a few implications of this framework.

**Summary Of The Review:**

I find this paper to be good overall. It provides new algorithms and analyses that generalize the existing ones to the FDA setting, and is quite intensive and comprehensive, covering almost all aspects of value iteration with offline data. This is a solid work although I find the technical novelty to be not very pronounced.

---

> ### Author Response · Authors · 2022-11-18
> **Response to Reviewer tREF**
>
> - **It may be helpful if the technical novelty can be discussed more ...**
>     - Thank you for the question. Our generalization from linear to differentiable function approximation is technically challenging since the function would incorporate abundant function class with non-linear and non-convex structures, which makes a lot of analysis used in the linear case not work. We mention two of them.
>     - First of all, in the linear case, the ridge regression problem has a closed-form solution (echo line 4 of our Algorithm 1), and we don’t have the analytical solution for $\hat{\theta}$ in the differentiable function approximations. This makes our analysis much harder since for the linear case, the sub-optimality gap can be computed explicitly, while in our case, the gap $|f(\hat{\theta})-f(\theta^\star)|$ is not. The novel resolution is to use that $\hat{\theta}$ is a stationary point of the regression objective and work with the objective as a surrogate. This makes the whole algorithm work even though the solution $\hat{\theta}$ is not trackable.
>     - Second, to bound the parameter difference $\|\hat{\theta}-\theta^\star\|$, we use a novel reduction to general function approximation loss estimation $E_\mu[\ell(\hat{\theta})]-E_\mu[\ell(\theta_{TV})]$. This technique is totally new to offline learning tasks (the detail is in Section G). We also have concrete discussions about this right before section 3.1 in our paper.

---

### Official Review · Reviewer_4N32 · 2022-10-24

**Confidence:** 4
**Correctness:** 4
**Technical Novelty And Significance:** 3
**Empirical Novelty And Significance:** Not applicable
**Recommendation:** 6

**Clarity, Quality, Novelty And Reproducibility:**

- RHS of the equation in Corollary 3.3.: Remove the dot product <,> inside $\hat{f}$

- ”PFQL draws a unified view of model-based and model-free learning”: This is a bit unclear to me how PFQL is relevant ot model-based learning.

- It’s interesting to root of 1/4 in Theorem 3.1. Where it comes from?

- ”Assumption 2.3 reduces to 2.4” —> “Assumption 2.3 reduces to Example 2.4”

**Strength And Weaknesses:**

**Strengths**

- A generalization result of offline linear MDP to differentiable function class and incorporation of variance information

- The paper is well-written and easy to read

- The remark about feature representation and parameters looks interesting and can give some room to think more about the problem of representation learning in offline RL.

**Weaknesses**

- The data coverage assumption 2.3 is pretty some as it is equivalent to the uniform feature coverage assumption when realized to linear MDP, which is among the strongest data coverage assumption.

- Optimization oracle at line 4 of Algorithm 1 is pretty strong as obtaining the global minimizer for a non-convex differentiable objective is difficult unless it is an overparameterized neural nets (see also the next point)

- However, the dependence on $d$ in the learning bound makes their result inapplicable to overparameterized neural networks where $d$ could be a high-order polynomial of the number of samples

- The assumption about third-time differentiability also cannot apply to neural networks (not necessarily overparameterized) with non-smooth activation such as ReLU.

**Summary Of The Paper:**

This paper studies parametric offline rl with a class of general differentiable function class approximation (DFA). The differentiable function class they consider is the set of functions $f(\theta, \phi)$ whose argument can be decomposed into two parts: $\theta$ is the parameter vector and $\phi$ is the feature map. They further assume that $f$ is third-time differentiable with respect to $\theta$ and their derivatives are jointly continuous wrt $\theta$ and $\phi$. This function class is a generalization of linear model where $f$ is not necessarily dot product of its arguments.

Under two assumptions: (1) realizability + Bellman completeness, and (2)  the behavior policy is well exploratory over all dimensions of the gradient of DFA wrt $\theta$, they obtain a guarantee in a form that is similar to the linear MDP; though here they consider a much more general function class. Their bound scales linearly with d, H times the uncertainty quantifier across the trajectory of target policy. They also incorporate variance information to remove the factor $H$ in their learning bound.

**Summary Of The Review:**

See the strengths and weaknesses section.

---

> ### Author Response · Authors · 2022-11-18
> **Response to Reviewer 4N32**
>
> Thank you and we have removed all the minor clarity issues you mentioned.
>
> - **The data coverage assumption 2.3 is pretty some as it is equivalent to the uniform feature coverage assumption when realized to linear MDP, which is among the strongest data coverage assumption.**
>     - Yes, we agree with this and have mentioned this point in the limitation Section 5. Yet, as an initial study for differentiable function approximation, we stick with the curvature condition 2.3 for our analysis. Indeed, we are not aware of any algorithm that can achieve near-optimal guarantee without uniform coverage assumption beyond the tabular case. In the tabular case, optimal rate can be obtained with only single-policy concentrability assumption, but for linear MDPs, optimal results [1],[2] require uniform coverage to bound the residual terms so that lower orders term has a $\frac{1}{K}$ convergence rate.
>     - [1] Near-optimal Offline Reinforcement Learning with Linear Representation: Leveraging Variance Information with Pessimism, ICLR22.
>     - [2] Nearly Minimax Optimal Offline Reinforcement Learning with Linear Function Approximation: Single-Agent MDP and Markov Game, 2022.
> - **Optimization oracle at line 4 of Algorithm 1 is pretty strong as obtaining the global minimizer for a non-convex differentiable objective is difficult unless it is an overparameterized neural nets ...**
>     - Thank you. Solving the whole oracle (as stated in the algorithm) it is only for theoretical purpose/analysis. In practice, one could simply perform stochastic gradient updates for the objective function in Line 4 and stops when it converges. Finding the global optimal in a computationally efficient way for the non-convex differentiable objectives is a non-trivial question. This is an excellent question but it is not the focus of our study.
> - **However, the dependence on d in the learning bound ... inapplicable to overparameterized neural networks where d could be a high-order polynomial of the number of samples**
>     - Thanks. In our paper, we do not claim our results subsume the existing overparameterized neural network results. Our study relies on the Assumption 2.2 and 2.3 while overparametrized neural networks analysis relies on different set of assumptions (e.g. [3]). Also, the learning bounds of the two setups have different expressions. We leave drawing the connection between those two as future work.
>     - [3] Provably Efficient Offline Reinforcement Learning with Trajectory-Wise Reward. 2022
> - **The assumption about third-time differentiability also cannot apply to neural networks (not necessarily overparameterized) with non-smooth activation such as ReLU.**
>     - Thank you for the question. We thought about this when we draft this paper. The ReLu activation is not differentiable only at a single point. Therefore it is almost surely (3rd) differentiable. Our results still hold for the a.s. differentiable functions since our guarantees are high probability guarantees. Adding it rigorously will invlove more profound mathematical formulations. We didn’t include it since it will increase the over-technicality of the paper and it is not a major contribution.
>
> - **RHS of the equation in Corollary 3.3.: Remove the dot product $\langle,\rangle$ inside $\hat{f}$**
>
>     - We want to kindly mention that the dot product should be there by the definition of the generalized linear model.
>
>
> - **”PFQL draws a unified view of model-based and model-free learning”:**
>     - Standard value iteration algorithm requires the knowldge of the transion model $P$. Also, PFQL is a special version of **approximation value iteration** algorithm, and AVI can be considered as a model-based learning (check Model-based AVI section in [4])
>
>         [4] Error bounds for approximate value iteration, 2005.
>
> - **It’s interesting to root of 1/4 in Theorem 3.1. Where it comes from?**
>
>     - It is a common term that comes from the misspecification of the Bellman completeness. For example, check Theorem 5 of [5].
>
>       [5] Q* Approximation Schemes for Batch Reinforcement Learning: A Theoretical Comparison, 2020.

---

### Official Review · Reviewer_eR8G · 2022-10-26

**Confidence:** 4
**Correctness:** 4
**Technical Novelty And Significance:** 3
**Empirical Novelty And Significance:** Not applicable
**Recommendation:** 8

**Clarity, Quality, Novelty And Reproducibility:**

Overall, the writing of this paper is clear to readers. The novelty is significant. And the authors provide detailed proof which can reproduce the main theoretical results.

**Strength And Weaknesses:**

This paper provides novel theoretical results for the offline RL with function approximation. The authors of this paper study the differentiable function approximation, which is new to the area of offline RL, though some parts of the related theory are first studied by [Zhang et al. (2022)]. Their analysis is further improved based on the idea of variance awareness. Thus, the contribution is significant to this area.

There are also several questions regarding this submission:

1. Can the authors provide some explanations on why in Theorem 3.2 and Theorem 4.1, the results only hold for $K \geq K_0$? What is the technical consideration behind $K_0$. How are $\kappa_1$ and $\kappa_2$ defined?

2. Is it possible to have an improvement on other factors ( e.g., $d$ ) beyond only an $H$ in the result in Theorem 4.1 based on the idea of variance awareness?


**Summary Of The Paper:**

This paper studies offline reinforcement learning with differentiable function class approximation (DFA). This function class incorporates a wide range of models with nonlinear/nonconvex structures. The authors show that offline RL with DFA is provably efficient via analyzing the pessimistic fitted Q-learning (PFQL) algorithm. They further improve the guarantee with a tighter instance-dependent characterization.

**Summary Of The Review:**

The technical contribution of this paper is significant. It provides novel ideas in the theoretical analysis of offline RL.

---

> ### Author Response · Authors · 2022-11-18
> **Response to Reviewer 7cjK**
>
> We thank the reviewer for the encouraging feedback and for understanding the value of our paper.
>
> - **What is the technical consideration behind $K_0$. How are $\kappa_1$ and $\kappa_2$ defined?**
>
>     - Thank you for the great technical question! The technical requirement comes from applying Lemma H.5 of Min et al. (2021), which requires $K$ to be larger than certain threshold (the condition is in equation (35)). The purpose of using Lemma H.5 is that we can bound the data-dependent quantity $||u||_{\bar{G}^{-1}}$
>
>         by the $\frac{1}{\sqrt{K}}||u||_{G^{-1}}$ (for details, see the proof of Lemma L.5.).
>
>     - This is the key for further analysis since $G^{-1}$ is population level quantity that does not involve randomness. Dealing with quantity $\frac{1}{\sqrt{K}}\|u\|_{G^{-1}}$ is more convenient for the analysis purpose since it does not require covering.
>     - $\kappa_2$ is defined as the upper bound of the 2-norm of the second-order derivate and $\kappa_3$ is defined as the upper bound of the 2-norm of the third-order derivate.
> - **Is it possible to have an improvement on other factors ( e.g., d ) beyond only an H in the result in Theorem 4.1 based on the idea of variance awareness?**
>     - Thanks. The power of variance awareness is to replace the standard factor $H^2$ by the $Var_P(V^\star)$. We are not aware whether this can be useful for improving the parameter dimension, but we believe with the existing technique the answer is no since the sharp tool like self-normalized martingale will incur at least a $\sqrt{d}$ dependence. This is a good question and we leave it as future work.

---

### Official Review · Reviewer_7cjK · 2022-10-27

**Confidence:** 3
**Correctness:** 3
**Technical Novelty And Significance:** 3
**Empirical Novelty And Significance:** Not applicable
**Recommendation:** 6

**Clarity, Quality, Novelty And Reproducibility:**

I find some of the claims in this paper confusing:

- The paragraph after assumption 2.2 states that “we make the following stronger coverage assumption …”. Why the following coverage assumption is stronger? Does Assumption 2.3 imply Assumption 2.2?
- The “Faster convergence in learning” paragraph (Page 7) uses the example that $\nabla f(\theta^\star,\phi)=0$ to demonstrate the faster rate. Does this statement requires  $\nabla f(\theta^\star,\phi(s,a))=0$ for every possible $s,a$? Does this example contradict with Assumption 2.3 (in particular, $\mathbb{E}[\nabla f(\theta^\star,\phi(s,a)) \nabla f(\theta^\star,\phi(s,a))^\top]\ge \kappa I$?
- The “Feature representation vs. Parameters” paragraph (Page 7) states that “when changing the model f with more complex representations, the learning hardness will not grow as long as the number of parameters needs to be learned does not increase”. It’s unclear to me what is the implication of this statement. It seems to me that this statement exactly means that the complexity of the function class does not increase with the feature dimension (when the number of parameters is fixed), so increasing the feature dimension further does not increase the expressivity of the function class fundamentally.
- It’s unclear to me why there is an extra $\sqrt{d}$ factor in Theorem 4.1. The conclusion section suggests that it comes from the covering argument for the differentiable function class. Could the authors elaborate? In particular, why can’t the uniform convergence techniques used in linear MDPs be applied here?


**Strength And Weaknesses:**

Strength:
- The differentiable function class considered in this paper generalizes linear/generalized linear models. The sample complexity bounds for generalized linear models in this setting seem to be new.

Weakness:
- The two equations in Assumption 2.3 is required for every possible parameter $\theta$, which seems to be very strong. It’s unclear to me whether there are other function classes (beyond linear and generalized linear models) that can satisfy Assumption 2.3. For example, does a two-layer neural network with constant number of neurons satisfy Assumption 2.3?


**Summary Of The Paper:**

This paper studies offline RL with a differentiable function class. With the assumption that the covariance of the gradient of every function (w.r.t. the parameters) in the function class has a positive minimum eigenvalue, this paper proves sample complexity bounds that scale with the number of the parameters. The results in this paper recover existing results for linear models and generalized linear models (up to an additional $\sqrt{d}$ factor in the instance-dependent bound).

**Summary Of The Review:**

My main concern is whether the differentiable function approximation setting considered in this paper contains new settings that is unknown in prior works. However, given that the results in this paper require some novel techniques, I will recommend a weak accept at this point.

=== after rebuttal ===
After reading through the authors' response, I tend to keep my score as weak accept.

---

> ### Author Response · Authors · 2022-11-18
> **Response to Reviewer 7cjK**
>
> * **The paragraph after assumption 2.2 states that “we make the following stronger coverage assumption …”. Why the following coverage assumption is stronger? Does Assumption 2.3 imply Assumption 2.2?**
>
> Thank you for the comment and you are right! Concretely, Assumption 2.2 defines the concentrability coefficient that solely depends on the Markov Decision Process, and the Uniform Coverage in Assumption 2.3 depends on both the MDP and the function class $\mathcal{F}$. They are not directly comparable. However, for some special cases such as tabular MDPs, if we specify the function model $f(\theta,\phi)=\langle \theta,\phi\rangle$ and $\phi(s,a)=\mathbf{1}(s,a)$, then Assumption 2.3 reduces to $\min_{s,a}d^\mu(s,a)\geq \kappa$, and this implies the concentrability coefficient $C_{eff}\leq 1/\kappa$ (therefore in this case 2.3 is stronger than 2.2). We have modified the statement in the revision.
>
> * **The “Faster convergence in learning” paragraph (Page 7) uses the example.... . Does this statement requires for every possible s,a ?**
>
> Thank you for the good catch and that’s correct. In fact, the emphasis of this paragraph is to describe the instance-dependent feature of the Fisher-Information style quantity that is characterized by the gradient $\nabla f(\theta^\star_h,\phi)$. For different MDP instances $M_1$, $M_2$, the coupled $\theta^\star_{h,M_1},\theta^\star_{h,M_2}$ will generate different performances for $M_1$, $M_2$. Standard worst-case bounds (from GFA approximation) cannot explicitly differentiate between problem instances. We have removed the previous example in the revision and thanks for pointing out this.
>
> * **The “Feature representation vs. Parameters” paragraph (Page 7) states that “when changing the model f with more complex representations, the learning hardness will not grow as long as the number of parameters needs to be learned does not increase”. It’s unclear to me what is the implication of this statement.**
>
> Thank you. We are not driving any conclusion/implication here but rather provide our observation that “learning complexity only (explicitly) depends on the parameter dimension”, as we stated in the first line of the paragraph. Understanding the relationship between parameters and features is an interesting question to us, but the systematic study of this phenomenon is beyond the scope of this research. We made this clear in our revision.
>
> * **It’s unclear to me why there is an extra $\sqrt{d}$ factor in Theorem 4.1. The conclusion section suggests that it comes from the covering argument for the differentiable function class. Could the authors elaborate? In particular, why can’t the uniform convergence techniques used in linear MDPs be applied here?**
>
> - This is a great in-depth technical question! Indeed, we are using the same self-normalized concentration here for our problem, but, since the differentiable function class is a much richer class than the linear function class, we need to do an extra level of covering, which causes this extra $\sqrt{d}$ factor.
> - Concretely, if $\theta$ is fixed, then we can conduct only Step1 of Lemma E.7, which will have only $\sqrt{d}$ dependence.
> - However, since $\hat{\theta}_h$ is data-dependent, for the differentiable function we need to cover $x_k(\hat{\theta}_h)$ and $G(\hat{\theta}_h)$ (in Lemma E.7.) as well, so the extra $\sqrt{d}$ pops out.
> - Note for linear function approximation, this extra covering is not required since $x_k(\hat{\theta}_h)\equiv \phi$ and also $G(\hat{\theta}_h)$ do not depend on $\theta$ (the gradient at different $\theta$ location is identical due to the linear structure), therefore making it an easier question [1],[2].
> - [1] Near-optimal Offline Reinforcement Learning with Linear Representation: Leveraging Variance Information with Pessimism, ICLR22.
> - [2] Nearly Minimax Optimal Offline Reinforcement Learning with Linear Function Approximation: Single-Agent MDP and Markov Game, 2022.

---

> > ### Comment · Reviewer_7cjK · 2022-11-26
> > **Follow-up questions**
> >
> > Thanks for addressing most of my concerns. Could you also comment on the question raised in the weakness section?
> >
> > > It’s unclear to me whether there are other function classes (beyond linear and generalized linear models) that can satisfy Assumption 2.3. For example, does a two-layer neural network with constant number of neurons satisfy Assumption 2.3?

---

> > > ### Author Response · Authors · 2022-11-28
> > > **RE**
> > >
> > > Hello Reviewer 7cjK, thank you for the important question. It depends on formulation of the neural network model. Since we have known feature representation in the model, if we formulate all the parameters in the output layer of NN as parameters and the previous layers as the known (pre-trained) representation, then this can be satisfied by 2.3 (since this is essentially a linear model). However, it is hard to verify for the case where all the parameters are considered for training, and existing analysis for (over-parameterized) NN uses different of set of assumptions than 2.3 (NTK requires non-parametric assumptions and we are the parametric class). Relaxing 2.3 would make the result adapt to more differentiable function classes, but as an initial study we don't know the answer and we leave it as future study. Thank you again for the great question.

---

### Decision · Program_Chairs · 2023-01-20

**Decision:**

Accept: poster

**Justification For Why Not Higher Score:**

Although the paper is well-written and significant or important contribution which would hopefully open more research in this direction. The results still only applies to generalized linear models only. The results are only theoretical for now.

**Justification For Why Not Lower Score:**

The paper is a novel contribution to an important growing area of research. Possibly opens up new research in this direction. Thus it is worth publishing at ICLR 2023.

**Metareview: Summary, Strengths And Weaknesses:**

## Summary
This paper provides a systematic study of understanding of the statistical complexity for offline RL with differentiable function approximation (DFA). The paper analyses the *pessimistic fitted Q-learning* (PFQL) algorithm and show that PFQL with DFA is provably efficient. The paper improves the sample-complexity with horizon dependence but the sample complexity still depends on $d$. The paper focuses on theoretical results without any empirical verification.

*Decision:* Overall the feedback from the reviewers were positive. The initial reviews were all leaning towards acceptance. It seems like authors have done a good job on addressing some of the feedback and concerns raised by the reviewers during the rebuttal. I think this work is a valuable contribution to the ICLR community.

## Strengths
- The theoretical results on sample complexity with DAF for ORL is novel and a valuable contribution.
- The paper is well-written .
- Opens up a pathway for further research on problem of representation learning in offline RL.

## Weaknesses
- Some unclear statements pointed out by Reviewer 7cjK and Reviewer eR8G which are mainly addressed by the authors in the final revision.
- The data coverage assumption 2.3 is very strong.
- The theoretical results in this paper don't apply to existing overparameterized neural networks.



**Note From Pc:**

if the above contains the word "oral" or "spotlight" please see: "oral" presentation means -> notable-top-5% and "spotlight" means -> notable-top-25%. As stated in our emails, we are disassociating presentation type from AC recommendations